# Uncertainty quantification of the multi-centennial response of the Antarctic Ice Sheet to climate change

Kevin Bulthuis[1,2], Maarten Arnst[1], Sainan Sun[2], and Frank Pattyn[2]

[1]Computational and Stochastic Modeling, Aerospace and Mechanical Engineering, Université de Liège, Allée de la Découverte 9, Quartier Polytech 1, 4000 Liège, Belgium

[2]Laboratoire de Glaciologie, Department of Geosciences, Environment and Society, Université Libre de Bruxelles, Av. F.D. Roosevelt 50, 1050 Brussels, Belgium

**Correspondence:** Kevin Bulthuis (kevin.bulthuis@uliege.be)

**Abstract.** Ice loss from the Antarctic ice sheet (AIS) is expected to become the major contributor to sea level in the next centuries. Projections of the AIS response to climate change based on numerical ice-sheet models remain challenging due to the complexity of physical processes involved in ice-sheet dynamics, including instability mechanisms that can destabilise marine basins with retrograde slopes. Moreover, uncertainties in ice-sheet models limit the ability to provide accurate sea-level rise projections. Here, we apply probabilistic methods to a hybrid ice-sheet model to investigate the influence of several sources of uncertainty, namely sources of uncertainty in atmospheric forcing, basal sliding, grounding-line flux parameterisation, calving, sub-shelf melting, ice-shelf rheology and bedrock relaxation, on the continental response of the Antarctic ice sheet to climate change over the next millennium. We provide probabilistic projections of sea-level rise and grounding-line retreat and we carry out stochastic sensitivity analysis to determine the most influential sources of uncertainty. We find that all investigated sources of uncertainty, except bedrock relaxation time, contribute to the uncertainty in the projections. We show that the sensitivity of the projections to uncertainties increases and the contribution of the uncertainty in sub-shelf melting to the uncertainty in the projections becomes more and more dominant as atmospheric and oceanic temperatures rise, with a contribution to the uncertainty in sea-level rise projections that goes from 5–25 % in RCP 2.6 to more than 90 % in RCP 8.5. We show that the significance of the AIS contribution to sea level is controlled by the marine ice-sheet instability (MISI) in marine basins, with the biggest contribution stemming from the more vulnerable West Antarctic ice sheet. We find that, irrespectively of parametric uncertainty, the strongly mitigated RCP 2.6 scenario prevents the collapse of the West Antarctic ice sheet, that in both RCP 4.5 and RCP 6.0 scenarios the occurrence of MISI in marine basins is more sensitive to parametric uncertainty and that, almost irrespectively of parametric uncertainty, RCP 8.5 triggers the collapse of the West Antarctic ice sheet.

# 1 Introduction

The Antarctic ice sheet (AIS) is the largest reservoir of freshwater on Earth ($\sim$60 m sea-level equivalent (Fretwell et al., 2013; Vaughan et al., 2013)) and has the potential to become one of the largest contributors to sea level in the next centuries. Yet, studies, such as the IPCC Fifth Assessment Report (AR5) (Stocker et al., 2013) and Kopp et al. (2014), have identified the
magnitude of the AIS response as the largest source of uncertainty in projecting sea-level rise, when compared with other contributors, such as thermal expansion, glaciers and the Greenland ice sheet. Recent observations (Shepherd et al., 2018) have shown an acceleration in the rate of ice loss from the Antarctic ice sheet, especially in West Antarctica where an irreversible retreat may be underway in the Amundsen Sea Embayment as a consequence of a marine ice-sheet instability (MISI) (Favier et al., 2014; Joughin et al., 2014; Rignot et al., 2014).

Assessing the future response of the Antarctic ice sheet requires numerical ice-sheet models amenable to large-scale and long-term simulations and quantification of the impact of modelling hypotheses and parametric uncertainty. So far, there exist only a limited number of projections (Golledge et al., 2015; Ritz et al., 2015; DeConto and Pollard, 2016; Schlegel et al., 2018) for the whole Antarctic ice sheet on a (multi-)centennial time scale. These projections show similar trends for the next centuries, but they differ in the magnitude of the mass loss they predict, with differences and uncertainty ranges that can
reach several metres for eustatic sea level. Using a numerical ice-sheet model supplemented with a statistical approach for the probability of MISI onset, Ritz et al. (2015) have projected a contribution to sea level around 0.1 m by 2100 with a very low probability of exceeding 0.5 m in the A1B scenario. Running their simulations with and without sub-grid melt interpolation at the grounding line, Golledge et al. (2015) have projected that sea-level rise could range from 0.01 m to 0.38 m by 2100 and from 0.21 m to more than 5 m by 2500 considering all RCP scenarios. Including the marine ice-cliff instability mechanism (Pollard
et al., 2015) in a numerical ice-sheet model, DeConto and Pollard (2016) have projected that sea level could rise much more significantly, with projections exceeding 1 m by 2100 and 15 m by 2500 in the RCP 8.5 scenario. Synthesising recent results for the expected response of the Antarctic ice sheet to climate change, Pattyn et al. (2018) have highlighted that the projected contribution to sea level of the Antarctic ice sheet is limited to well below one metre by 2500 in the RCP 2.6 scenario while a key threshold for the stability of the Antarctic ice sheet is expected to lie between 1.5 and 2 °C mean annual air temperature
above present and the activation of larger ice systems, such the Ross and Ronne-Filchner drainage basins, could be triggered by global warming between 2 and 2.7 °C. Emulating the numerical ice-sheet model by DeConto and Pollard (2016) with a Gaussian process emulator, Edwards et al. (2019) have established new probabilistic projections for the AIS contribution to sea level by 2100, with the probability to exceed 0.5 m that reaches 4 % in RCP 2.6 and 71 % in RCP 8.5 when considering the marine ice-cliff instability mechanism. Coupling an ice-sheet model with a climate model, Golledge et al. (2019) have
shown that freshwater released by the Antarctic ice sheet can trap warm waters below the sea surface, thus leading to higher projections by 2100, with a contribution to sea level that could reach 0.05 m in RCP 4.5 and 0.14 m in RCP 8.5.

Despite recent progress in the numerical modelling of ice-sheet dynamics (Pattyn et al., 2017), differences in modelling hypotheses between ice-sheet models remain a major source of uncertainty for sea-level rise projections. Intercomparison projects such as the SeaRise project (Bindschadler et al., 2013) and the ongoing ISMIP6 project (Nowicki et al., 2016; Goelzer

et al., 2018) aim at quantifying the impact of such so-called structural uncertainty in ice-sheet models by comparing projections from multiple numerical models. Such multimodel ensembles give some insight into the impact of the structural uncertainty, although challenges remain in combining the different results (Knutti et al., 2010).

Uncertainties in the ice-sheet initial state, climate forcing and parameters in numerical ice-sheet models are another major
limitation for accurate projections. To date, the impact of such parametric uncertainty is assessed most often by using large ensemble analysis, that is, the model is run for different values of the parameters and the uncertainty in the projections is estimated from the spread in the model runs. For example, Golledge et al. (2015) ran their model with simplified sensitivity experiments to evaluate qualitatively the sensitivity of the Antarctic ice sheet to temperature, precipitation and sea surface temperature individually, while DeConto and Pollard (2016) assessed the sensitivity of their model to parametric uncertainty
by evaluating the model response at a few samples in the parameter space. By contrast, Ritz et al. (2015) adopted a probabilistic approach and estimated the uncertainty in sea-level rise projections by running an ice-sheet model with samples of parameters drawn randomly from probability density functions for the parameters and subsequently deducing probability density functions for sea-level rise projections.

The field of uncertainty quantification (UQ) develops theory and methods to describe quantitatively the origin, propagation
and interplay of sources of uncertainty in the analysis and projection of the behaviour of complex systems in science and engineering; see, for instance, Ghanem et al. (2017) for a recent handbook and Arnst and Ponthot (2014) for a recent review paper. Most of this theory and these methods are based on probability theory, in the context of which uncertain parameters and projections are represented as random variables characterised by their probability density function. Theory and methods are under development to characterise sources of uncertainty by probability density functions inferred from observational data and
expert assessment (characterisation of uncertainty), to deduce the impact of sources of uncertainty on projections (propagation of uncertainty) and to ascertain the impact of each source of uncertainty on the projection uncertainty and rank them in order of significance (stochastic sensitivity analysis). These developments have led to new theories and new methods that are of interest to be applied to uncertainty quantification of ice-sheet models, beyond the theory and methods that Golledge et al. (2015), Ritz et al. (2015), DeConto and Pollard (2016) and Schlegel et al. (2018) have already applied.

In this paper, we apply probabilistic methods to assess the impact of uncertainties on the continental AIS response over the next millennium. We use the fast Elementary Thermomechanical Ice Sheet (f.ETISh) model (Pattyn, 2017), a hybrid ice-sheet model that captures the essential characteristics of ice flow and allows large-scale and long-term projections at a reasonable computational cost. To reduce the computational cost of assessing the impact of uncertainties on the change in global mean sea level (GMSL), we draw from UQ methods based on the construction of an emulator (Le Maître and Knio, 2010; Ghanem
et al., 2017), also known as a surrogate model, that is, a computational model that mimics the response of the original ice-sheet model at a reduced computational cost. To assess the significance of each individual source of uncertainty in inducing uncertainty in the change in GMSL, we draw from UQ methods for stochastic sensitivity analysis (Saltelli et al., 2008). To express the uncertainty in projections of the retreat of grounded ice, we draw from UQ methods for constructing confidence regions for excursion sets (Bolin and Lindgren, 2015; French and Hoeting, 2015), with a confidence region for grounded ice
interpreted as a region of Antarctica that remains covered everywhere with grounded ice for a given level of probability.

On the one hand, our study adds to previous studies (Golledge et al., 2015; Ritz et al., 2015; DeConto and Pollard, 2016) that also provided projections for the multi-centennial response of the whole Antarctic ice sheet: whereas these previous studies provided projections by running the ice-sheet model with samples of parameters from the parameter space, our study differs by the adoption of additional methods from UQ and the analysis of a broader set of parameters that includes uncertainty in

climate forcing, basal sliding, grounding-line flux parameterisation, calving, sub-shelf melting, ice-shelf rheology and bedrock relaxation. On the other hand, our study also adds to previous studies (Pollard et al., 2016; Schlegel et al., 2018; Edwards et al., 2019) that also applied methods from UQ for uncertainty propagation in ice-sheet models: whereas these previous studies analysed AIS paleoclimatic responses with Gaussian process (kriging) emulators (Pollard et al., 2016) and multi-decadal forecasts with Latin hypercube sampling (Schlegel et al., 2018), our study differs by its focus on the analysis of the multi-

centennial response of the whole Antarctic ice sheet with polynomial emulators; and in addition to uncertainty propagation, we complement our uncertainty analysis with stochastic sensitivity analysis and confidence regions.

This paper is organised as follows. First, Sect. 2 describes the f.ETISh model, the uncertain processes and parameters and the UQ methods, including the use of an emulator, stochastic sensitivity analysis and the construction of confidence regions. Subsequently, Sect. 3 shows and interprets the results for the multi-centennial response of the Antarctic ice sheet as well as the

advantages of the adopted UQ methods. Finally, Sect. 4 provides an overall discussion of the results.

## 2  Model description and methods

### 2.1  Ice-sheet model and simulations

We perform simulations of the response of the Antarctic ice sheet (Fig. 1a) to environmental and parametric perturbations over the period 2000–3000 CE with the fast Elementary Thermomechanical Ice Sheet (f.ETISh) model (Pattyn, 2017) version 1.2.

The f.ETISh model is a vertically integrated, thermomechanical, hybrid ice-sheet/ice-shelf model that incorporates essential characteristics of ice-sheet thermomechanics and ice-stream flow, such as the mass-balance feedback, bedrock deformation, sub-shelf melting and calving. The ice flow is represented as a combination of the shallow-ice (SIA) (Hutter, 1983; Greve and Blatter, 2009) and shallow-shelf (SSA) (Morland, 1987; MacAyeal, 1989; Weis et al., 1999) approximations for grounded ice (Bueler and Brown, 2009), while only the shallow-shelf approximation is applied for floating ice shelves. Bedrock deformation

is represented as a combined time-lagged asthenospheric relaxation and elastic lithospheric response (Greve and Blatter, 2009; Pollard and DeConto, 2012a), in which the lithosphere relaxes towards isostatic equilibrium due to the viscous properties of the underlying asthenosphere. Calving at the ice front is parameterised based on the large-scale stress field, represented by the horizontal divergence of the ice-shelf velocity field (Pollard and DeConto, 2012a). Prescribed input data include the present-day ice-sheet geometry and bedrock topography from the Bedmap2 dataset (Fretwell et al., 2013), the basal sliding coefficient,

and the geothermal heat flux by An et al. (2015).

We perform simulations at a spatial resolution of 20 km while accounting for grounding-line migration at coarse resolution with a flux condition derived from a boundary layer theory at steady state based on either a Weertman (or power) friction law (Schoof, 2007b) or a Coulomb friction law (Tsai et al., 2015) at the grounding line. This flux condition at the grounding

line is imposed as an internal boundary condition following the implementation by Pollard and DeConto (2009, 2012a). This implementation has been shown to reproduce the migration of the grounding line and its steady-state behaviour (Schoof, 2007a) at coarse resolution for the SSA and hybrid SSA/SIA models, while these models can only reproduce the migration of the grounding line and its steady-state behaviour at sufficiently fine resolution in the absence of a flux condition (Docquier et al., 2011; Pattyn et al., 2012). Numerical simulations (Pollard and DeConto, 2012a; DeConto and Pollard, 2016) of the Antarctic ice sheet using a flux condition have also been able to reproduce MISI in large-scale ice-sheet simulations. In addition, Pattyn (2017) has shown that a flux condition makes the f.ETISh model rather independent of the resolution for a spatial resolution of the order of 20 km. While the implementation of a flux condition at the grounding line has been shown to reproduce qualitatively ice-sheet dynamics as determined with other ice-sheet models, it cannot reproduce quantitatively changes in ice-sheet mass and the contribution to sea level as determined from models with a higher level of complexity, such as the Blatter–Pattyn (Blatter, 1995; Pattyn, 2003) and the full Stokes (Greve and Blatter, 2009) models that include vertical shearing at the grounding line, especially for short transients (decadal time scales) as these flux conditions have been derived at steady state (Drouet et al., 2013; Pattyn et al., 2013; Pattyn and Durand, 2013). A proper representation of grounding-line migration without the need for a flux condition would require a very fine grid resolution (possibly less than 200 metres). In addition, the flux condition has been derived for unbuttressed ice shelves (Schoof, 2007b) and may fail to appropriately represent grounding-line migration for buttressed ice shelves (Reese et al., 2018b) as found mostly around Antarctica even when the flux condition is corrected for buttressing with a buttressing factor accounting for back stress at the grounding line. In particular, Pattyn et al. (2013) have shown that models that implement the flux condition cannot reproduce compressional stresses that may apply in the presence of buttressing. Moreover, using a spatial resolution of 20 km does not allow to capture properly certain mechanisms that control grounding-line migration such as bedrock irregularities and ice-shelf pinning points even with sub-grid parameterisations of these mechanisms. Therefore, we may expect discrepancies between our results and results at higher spatial resolutions ($< 5$ km), especially for important small ice streams such as Pine Island and Thwaites glaciers, which represent only a few grid points with a 20-km resolution. Despite the limitations of the flux condition and our coarse grid resolution, we have adopted these modelling assumptions as an efficient way to capture the essential mechanisms of grounding-line migration in large-scale, long-term and large-ensemble ice-sheet simulations while keeping the computational cost tractable. The computational cost of a single forward simulation with a time step of 0.05 years on the CÉCI clusters (F.R.S-FNRS & Walloon Region) on 2 threads is approximately 8 h. To investigate the impact of the spatial resolution on the results, we performed additional runs at a spatial resolution of 16 km (Fig. S1). We found that the uncertainty in the projections due to the spatial resolution is (far) less important than the uncertainty due to the uncertainty in the parameters.

The main changes in f.ETISh version 1.2 as compared to version 1.0 (Pattyn, 2017) consist of the computation of sub-shelf melt rates with an ocean-model coupler based on the Potsdam Ice-shelf Cavity mOdel (PICO) ocean-model coupler (Reese et al., 2018a) rather than simpler parameterisations of sub-shelf melt rates (Beckmann and Goosse, 2003; Holland et al., 2008; Pollard and DeConto, 2012a; de Boer et al., 2015; Cornford et al., 2016), the inclusion of a laterally varying flexural rigidity for the lithosphere (Chen et al., 2018) to compute glacial isostatic adjustment more realistically with an elastic lithosphere/relaxed asthenosphere model, an improvement of the SSA numerical scheme based on the implementation by Rommelaere and Ritz

(1996) and a description of atmospheric forcing based on a parameterisation of the changes in atmospheric temperature and precipitation rate (Huybrechts et al., 1998; Pollard and DeConto, 2012a), a parameterisation of surface melt with a positive degree-day model (Janssens and Huybrechts, 2000) and the inclusion of meltwater percolation and refreezing (Huybrechts and de Wolde, 1999).

We drive our simulations with both atmospheric and oceanic forcings. Present-day mean surface air temperature and precipitation are obtained from Van Wessem et al. (2014), based on the regional atmospheric climate model RACMO2. Changes in atmospheric temperature and precipitation rate induced by a forcing temperature change $\Delta T$ are applied in a parameterised way that accounts for elevation changes (Huybrechts et al., 1998; Pattyn, 2017). We use a positive degree-day (PDD) model to calculate surface melt (Janssens and Huybrechts, 2000), assuming a PDD factor of 3 mm$°$C$^{-1}$day$^{-1}$ and 8 mm$°$C$^{-1}$day$^{-1}$

for snow and ice, respectively. The PDD model also includes meltwater percolation and refreezing (Huybrechts and de Wolde, 1999).

    Basal melting underneath ice shelves is determined from the PICO ocean-model coupler (Reese et al., 2018a), which evaluates sub-shelf melting from the ocean temperature $T_{oc}$ and salinity fields on the continental shelf via an ocean box model that captures the basic overturning circulation within ice-shelf cavities. We employ data by Schmidtko et al. (2014) for present-day

ocean temperature $T_{oc}^{obs}$ and salinity on the continental shelf. For a change in background atmospheric temperature $\Delta T$, we assume that the ocean temperature $T_{oc}$ on the continental shelf changes as

$$T_{oc} = T_{oc}^{obs} + F_{melt}\Delta T, \tag{1}$$

where $F_{melt}$ is an ocean melt factor that represents the ratio between oceanic and atmospheric temperature changes (Maris et al., 2014; Golledge et al., 2015). Equation (1) with $F_{melt} = 0.25$ has been shown to reproduce trends in ocean temperatures

following an analysis of the Climate Model Intercomparison Project phase 5 (CMIP5) data set (Taylor et al., 2012) for changes in atmospheric and ocean temperatures (Golledge et al., 2015). Figure 1b shows the initial sub-shelf melt rate as computed with the PICO model. The PICO model is able to reproduce the general pattern of sub-shelf melting with higher melt rates near the grounding line and lower melt rates near the calving front. Sub-shelf melt rates are also higher in the Amundsen Sea sector and lower underneath the large Ross and Filchner–Ronne ice shelves.

The calibration of the basal sliding coefficient follows the data assimilation method of ice-sheet geometry by Pollard and DeConto (2012b). This approach is based on a fixed-point iteration scheme that adjusts the basal sliding coefficient iteratively so as to match the present-day ice-sheet configuration while assuming that the present-day configuration is in steady state. After applying this approach, we further adjust the basal sliding coefficient in ice streams with a sliding multiplier factor similar to Bindschadler et al. (2013) to reduce the initial drift. We carry out the calibration step independently for the different sliding

and grounding-line flux conditions investigated in this paper. For the nominal values of the model parameters and without forcing anomaly, the initial drifts (reference runs) range from 0.1 to 0.2 metre of sea-level rise by 3000. To make our analysis insensitive to model initialisation, we correct all our results for this initial bias by subtracting the reference runs from the results. Therefore, after corrections for present-day dynamical changes, our results reflect the model response to perturbations.

These corrections have in general little impact on medium-term and long-term projections as the initial drift is small compared to the projections but leads to a more significant bias for short-term projections.

## 2.2 Sources of uncertainty

We aim at quantifying the response of the Antarctic ice sheet to climate change while accounting for uncertainty in atmospheric forcing, basal sliding, grounding-line flux parameterisation, calving, sub-shelf melting, ice-shelf rheology and bedrock relaxation. We account for uncertainty in these physical processes by introducing uncertainty in parameters in the f.ETISh model (see Table 1). Here, we choose the uncertainty ranges for the parameters sufficiently large so as to encompass extreme conditions. The effect of choosing narrower uncertainty ranges for the parameters is discussed in Sect. 3.6. We assume the parameters to take the same value everywhere over the whole Antarctic ice sheet. This approach is likely to be conservative as the parameters may vary at least regionally. We refer to Schlegel et al. (2018) for a discussion about uncertainty quantification based on a partitioning of the ice-sheet domain. In the remainder of this section, we list the uncertain parameters and briefly discuss their influence on the AIS response.

**Table 1.** List of parameters and parameter ranges used in the uncertainty analysis.

| Definition | Parameter | Nominal | Min | Max | Units |
|---|---|---|---|---|---|
| Sliding exponent | $m$ | 2 | 1 | 5 | |
| Calving multiplier factor | $F_{\mathrm{calv}}$ | 1 | 0.5 | 1.5 | |
| Ocean melt factor | $F_{\mathrm{melt}}$ | 0.3 | 0.1 | 0.8 | |
| Shelf anisotropy factor | $E_{\mathrm{shelf}}$ | 0.5 | 0.2 | 1 | |
| East Antarctic bedrock relaxation time | $\tau_{\mathrm{e}}$ | 3000 | 2500 | 5000 | yr |
| West Antarctic bedrock relaxation time | $\tau_{\mathrm{w}}$ | 3000 | 1000 | 3500 | yr |

### 2.2.1 Atmospheric forcing

Alongside oceanic forcing, atmospheric forcing is generally considered to be the primary driver of future changes in the AIS mass balance (Lenaerts et al., 2016; Pattyn et al., 2018). Simulating atmospheric forcing over several centuries with regional climate models is computationally prohibitive. In addition, the accuracy of the climate models is limited by uncertainties such as the possible trajectories of anthropogenic greenhouse gas emissions. Therefore, we adopt here the four schematic extended representative concentration pathway (RCP) scenarios RCP 2.6, RCP 4.5, RCP 6.0 and RCP 8.5 introduced by Golledge et al. (2015) for temperature changes $\Delta T$ in the atmosphere (Fig. 2) as a means of representing different atmospheric forcings relevant for policymakers.

The scenario plays a significant role in the amplitude and speed of the AIS retreat. Recent studies (Golledge et al., 2015; DeConto and Pollard, 2016) have shown no substantial retreat of the grounding-line position in the strongly mitigated RCP 2.6 scenario. The other scenarios lead to a reduction in the extent of the major ice shelves (Ross, Filchner–Ronne and Amery

ice shelves) within 100–300 years, leading to accelerated grounding-line recession due to reduced buttressing. DeConto and Pollard (2016) have also highlighted that the hydrofracturing and ice-cliff failure mechanisms (not included in the f.ETISh model version 1.2) driven by increased surface melt and sub-shelf melting could potentially lead to an accelerated collapse of the West Antarctic ice sheet and a deeper grounding-line retreat in the East Antarctic subglacial marine basins.

### 2.2.2 Basal sliding

Basal sliding controls the motion of fast-flowing ice streams, which drain about 90 % of the total Antarctic ice flux (Bennett, 2003). Several studies have shown the importance of basal sliding on the behaviour of ice streams and stressed the need for the understanding of physical processes at play at the ice-bedrock interface (Joughin et al., 2009, 2010; Ritz et al., 2015; Brondex et al., 2017, 2019). In particular, Ritz et al. (2015) have shown that, under a power-law rheology for basal sliding, the
contribution to future sea level is an increasing function of the sliding exponent.

We introduce basal sliding as a Weertman sliding law, that is,

$$\boldsymbol{v}_b = -A_b \|\boldsymbol{\tau}_b\|^{m-1} \boldsymbol{\tau}_b, \qquad (2)$$

where $\boldsymbol{\tau}_b$ is the basal shear stress, $\boldsymbol{v}_b$ the basal velocity, $A_b$ the basal sliding coefficient and $m$ a sliding exponent. The sliding exponent is often related to Glen's flow law exponent $n$ as $m = n$ for sliding over hard bedrock (Weertman, 1957). The value
$m = 3$, related to the usual value $n = 3$ for Glen's flow law exponent, has been applied in a number of studies (Schoof, 2007a; Pattyn et al., 2012, 2013; Brondex et al., 2017; Gladstone et al., 2017) but the value $m = 1$ has also been commonly used in ice-flow models (Larour et al., 2012; Schäfer et al., 2012; Gladstone et al., 2014; Yu et al., 2018).

In addition to the usual exponents $m = 1$ and $m = 3$, we consider for the sliding exponent a nominal value of 2 as an intermediate sliding condition between the two usual exponents used in ice-flow models. Hereafter, we refer to the exponents
$m = 1, 2$ and 3 as the viscous (or linear), the weakly nonlinear and the strongly nonlinear sliding law, respectively. We consider the discrete values $m = 1, 2$ and 3 as representative of the most common values in large-scale ice-sheet modelling and discuss in Sect. 3.7 the impact of a more plastic sliding law ($m = 5$) to represent a quasi-plastic deformation of the till in ice streams (Gillet-Chaulet et al., 2016).

### 2.2.3 Grounding-line flux parameterisation

The f.ETISh model employs a parameterisation of the grounding-line flux based on a boundary layer theory at steady state by either Schoof (2007b) (SGL) or Tsai et al. (2015) (TGL). For the TGL parameterisation, Pattyn (2017) has shown an increased AIS contribution to sea level and a more significant retreat of the grounding line. We consider SGL parameterisation as the reference parameterisation for most of the simulations and discuss the impact of TGL parameterisation in Sect. 3.8.

We applied the TGL parameterisation under the weakly nonlinear sliding law. There is no consensus on the compatibility
between the TGL parameterisation and the Weertman sliding law, as the TGL parameterisation was derived from the Coulomb friction law near the grounding line (Tsai et al., 2015). Yet, the Coulomb friction law is applicable in a narrow transition region

in the vicinity of the grounding line not resolved on our coarse mesh, which lends validity to the combination of the TGL parameterisation and Weertman's sliding law.

### 2.2.4 Calving

Ice loss due to ice calving at the edges of ice shelves is responsible for almost half of the present-day ice mass loss of the Antarctic ice sheet (Depoorter et al., 2013; Rignot et al., 2013). Iceberg calving can have strong feedback effects as it affects ice-shelf buttressing (Fürst et al., 2016) and therefore ice flux at the grounding line and the stability of marine ice sheets (Schoof et al., 2017). It can also lead to a total disintegration of ice shelves followed by a potential marine ice-cliff instability (Pollard et al., 2015).

The nominal calving rate $C_r$ (in m yr$^{-1}$) in the f.ETISh model follows the following parameterisation (Pollard and DeConto, 2012a; Pattyn, 2017)

$$C_r = 30\,(1-w_c) + 3 \times 10^5 \max\,(\mathrm{div}\,\boldsymbol{v},0)\,\frac{w_c h_e}{\Delta x}, \tag{3}$$

where $\boldsymbol{v}$ is the vertical mean of the horizontal velocity, $h_e$ is the sub-grid ice thickness within a fraction of the ice-edge grid cell that is occupied by ice (Pollard and DeConto, 2012a), $\Delta x$ is the spatial resolution and $w_c = \min\,(1, h_e/200)$ is a weight factor.

We introduce uncertainty in sub-shelf melting by controlling the magnitude of the calving rate with a scalar multiplier factor $F_{\mathrm{calv}}$. This approach is similar to Briggs et al. (2013) and Pollard et al. (2016). Here, we consider for $F_{\mathrm{calv}}$ a nominal value of 1.0 and an uncertainty range from 0.5 to 1.5, that is, we consider the calving rate to vary between 50 % and 150 % of the nominal calving rate $C_r$.

### 2.2.5 Oceanic forcing

Sub-shelf melting is mainly controlled by sub-shelf ocean circulation, which can be affected by atmospheric changes. Ice-shelf thinning caused by increased sub-shelf melting leads to a reduction in ice-shelf buttressing. West Antarctica, where the bedrock lies mainly below sea level, is particularly vulnerable as suggested by observational (Rignot et al., 2014) and modelling (Favier et al., 2014; Joughin et al., 2014) studies.

High melt rates at the base of ice shelves result from the inflow of relatively warm Circumpolar Deep Water in ice-shelf cavities (Hellmer et al., 2012; Schmidtko et al., 2014). Changes in ocean circulation resulting in stronger sub-ice-shelf circulation are expected to increase basal melt rates at the base of ice shelves (Jacobs et al., 2011; Hellmer et al., 2017). Increase in atmospheric temperature leading to the presence of warmer deep water on the continental shelf is expected to strengthen sub-ice-shelf circulation, thus leading to an increase in sub-shelf melting. Yet, the link between global climate change and changes in sub-shelf melting is not always clear: it has been suggested that future climate change could lead to an increase in sub-shelf melting (Hellmer et al., 2012; Timmermann and Hellmer, 2013), a positive meltwater feedback that enhances sub-ice-shelf circulation and can trigger a climate tipping point (Hellmer et al., 2017) and even to a decrease in sub-shelf melting (Dinniman et al., 2012). Golledge et al. (2019) have also highlighted the need to couple ice-sheet models to climate models as

the increased discharge of freshwater from the Antarctic ice sheet could trap warm waters of the Southern Ocean below the sea surface.

Here, we capture the basic overturning circulation in ice-shelf cavities with the PICO box model. In the PICO model, the strength of the overturning flux is represented by a single parameter that depends on the density difference, or equivalently on both the salinity and temperature differences, between the incoming water masses on the continental shelf and the water masses near the deep grounding line of the ice shelf. An increase in the ocean temperature on the continental shelf leads to a stronger overturning flux and higher melt rates at the base of ice shelves. The ocean temperature on the continental shelf is determined from the present-day ocean temperature $T_{\mathrm{oc}}^{\mathrm{obs}}$ and the change in background atmospheric temperature $\Delta T$ via the linear relationship in Eq. (1).

We introduce uncertainty in sub-shelf melting by controlling the strength of the overturning flux through uncertainty in the ocean temperature on the continental shelf. Hence, we consider the ocean melt factor $F_{\mathrm{melt}}$ as an uncertain parameter with a nominal value of $0.3$ (Maris et al., 2014; Golledge et al., 2015) and an uncertainty range from 0.1 to 0.8.

### 2.2.6 Ice-shelf rheology

Ice rheology in large-scale ice-sheet models is usually described as an isotropic material obeying Glen's flow law (Greve and Blatter, 2009). However, ice is known to be an anisotropic material whose fabric is dependent on the temperature field, the strain-rate history and the stress history (Ma et al., 2010; Calonne et al., 2017). For a given fabric, the anisotropic response depends on the stress regime, which explains the ice stiffening when moving from a shear-dominated stress regime for grounded ice to an extension-dominated stress regime for floating ice.

We introduce an ice-shelf tune parameter $E_{\mathrm{shelf}}$ that accounts for anisotropy between grounded and floating ice. A lower value makes ice shelves more viscous (Briggs et al., 2013; Maris et al., 2014). We consider for $E_{\mathrm{shelf}}$ a nominal value of $0.5$ and an uncertainty range from 0.2 to 1, where a value of $0.5$ means that the ice in the ice shelves is two times more viscous than without shelf tuning.

### 2.2.7 Bedrock relaxation

Bedrock relaxation due to deglaciation may induce a negative feedback that promotes stability in marine portions and mitigates the effect of a marine ice-sheet instability (Gomez et al., 2010, 2013; Adhikari et al., 2014). The amplitude of the glacial isostatic uplift is determined by the flexural rigidity of the lithosphere and the viscous relaxation time of the asthenosphere. Recent studies (Van der Wal et al., 2015; Chen et al., 2018) have shown significant differences in the properties of the lithosphere and the asthenosphere between West and East Antarctica. Van der Wal et al. (2015) have found a lower viscosity and therefore a lower relaxation time for Earth's mantle underneath West Antarctica, making the glacial isostatic uplift in this region more sensitive to changes in ice thickness.

We account for the differences between East and West Antarctica by introducing two characteristic relaxation times $\tau_{\mathrm{e}}$ and $\tau_{\mathrm{w}}$ that we consider to both have a nominal value of 3000 years. We consider that $\tau_{\mathrm{e}}$ has an uncertainty range from 2500 to 5000 years and $\tau_{\mathrm{w}}$ has an uncertainty range from 1000 to 3500 years.

## 2.3 Uncertainty quantification methods

### 2.3.1 Characterisation of uncertainty

To quantify the impact of uncertainties on the AIS response, we adopt a probabilistic framework. Here, we assume, in the absence of any prior information other than the aforementioned nominal values and minimal and maximal values of the uncertainty ranges, that the parameters $F_{calv}$, $F_{melt}$, $E_{shelf}$, $\tau_e$ and $\tau_w$ are uniform independent random variables with bounds given by the minimal and maximal values of their uncertainty ranges. We explore the uncertainty in the sliding exponent $m$ by considering its nominal and extremal values separately, thus allowing to reduce the dimension of the parameter space while being consistent with other studies (Ritz et al., 2015). We explore the uncertainty in the atmospheric forcing $\Delta T$ by considering the 4 RCP scenarios. We limit the probabilistic characterisation to assuming uniform probability density functions and we do not address how this probabilistic characterisation could be refined by using expert assessment, data and statistical methods such as Bayesian inference. Yet, we provide results that give some insight into the impact of the choice of this probabilistic characterisation later in Sect. 3.6. We refer the reader to Petra et al. (2014), Isaac et al. (2015), Ruckert et al. (2017), Gopalan et al. (2018) and Conrad et al. (2018) for applications of Bayesian inference in glaciology and to Aschwanden et al. (2016) and Gillet-Chaulet et al. (2016) for examples of a calibration of the sliding exponent based on a comparison between simulated and observed surface velocities that can be used to prescribe a probabilistic characterisation of the sliding exponent.

### 2.3.2 Propagation of uncertainty

Given the probabilistic characterisation of the uncertainty in the parameters, the propagation of uncertainty serves to assess the impact of the uncertainty on the global mean sea-level change. In particular, its intent is to estimate the probability density functions for the change in GMSL as well as some of its statistical descriptors such as its mean, variance and quantiles. Various methods have been developed in UQ to estimate these statistical descriptors in a nonintrusive manner treating the ice-sheet model as a black box. Here, we use emulation methods based on a polynomial chaos (PC) expansion.

An emulator, also known as a surrogate model, is a computational model that mimics the ice-sheet model at low computational cost. Although emulators can also be obtained by Gaussian process regression (Rasmussen and Williams, 2006), we use polynomial chaos (PC) expansions (Ghanem and Spanos, 2003; Le Maître and Knio, 2010), which involve approximating the parameters-to-projection relationship as a polynomial in the parameters. We write this polynomial as a linear combination of polynomial basis functions and use least-squares regression (Appendix A) to evaluate the coefficients from a limited number of ice-sheet model runs at an ensemble of training points in the parameter space. The training points must be adequately chosen in the parameter space (Hadigol and Doostan, 2018) and the convergence of the PC expansion must be assessed to ensure accuracy. PC expansion may suffer from limitations: PC expansions require the parameters-to-projection relationship to be sufficiently smooth (no discontinuity or highly nonlinear behaviour) to allow an efficient approximation as a low-degree polynomial and PC expansions may be inefficient in high-dimensional problems.

The emulator of the relationship between the parameters and the projection is then used as a substitute for the ice-sheet model in a Monte Carlo method (Robert and Casella, 2013), in which samples of the parameters are drawn randomly from their

probability density function and mapped through the emulator into corresponding samples of the projections. Approximations to the probability density function of the projection and its statistical descriptors are then obtained from these samples of the projections by using statistical estimation methods: for instance, the mean and quantiles are approximated with the sample mean and quantiles.

The use of an emulator has the following advantages: (i) it provides an inexpensive approximation of the ice-sheet model that accelerates UQ; (ii) it provides an explicit view on the relationship between the parameters and the projection, highlighting potential linear or nonlinear dependences and interactions between the parameters; (iii) it allows to interpolate efficiently the projections in the parameter space; (iv) it can be used to carry out stochastic sensitivity analysis to assess the influence of each parameter on the projections; (v) under certain conditions, the same emulator can be reused between UQ analyses with

different probability density functions for the parameters and (vi) it can be used for Bayesian calibration (Ruckert et al., 2017).

We consider an ensemble of 20 distinct model configurations given by each combination of RCP scenario with a sliding law ($m = 1, 2, 3$ or $5$) and each combination of RCP scenario with the TGL parameterisation under the weakly nonlinear sliding law ($m = 2$). For each model configuration, we built a separate PC expansion to investigate the impact of uncertainty in the five parameters $F_{\text{calv}}$, $F_{\text{melt}}$, $E_{\text{shelf}}$, $\tau_{\text{e}}$ and $\tau_{\text{w}}$ on the uncertainty in $\Delta$GMSL. Here, we assume the continental response of

the Antarctic ice sheet, as measured through $\Delta$GMSL, to be sufficiently smooth to be represented with a PC expansion. For each model configuration, we generated an ensemble of 500 training points in the parameter space with a maximin Latin hypercube sampling design (Stein, 1987) and performed 500 forward simulations at these training points. In total, we carried out 10000 forward simulations of the f.ETISh model. For each model configuration and time instant of analysis, we fitted to the corresponding ensemble of 500 training points and forward simulations a polynomial chaos expansion of degree 3, which

we then used as emulator to evaluate the statistical descriptors either directly from the coefficients of the PC expansion or by running the emulator with an ensemble of $10^6$ independent and identically distributed samples from the parameter space. We present a convergence study and cross-validation for the PC expansion in Appendix A. Finally, the confidence regions for each model configuration are determined directly from the corresponding ensemble of 500 training points and forward simulations without the need for an emulator.

### 2.3.3   Stochastic sensitivity analysis

Stochastic sensitivity analysis serves to identify which uncertain parameters and their associated physical phenomenon are most influential in inducing uncertainty in the ice-sheet response. Here, we adopt the variance-based sensitivity indices (Saltelli et al., 2008), also called Sobol indices, described in more detail in Appendix B. Variance-based sensitivity indices rely on the decomposition of the variance of the projections as a sum of contributions from each uncertain parameter taken individually

and an interaction term. Then, the Sobol index of a given uncertain parameter represents the fraction of the variance of the projections explained as stemming from this sole uncertain parameter. A Sobol index takes values between 0 and 1, whereby a value of 1 indicates that the entire variance of the projections is explained by this sole uncertain parameter and a value of 0 indicates that the uncertain parameter has no impact on the projection uncertainty.

We compute the Sobol indices directly from the PC coefficients (Crestaux et al., 2009; Le Gratiet et al., 2017).

### 2.3.4 Confidence regions for grounded-ice retreat

To gain insight into the impact of the uncertainty in determining which regions of Antarctica are most at risk of ungrounding, we construct confidence regions for grounded ice for several probability levels. We define a confidence region for grounded ice for a given probability level as a region of Antarctica that remains covered everywhere with grounded ice with a probability of at least the given probability level under the uncertainty introduced in the ice-sheet model (see Appendix C for the mathematical definition). The differences between these confidence regions for grounded ice for different probability levels provide insight into the risk of ungrounding (see Sect. 3.5). Such confidence regions are useful because confidence regions with boundaries far from the initial grounding line may indicate an important MISI and large differences between confidence regions for different probability levels indicate a significant impact of the uncertainty on the ice-sheet ungrounding. We construct these confidence regions for grounded ice based on an extension of previous work by Bolin and Lindgren (2015) for Gaussian random fields to our glaciological context.

## 3 Results

We present nominal and probabilistic projections (relative to 2000 CE) for short-term (2100), medium-term (2300) and long-term (3000) time scales under different RCP scenarios and sliding laws.

### 3.1 Nominal projections

We first present nominal projections obtained using the nominal values of the parameters given in Table 1. We first present results under nominal conditions in order to assess subsequently the impact of uncertainties on AIS sea-level rise projections. Under nominal conditions, we find (Table 2) in RCP 2.6 an AIS contribution to sea level of 0.02 m by 2100, 0.07 m by 2300 and 0.20 m by 3000 and in RCP 8.5 an AIS contribution to sea level of 0.05 m by 2100, 0.59 m by 2300 and 3.90 m by 3000. In Fig. 3a, we represented the nominal projections as a function of time in all RCP scenarios. We find that the nominal AIS contribution to sea level is rather small in the first decades and starts to increase more significantly around 2100 with a rather constant growth rate. In Figs. 3b–e, we represented the nominal grounded ice region by 3000 for all RCP scenarios. We find that there is little ungrounding by 3000 in RCP 2.6 and RCP 4.5, while we observe a more significant ungrounding in Siple Coast and the Ronne basin in RCP 6.0 and a much more significant ungrounding in Siple Coast and the Amundsen Sea sector in RCP 8.5.

| Year | RCP 2.6 | RCP 4.5 | RCP 6.0 | RCP 8.5 |
|------|---------|---------|---------|---------|
| 2100 | 0.02 | 0.01 | 0.03 | 0.05 |
| 2300 | 0.07 | 0.10 | 0.22 | 0.59 |
| 3000 | 0.20 | 0.39 | 0.86 | 3.90 |

**Table 2.** Nominal projections (in metres) of AIS contribution to sea level on short-term (2100), medium-term (2300) and long-term (3000) time scales in different RCP scenarios.

## 3.2 Parameters-to-projections relationship

One of the advantages of a polynomial chaos expansion is that it provides an explicit approximation to the parameters-to-projections relationship, which can be visualised to gain insight into the relationship between the parameters and the projections.

In Figs. 4 and 5, we used the PC expansions to visualise how the projections depend on each parameter individually (one-at-a-time) while keeping the other parameters fixed at their nominal value under the weakly nonlinear sliding law in RCP 2.6 and RCP 8.5, respectively.

Figures 4a–c show that in RCP 2.6 $\Delta$GMSL increases with an increase in the calving factor nonlinearly. The slope is steeper for small values of this parameter, thus suggesting that $\Delta$GMSL is more sensitive to small changes about small values than about higher values. Figures 4d–f indicate that $\Delta$GMSL increases rather linearly with an increase in the melt factor. Figures 4g–i indicate a rather quadratic dependence on the shelf anisotropy factor for short-term and medium-term projections, with small and large values of this factor leading to more significant ice loss than the nominal value. This quadratic dependence can be explained by the influence of $E_{\text{shelf}}$ on two competing processes: a higher value of $E_{\text{shelf}}$ softens the ice, thus leading to faster ice flow in the ice shelves; but a higher value of $E_{\text{shelf}}$ also leads to ice-shelf thinning, thus reducing ice flux at the grounding line. In addition, we find that $\Delta$GMSL depends only little on the bedrock relaxation times (Figs. 4j–o). In fact, lower bedrock relaxation times do tend to stabilise the ice sheet and lower sea-level rise but this impact is weak compared to the influence of the other parameters. This result may be explained by our orders of magnitude of the bedrock relaxation times being of the order of a few millennia, thus preventing any significant uplift in the next few centuries. Yet, a recent study by Barletta et al. (2018) has suggested a bedrock relaxation time scale of the order of decades to a century in the Amundsen Sea sector, thus making glacial isostatic adjustment significant in the next decades and centuries in this region. To assess the influence of shorter relaxation times, we performed additional numerical experiments, albeit not reported in this paper, with a relaxation time for the whole West Antarctic ice sheet that varies widely from a few decades to a few millennia. For this range of values, we found that bedrock relaxation has a more significant influence on the AIS response, with a contribution to the uncertainty in $\Delta$GMSL that can reach 10 % in RCP 2.6 when $\tau_{\text{w}}$ is allowed to vary widely between 50 years and 3500 years.

We find in RCP 8.5 similar trends. However, whereas $F_{\text{calv}}$, $F_{\text{melt}}$ and $E_{\text{shelf}}$ influence $\Delta$GMSL equally in RCP 2.6, the melt factor influences $\Delta$GMSL most significantly in RCP 8.5. Whereas Figs. 5d–e show that $\Delta$GMSL increases rather linearly with

an increase in the melt factor, Fig. 5f shows that $\Delta$GMSL rather levels off at a plateau for large values of $F_{\mathrm{melt}}$ for long-term projections in RCP 8.5.

Additionally, Fig. S2 shows the emulators for several pairs of parameters with the other parameters fixed at their nominal values in RCP 2.6 and RCP 8.5. These figures show essentially the same trends as those identified in Figs. 4 and 5, in addition to interaction effects between the parameters. In particular, we find that smaller values of the calving and melt factors lead to mass gain (Fig. S2c), while larger values of the shelf anisotropy and melt factors lead to an important mass loss (Fig. S2d).

## 3.3 Sea-level rise projections

Under parametric uncertainties, we find (Table 3, Fig. 6) in RCP 2.6 a median AIS contribution to sea level that ranges from 0.02 to 0.03 m by 2100, from 0.07 to 0.13 m by 2300 and from 0.18 to 0.30 m by 3000 for the three sliding laws, as compared with the nominal projection of 0.02 m by 2100, 0.07 m by 2300 and 0.20 m by 3000; and we find in RCP 8.5 a median AIS contribution to sea level that ranges from 0.09 to 0.11 m by 2100, from 0.78 to 1.15 m by 2300 and from 3.18–6.12 m by 3000, as compared with the nominal projection of 0.05 m by 2100, 0.59 m by 2300 and 3.90 m by 3000. The median AIS sea-level rise projections are higher than the nominal projections, except for certain cases under the viscous sliding law. The nominal projections are not equal to the median projections because the ice-sheet model exhibits nonlinearities, as illustrated in Figs. 4 and 5, and the probability density functions for the parameters are not symmetric about their nominal value. As in the nominal projections, we find that in all RCP scenarios and under all sliding laws, the median AIS contribution to sea level is rather small in the first decades and starts to increase around 2100. As in the nominal projections, the median AIS contribution to sea level shows a rather constant growth rate in RCP 2.6, RCP 4.5 and RCP 6.0; contrary to the nominal projection, the median AIS contribution to sea level exhibits initially an acceleration before subsequently exhibiting a deceleration in RCP 8.5 under both nonlinear sliding laws (Fig. 6). This behaviour in the median projections in RCP 8.5 is a consequence of the initial rapid collapse of the West Antarctic ice sheet, which contributes by 3–3.5 metres to sea level, followed by a slower retreat in the East Antarctic ice sheet. By contrast, the nominal projections, which involve a melt factor of 0.3, indicate that the West Antarctic ice sheet does not completely collapse by the year 3000.

In RCP 2.6, we find (Table 3) 5–95 % probability intervals for sea-level rise projections that range from -0.06 to 0.10 m by 2100, from -0.14 to 0.31 m by 2300 and from -0.36 to 0.91 m by 3000. In RCP 8.5, these probability intervals range from -0.03 to 0.23 m by 2100, 0.17 to 2.01 m by 2300 and 0.82 to 7.68 m by 3000. The nominal projections are inside the 5–95 % probability intervals. We see that the 5–95 % probability intervals indicate an increase in sea level, though a decrease cannot be ruled out for the viscous sliding law and cooler atmospheric conditions. Figure 6 highlights that the 33–66 % probability intervals become wider under warmer atmospheric conditions and, to a lesser extent, more nonlinear sliding conditions. The uncertainty in the projections due to parametric uncertainty is rather significant, with possible overlaps between the 5–95 % probability intervals for different RCP scenarios. For instance, Fig. 4i shows that a contribution to sea level of about 0.7 m can be reached by 3000 in RCP 2.6 for the extreme value $E_{\mathrm{shelf}} = 1$, while Fig. 5f shows that this value is reached in RCP 8.5 for very limited sub-shelf melting ($F_{\mathrm{melt}}$ of about 0.12). This suggests that projections with similar contributions to sea level can arise in different RCP scenarios with different combinations of parameter values. Measuring the relative dispersion in $\Delta$GMSL

via the coefficients of variation, that is, the ratio between the standard deviation and the mean value, we find a coefficient of variation of 0.74 in RCP 2.6, 0.67 in RCP 4.5, 0.59 in RCP 6.0 and 0.33 in RCP 8.5 by 3000 under the weakly nonlinear sliding law. This demonstrates that the relative uncertainty in $\Delta$GSLM projections is higher in cooler RCP scenarios. This results from a much larger increase in the values of $\Delta$GSLM projections compared to the increase in their dispersion when the RCP scenario gets warmer.

Figure 7 shows the probability density functions for the change in GMSL at different time scales. The results display essentially unimodal probability density functions with wider tails for warmer scenarios and longer time scales. In RCP 2.6, the probability density functions resemble Gaussian probability density functions. In RCP 8.5 and at the short and medium time scales, the probability density functions are rather flat, which can be explained by the dependence of $\Delta$GMSL on $F_{\mathrm{melt}}$ being rather linear (Figs. 5d and e). In RCP 8.5 and at the long time scale, the probability density functions exhibit a more localised mode at higher values of $\Delta$GMSL, which can be explained by the collapse of the West Antarctic ice sheet and thus the presence of the plateau for higher values of the melt factor (Fig. 5f).

Figure 8 gives the probability of exceeding the threshold values of 0.5 m, 1.0 m and 1.5 m as a function of time. We find that the probability of exceeding 0.5 m by 2100 is negligible (probability of less than 1 %) in all RCP scenarios and under all sliding conditions. In RCP 2.6, the AIS contribution to sea level in the next centuries is strongly limited, with a probability of exceeding 0.5 m by 3000 reaching at most 30 %. In RCP 4.5 and RCP 6.0, we found nominal sea-level rise projections well below 1.5 m, while in RCP 8.5 this threshold is exceeded. In the presence of uncertainties, we find that the probability of exceeding 1.5 m of sea-level rise by 3000 can reach about 35 % in RCP 4.5, about 70 % in RCP 6.0 and about 95 % in RCP 8.5. The last result can be seen from Fig. 5, which indicates that $\Delta$GMSL is below 1.5 m only in a small region of the parameter space associated with small values of the melt factor. Furthermore, the shape of the exceedance curves in Fig. 8 provides some insight into the uncertainty in the time when a certain threshold value is exceeded. The time when $\Delta$GMSL exceeds 0.5 m with a probability of 33 % under the weakly nonlinear sliding law is 2415 in RCP 4.5, 2270 in RCP 6.0 and 2185 in RCP 8.5 while this value is exceeded with a probability of 66 % at 2790 in RCP 4.5, 2430 in RCP 6.0 and 2245 in RCP 8.5. We find that nominal projections overestimate the time when $\Delta$GMSL exceeds 0.5 m, with the exceedance time being beyond 3000 in RCP 4.5, around 2620 in RCP 6.0 and around 2280 in RCP 8.5.

| Year | Sliding (m) | RCP 2.6 | RCP 4.5 | RCP 6.0 | RCP 8.5 |
|------|-------------|---------|---------|---------|---------|
| 2100 | 1 | 0.02 (-0.06–0.10) | 0.04 (-0.05–0.12) | 0.05 (-0.04–0.13) | 0.09 (-0.03–0.20) |
|      | 2 | 0.03 (-0.01–0.09) | 0.05 (0.00–0.11) | 0.06 (0.00–0.14) | 0.11 (0.01–0.22) |
|      | 3 | 0.03 (-0.03–0.09) | 0.04 (-0.02–0.11) | 0.06 (-0.01–0.14) | 0.11 (0.00–0.23) |
| 2300 | 1 | 0.07 (-0.14–0.31) | 0.20 (-0.09–0.48) | 0.36 (-0.04–0.72) | 0.78 (0.17–1.35) |
|      | 2 | 0.13 (0.01–0.30) | 0.26 (0.03–0.54) | 0.45 (0.06–0.88) | 1.04 (0.27–1.81) |
|      | 3 | 0.09 (-0.09–0.29) | 0.28 (-0.03–0.58) | 0.50 (0.02–0.96) | 1.15 (0.31–2.01) |
| 3000 | 1 | 0.18 (-0.36–0.82) | 0.77 (-0.14–1.82) | 1.60 (-0.01–2.71) | 3.18 (0.82–4.25) |
|      | 2 | 0.30 (-0.01–0.82) | 0.93 (0.08–2.24) | 2.16 (0.15–4.06) | 5.08 (1.36–6.00) |
|      | 3 | 0.30 (-0.25–0.91) | 1.15 (0.08–2.56) | 2.50 (0.34–5.17) | 6.12 (1.65-7.68) |

**Table 3.** Probabilistic projections (in metres) of the AIS contribution to sea level on short-term (2100), medium-term (2300) and long-term (3000) time scales in different RCP scenarios and under different sliding conditions with Schoof's grounding-line parameterisation. ∆GMSL projections are the median projections with their 5–95 % probability intervals between parentheses.

### 3.4 Stochastic sensitivity analysis

Figure 9 provides the Sobol sensitivity indices for the change in GMSL on short-term, medium-term and long-term time scales in different RCP scenarios and under different sliding laws. We find that in RCP 2.6, the largest contribution to the uncertainty in ∆GMSL stems from the uncertainty in the ice-shelf rheology (Sobol indices ranging from 40 % to 60 %) followed by the uncertainty in the calving rate (Sobol indices ranging from 20 % to 40 %) and sub-shelf melting (Sobol indices ranging from 5 % to 25 %). Indeed, in RCP 2.6, sub-shelf melting plays only a limited role because ocean conditions remain essentially unchanged. Therefore, in RCP 2.6, the dispersion in ∆GMSL is mainly controlled by the ice-shelf rheology, which controls ice flow and buttressing in ice shelves, as well as calving, which reduces the extent of ice shelves and their buttressing.

By contrast, in warmer RCP scenarios and for longer time scales, the dominant source of uncertainty becomes the uncertainty in sub-shelf melting, which accounts in RCP 8.5 for more than 90 % of the uncertainty in sea-level rise projections. As shown in Fig. 5, in RCP 8.5 at 3000 ∆GMSL varies by several metres over the range of values of $F_{\mathrm{melt}}$, while ∆GMSL varies only by a few tens of centimetres over the range of values of the other parameters. Hence, the dominant influence of the uncertainty in the melting factor is also a consequence of the rather wide uncertainty range that we chose for this parameter.

Finally, we find that, in all RCP scenarios and under all sliding laws, the uncertainty in the bedrock relaxation times for West and East Antarctica has a limited impact (Sobol index smaller than 1 %), which is a direct consequence of the very limited dependence of the projections on the bedrock relaxation times. Moreover, the interactions between the parameters have a negligible impact as the sums of the individual Sobol indices account almost entirely for the variances of the projections.

## 3.5 Projections of grounded-ice retreat

Figure 10 provides insight into the regions of Antarctica that are most at risk of ungrounding in different RCP scenarios and at different time scales under the weakly nonlinear sliding law. Figure 10 was obtained as follows. First, we determined the 100 % confidence region for grounded ice, that is, the region of Antarctica where ice is certain to remain grounded and we coloured it in grey. Thus, there is no risk that the grounding line retreats to within the grey region. Then, we determined the 95 % confidence region for grounded ice, that is, the region of Antarctica that remains covered everywhere with grounded ice with a probability of more than 95 %, and we coloured the portion of the 95 % confidence region that extends beyond the 100 % confidence region in dark blue. Thus, there is a low risk of (less than) 5% that the grounding line retreats to within the dark blue region. We continued this procedure for decreasing values of the confidence level and using different colours as indicated in the legend in Fig. 10.

We find that ice remains grounded in regions above sea level. By contrast, in all RCP scenarios, the risk of ungrounding is highest in marine sectors of West Antarctica with fast-flowing ice streams, especially in Siple Coast, in the Ronne basin notably the Ellsworth land and in the Amundsen Sea sector. In warm RCP scenarios and at longer time scales, we also observe a risk of grounding-line retreat in the Wilkes marine basin in East Antarctica, where the grounding line could retreat between 100 km (with a risk of 95 %) and 500 km (with a risk of 5 %) from its present-day position. The risk of retreat in Wilkes basin may partially explain the acceleration in sea-level rise that we observed in Fig. 6 in RCP 8.5. The risk of grounding-line retreat in the Antarctic Peninsula is very limited due to the bedrock topography being above sea level, the marine glaciers being small and a high increase in precipitation in this region.

In RCP 2.6, we observe that the grounding line is quite stable over the next millennium, with the 100 % confidence region for grounded ice being almost unchanged from the present-day grounded ice region (the 100 % confidence region for grounded ice by 3000 only differ from the present-day grounded ice region by a few tens of kilometres). In RCP 4.5, ice remains grounded in most of the West Antarctic ice sheet over the next centuries, but our results also suggest a risk of retreat of the grounding line in some sectors of West Antarctica on longer time scales. In RCP 6.0, we find that the West Antarctic ice sheet belongs by 3000 to the 66 % confidence region for grounded ice, while in RCP 8.5, it belongs by 3000 only to the 5 % confidence region. This suggests a risk of 33 % that a major collapse of the West Antarctic ice sheet might occur in RCP 6.0 by 3000 and a risk of 95 % that a major collapse of the West Antarctic ice sheet might occur in RCP 8.5 by 3000.

As compared with the nominal projections of a limited retreat of the grounding line in West Antarctica in RCP 6.0 by 3000, we find that the impact of the parametric uncertainty is that a complete collapse of the West Antarctic ice sheet may occur with a risk of 33 % in RCP 6.0 by 3000. Moreover, Fig. 3e suggests that a complete disintegration of the West Antarctic ice sheet is underway by 3000 in RCP 8.5, while in Fig. 10l the West Antarctic ice sheet has already collapsed by 3000.

Additionally, we compared the projections under the weakly nonlinear sliding law with projections under the other sliding laws, that is the viscous sliding law (Fig. S7) and the strongly nonlinear sliding law (Fig. S8). We find a lower risk of retreat of the grounding line under the viscous sliding law with a slower disintegration of the West Antarctic ice sheet compared to the other sliding laws, especially in the drainage basins of Thwaites and Pine Island glaciers, which belong to the 50 % confidence

region for grounded ice by 3000 in RCP 8.5, while they belong to the 5 % confidence region by 3000 in RCP 8.5 for the other sliding laws. The strongly nonlinear sliding law seems to favour a faster and deeper retreat of the grounding line especially in the marine sectors of East Antarctica and the drainage basins of Thwaites and Pine Island glaciers. However, Fig. 10i and Fig. S8i suggest that ungrounding may be less significant in Siple Coast under the strongly nonlinear sliding law than the weakly nonlinear sliding law. Actually, MISI may occur when driving stresses overcome resistive stresses (Waibel et al., 2018). Driving stresses are primarily determined by the surface slopes, while resistive stresses depend on the basal sliding coefficient and the ice velocity through the sliding exponent. Driving stresses at the grounding line are higher in the Amundsen Sea sector than in Siple Coast due to steeper surface slopes, leading to a greater sensitivity of the grounding line in the former region to nonlinearity and a more plastic response.

## 3.6 Influence of the parameter probability density function

So far, we have represented the uncertain parameters $F_{\mathrm{calv}}$, $F_{\mathrm{melt}}$, $E_{\mathrm{shelf}}$, $\tau_{\mathrm{e}}$ and $\tau_{\mathrm{w}}$ by a uniform distribution on a fixed support. For instance, the uncertain parameter $F_{\mathrm{calv}}$ is represented by a uniform distribution with support $[F_{\mathrm{calv,min}}, F_{\mathrm{calv,max}}]$, where $F_{\mathrm{calv,min}}$ and $F_{\mathrm{calv,max}}$ are the minimum and maximum values in Table 1. We now address the influence of the probabilistic characterisation of the parametric uncertainty on our probabilistic projections by controlling the size of these supports. Hence, we represent the uncertain parameters $F_{\mathrm{calv}}$, $F_{\mathrm{melt}}$, $E_{\mathrm{shelf}}$, $\tau_{\mathrm{e}}$ and $\tau_{\mathrm{w}}$ by a family of uniform probability density functions indexed by a unique scaling factor $\alpha \in [0,1]$ that controls the supports of the uniform probability density functions. For instance, the uncertain parameter $F_{\mathrm{calv}}$ is represented for a given value of $\alpha$ by a uniform distribution with support $[F_{\mathrm{calv,nom}} + \alpha(F_{\mathrm{calv,min}} - F_{\mathrm{calv,nom}}), F_{\mathrm{calv,nom}} + \alpha(F_{\mathrm{calv,max}} - F_{\mathrm{calv,nom}})]$, where $F_{\mathrm{calv,nom}}$, $F_{\mathrm{calv,min}}$ and $F_{\mathrm{calv,max}}$ are the nominal, minimum and maximum values in Table 1. For the other parameters, the supports are defined similarly. The value $\alpha = 0$ represents the ice-sheet model without parametric uncertainty, that is, the ice-sheet model for the nominal values of the parameters, and the value $\alpha = 1$ represents the full uncertainty ranges considered so far. We propagated the uncertainty from the parameters to the sea-level rise projections for different values of the scaling factor reusing the emulator that we had built for $\alpha = 1$ over the whole parameter space. Hence, the projections to follow for $\alpha = 0$ based on this emulator are not exactly equal to the nominal projections determined directly from the ice-sheet model.

For the long-term projections, Figs. 11a–d show the median and the 33–66 % and 5–95 % probability intervals as a function of the scaling factor in the different RCP scenarios under the weakly nonlinear sliding law and Figs. 11e–h show the probability density functions for the values $\alpha = 0.2, 0.6$ and $1.0$. We find that the median projections increase with an increase in the scaling factor and range in RCP 2.6 from 0.16 m to 0.30 m, in RCP 4.5 from 0.37 m to 0.93 m, in RCP 6.0 from 1.11 m to 2.16 m and in RCP 8.5 from 3.80 m to 5.08 m. In addition, the width of the probability intervals increases with increasing uncertainty in the parameters. While the width of the 33–66 % probability interval increases rather linearly with the scaling factor and is rather symmetric about the median, the width of the 5–95 % probability interval increases more nonlinearly with an increase in the scaling factor, as illustrated in Fig. 11a in RCP 2.6 and Fig. 11d in RCP 8.5. The probability density functions attribute higher weight to larger values of $\Delta$GMSL under increased parametric uncertainty and increased weight given to larger values of in particular $E_{\mathrm{shelf}}$ in RCP 2.6 (Fig. 4i) and $F_{\mathrm{melt}}$ in RCP 8.5 (Fig. 5f). Figures 11a–d show that the sensitivity of the amount of

uncertainty in the projections (probability intervals) to the amount of uncertainty in the parameters (scaling factor) is higher for warmer scenarios, with an upper bound between RCP 6.0 and RCP 8.5 as a consequence of the collapse of the West Antarctic ice sheet.

### 3.7 Projections under a more plastic sliding law

We ran the same ensemble of simulations under a more plastic sliding law ($m = 5$). Table 4 and Fig. 12 give for the AIS contribution to sea level the median and the 33–66 % and 5–95 % probability intervals. As compared with less plastic sliding laws ($m = 1, 2, 3$), we find an increase in sea-level rise projections on short-term and medium-term time scales, thus suggesting a more significant and faster response to perturbations under more plastic sliding conditions. On a long-term time scale, the ice-sheet mass loss can be less important under $m = 5$ than $m = 3$, as observed in the median projections by 3000 in RCP 4.5

and RCP 8.5.

We find that overall the AIS contribution to sea level is an increasing function of the sliding exponent, with the differences between successive exponents getting smaller as $m$ increases, as already pointed out by Gillet-Chaulet et al. (2016); for instance, we observe a greater difference in the projections between $m = 1$ and $m = 2$ than between $m = 3$ and $m = 5$. On longer time scales, tipping points and nonlinearities associated with MISI may trigger a slightly different response of the ice sheet

depending on the initial conditions, which could explain the smaller ice loss in our results under $m = 5$ than $m = 3$.

| Year | RCP 2.6 | RCP 4.5 | RCP 6.0 | RCP 8.5 |
|---|---|---|---|---|
| 2100 | 0.03 (-0.02–0.08) | 0.05 (-0.01–0.12) | 0.07 (-0.01–0.15) | 0.13 (0.01–0.25) |
| 2300 | 0.14 (-0.02–0.32) | 0.34 (0.04–0.61) | 0.56 (0.10–0.99) | 1.24 (0.40–2.19) |
| 3000 | 0.37 (-0.07–0.92) | 1.07 (0.17–2.56) | 2.70 (0.41–5.16) | 5.74 (1.92–7.03) |

**Table 4.** Same as Table 3 but with sliding exponent $m = 5$. $\Delta$GMSL projections are the median projections with their 5–95 % probability intervals between parentheses.

### 3.8 TGL parameterisation

We ran the same ensemble of simulations under the weakly nonlinear sliding law using, this time, the TGL parameterisation instead of the SGL parameterisation. Under the TGL parameterisation, Table 5 and Fig. 13 give for the AIS contribution to sea level the median and the 33–66 % and 5–95 % probability intervals. We find an overall increase in sea-level rise projections

compared to our results under the SGL parameterisation. This result was expected as the TGL parameterisation has been shown to increase grounding-line sensitivity to environmental changes (Tsai et al., 2015; Pattyn, 2017). The probability of exceeding 0.5 m by 2100 is still negligible (probability of less than 1 %) in all RCP scenarios. However, the probability of exceeding 0.5 m and 1 m by 3000 in RCP 2.6 can reach more than 40 % and 10 %, respectively. For other scenarios, the retreat of the Antarctic ice sheet is much more pronounced and faster than under the SGL parameterisation and the probability to exceed

1.5 m of sea-level rise by 3000 can reach more than 50 % in RCP 4.5, 80 % in RCP 6.0 and 99 % in RCP 8.5.

Figures 10m–o show the confidence regions for grounded ice under the TGL parameterisation in RCP 8.5. As also pointed out by Pattyn (2017), the TGL parameterisation leads to a faster and more significant grounding-line retreat in the marine sectors and an additional mass loss from East Antarctica, especially in the Aurora basin.

| Year | RCP 2.6 | RCP 4.5 | RCP 6.0 | RCP 8.5 |
|------|---------|---------|---------|---------|
| 2100 | 0.06 (0.00–0.14) | 0.10 (0.02–0.19) | 0.12 (0.02–0.23) | 0.21 (0.05–0.37) |
| 2300 | 0.20 (0.02–0.47) | 0.47 (0.10–0.89) | 0.83 (0.18–1.43) | 1.85 (0.59–3.12) |
| 3000 | 0.46 (0.00–1.30) | 1.73 (0.22–2.91) | 3.47 (0.76–5.37) | 7.43 (3.85–10.79) |

**Table 5.** Same as Table 3 but with the TGL parameterisation. ΔGMSL projections are the median projections with their 5–95 % probability intervals between parentheses.

## 4 Discussion

### 4.1 Comparison of the sea-level rise projections with previous work

Regarding the short-term AIS contribution to sea level, we projected in the RCP 2.6 scenario a median of 0.02–0.03 m under the SGL parameterisation and 5–95 % probability intervals from -0.06 m to 0.10 m. These projections are similar to other estimates based on other mechanisms. Golledge et al. (2015) reported an AIS contribution to sea level between -0.01 m and 0.10 m with the lower and higher bounds corresponding to the absence and presence of sub-grid interpolation of basal melting at the grounding line. DeConto and Pollard (2016) reported an AIS contribution to sea level between -0.11 m and 0.15 m based on a model calibration with a range of Pliocene sea-level targets between 5 and 15 m higher than today. In the same scenario, we found an increased AIS response under the TGL parameterisation with a median AIS contribution to sea level of 0.06 m and a 5–95 % probability interval between 0.00 m and 0.14 m by 2100. These higher projections are similar to the higher projections (0–0.22 m) reported by DeConto and Pollard (2016) for a higher range of Pliocene sea-level targets between 10 and 20 m. In all RCP scenarios, under all sliding laws and under both grounding-line parameterisations, our results suggested that the AIS contribution to sea level does not exceed 0.5 m by 2100 with a probability of at least 99 %. These results are in agreement with Ritz et al. (2015), who determined an AIS contribution to sea level that lies between 0.05 m and 0.30 m, and Golledge et al. (2015), who found an AIS contribution to sea level that reaches at most 0.38 m in RCP 8.5 with sub-grid interpolation of basal melting at the grounding line. Yet, projections by DeConto and Pollard (2016) and Schlegel et al. (2018), who applied the more sensitive Budd-type friction law (Brondex et al., 2017), can exceed 0.5 m and even 1 m, but these higher projections are under extreme and maybe unrealistic warming conditions.

Regarding the long-term AIS contribution to sea level, our projections under the SGL parameterisation are similar to other estimates by Golledge et al. (2015). In particular, both studies suggest that the AIS contribution to sea level by 3000 in RCP 2.6 is limited to less than 1 m with a probability estimated to be at least 95 % in our study, while an AIS contribution to sea level above 1.5 m by 3000 may arise in all other RCP scenarios. Yet, our long-term projections are generally below projections by

DeConto and Pollard (2016) with hydrofracturing and ice-cliff failure mechanisms especially under warmer RCP scenarios, but the discrepancies between the projections of both models are reduced under the TGL parameterisation. In RCP 2.6, our projections by 2500 under the TGL parameterisation range from 0.04 m to 0.73 m, which is similar to projections by DeConto and Pollard (2016), which range respectively from -0.23 m to 0.61 m and 0.02 m to 0.48 m for the lower and higher range of Pliocene of sea-level targets.

## 4.2 Comparison of the impact of parametric uncertainty with previous work

Similarly to Golledge et al. (2015, 2017), our study emphasised the pivotal role played by the emission scenario and sub-shelf melting as critical drivers of the future changes in the AIS mass balance on medium-term and long-term time scales through ice-shelf thinning and subsequent reduced buttressing. As in Ritz et al. (2015), we found that the AIS contribution to sea level is an increasing function of the sliding exponent, thus meaning that more plastic sliding conditions speed up the ice flow and consequently ice loss. Following Pattyn (2017), we highlighted the greater sensitivity of the grounding-line migration under the TGL parameterisation, thus stressing the key role played by physical processes in the vicinity of the grounding line.

## 4.3 Comparison of projections of grounded-ice retreat with previous work

Consistent with our results, Golledge et al. (2015), with a 10-km resolution model, projected that grounding-line retreat is most significant in the Siple Coast region. However, Ritz et al. (2015), based on the probability of MISI onset, as well as Cornford et al. (2015), with a sub-kilometre resolution around the grounding line, projected that grounding-line retreat is most significant in the Amundsen Sea sector. Schlegel et al. (2018) found that grounding-line retreat is most significant in the Amundsen Sea sector under generalised ocean warming experiments for the Antarctic ice sheet, but, after calibrating sub-shelf melt rates with bounds that vary region by region and are assigned values deduced from the literature and model sensitivity studies, they found that the western Ronne basin has the greater sensitivity. These discrepancies between our findings and those of Cornford et al. (2015), Ritz et al. (2015) and Schlegel et al. (2018) may be explained by our ocean model which may overestimate sub-shelf melting underneath Ross ice shelf and underestimate ocean circulation in the Amundsen Sea and by our initialisation method which may underestimate the basal sliding coefficients for Thwaites and Pine Island glaciers. Moreover, the lower sensitivity of the Amundsen Sea sector may arise in our simulations from shortcomings in the buttressing parameterisation, our low resolution not capturing properly the bedrock topography, the small pinning points and the flow dynamics in the narrower sectors of the ice sheet and our representation of calving, which may increase the instability threshold; see, for instance, Arthern and Williams (2017) and Waibel et al. (2018) for more thorough discussions about the instability threshold in the Amundsen Sea sector.

## 4.4 Projections of ice loss and grounding-line retreat under parametric uncertainty

The significance of the contribution of the Antarctic ice sheet to sea level under climate change is primarily controlled by the sensitivity, the response time and the vulnerability of its marine drainage basins, with the West Antarctic ice sheet more

sensitive and vulnerable than the East Antarctic ice sheet. The instability of marine drainage basins and their ability to trigger accelerated ice loss and significant grounding-line retreat is determined by bedrock topography and ice-shelf buttressing which depends on the importance of ice-shelf thinning. Our nominal projections showed that the AIS contribution to sea level by 3000 is rather limited (less than 1 metre) in RCP 2.6, RCP 4.5 and RCP 6.0, while an accelerated ice loss that leads to a contribution of several metres is triggered in RCP 8.5 (Fig. 3a). In addition, the nominal retreat of the grounding line by 3000 is rather limited in RCP 2.6, RCP 4.5 and RCP 6.0 (Figs. 3b–d), while a significant retreat of the grounding line is triggered in the Siple Coast, Ronne–Filchner and Amundsen Sea sectors in RCP 8.5 (Fig. 3e).

Our probabilistic results provide insight into the impact of parametric uncertainty on these projections. In RCP 2.6, the projections hold irrespectively of parametric uncertainty: the AIS contribution to sea level by 3000 has a 95 % quantile of up to 0.91 m (Table 3) and there is a limited risk that the grounding line retreats beyond our nominal projections (Fig. 3b and Fig. 10c). In RCP 4.5 and RCP 6.0, the projections are more sensitive to parametric uncertainty than in RCP 2.6: the AIS contribution to sea level by 3000 has a 95 % quantile of up to 2.56 m in RCP 4.5 and up to 5.17 m in RCP 6.0 (Table 3) and both scenarios entail a risk of triggering a more significant retreat of the grounding line beyond our nominal projections (Figs. 3c–d). This risk is present especially in the Amundsen Sea sector and it is less significant in RCP 4.5 than in RCP 6.0, in which a complete disintegration of the West Antarctic ice sheet may be triggered (Figs. 10f and i). Finally, in RCP 8.5, our probabilistic results suggest an accelerated ice loss and a significant retreat of the grounding line in West Antarctica, as in our nominal projections: the AIS contribution to sea level by 3000 has an uncertainty range with a 5 % quantile above 0.82 m and a 95 % quantile of up to 7.68 m (Table 3) and there is a high risk of triggering a complete disintegration of the West Antarctic ice sheet (Fig. 10l).

In conclusion, the projections hold irrespectively of parametric uncertainty in the strongly mitigated RCP 2.6 scenario: accommodating parametric uncertainty in the ice-sheet model leads to projections in agreement with the nominal projections of limited ice loss and limited grounding-line retreat in RCP 2.6. However, the projections are more sensitive to parametric uncertainty for intermediate scenarios such as RCP 4.5 and RCP 6.0: accommodating parametric uncertainty in the ice-sheet model leads to projections in disagreement with the nominal projections and indicates instead some risk of triggering accelerated ice loss and significant grounding-line retreat for intermediate scenarios such as RCP 4.5 and RCP 6.0. Finally, the warm RCP 8.5 scenario triggers the collapse of the West Antarctic ice sheet, almost irrespectively of parametric uncertainty.

### 4.5   Structural uncertainty and limitations

A first limitation of our study is associated with the modelling hypotheses inherent to our ice-sheet model. The f.ETISh model is an ice-sheet model that focuses on essential marine ice-sheet mechanisms, similarly to the ice-sheet model by Pollard and DeConto (2012a). Certain physical processes, especially small-scale processes, may be represented imperfectly, especially with the 20-km resolution adopted for our simulations. This may reduce the ability to simulate important ice streams such as Pine Island and Thwaites glaciers (only represented by a few grid points), whose stability is controlled by local bedrock features (Waibel et al., 2018). Hence, grounding-line migration and thresholds for instabilities may be not captured properly even with a parameterisation of the grounding-line flux, especially in low-forcing scenarios and short-term projections. As discussed in

Sect. 2.1, the flux-condition at the grounding line is also questionable for buttressed ice shelves (Pattyn et al., 2013; Reese et al., 2018a) found around Antarctica and short transients (Drouet et al., 2013; Pattyn et al., 2013; Pattyn and Durand, 2013). Yet, we think that using a 20-km resolution and a flux condition at the grounding-line remains an acceptable assumption in large-scale and long-term ice-sheet simulations and large-ensemble simulations. We expect discrepancies between our results and results

at a higher spatial resolution or with a higher level of complexity to be limited when compared to the uncertainty in the results due to the uncertainty in the model parameters and forcing. Besides, sub-shelf melting may not be captured properly despite the use of the PICO ocean-model coupler, especially in the Amundsen Sea sector and underneath Ross ice shelf (Timmermann et al., 2012; Depoorter et al., 2013; Rignot et al., 2013; Moholdt et al., 2014). A second limitation comes from the hypotheses relevant to our characterisation of uncertainties. We adopted rather large uncertainty ranges for the parameters. As discussed

in Sect. 3.6, projections can be strongly affected by extreme conditions. Moreover, we chose the bounds of the uncertainty ranges quite heuristically. A third limitation comes from the fairly simple way in which certain sources of uncertainty were introduced. We assumed a direct influence of the atmospheric forcing on sub-shelf melting through the ocean melt factor. However, as discussed in Sect. 2.2.5, atmospheric forcing affects the Circumpolar Deep Water circulation in ice-shelf cavities and subsequently modifies sub-shelf melting (Pritchard et al., 2012). Still, the fate of the Southern Ocean and the evolution of

sub-shelf melting under global climate change remains unclear (Hellmer et al., 2012; Dinniman et al., 2012; Timmermann and Hellmer, 2013; Hellmer et al., 2017). Given the importance of sub-shelf melting in driving the future response of the Antarctic ice sheet, there is a clear need to better constrain future sub-shelf melt and incorporate it properly in ice-sheet models. We also introduced uncertainty in calving with a simple multiplier factor that does not take into account the stress regime in the ice shelves. A fourth limitation concerns the correction for the initial drift that adds a bias to the projections. We adopted this

correction to make our results insensitive to model initialisation but our corrected results are likely to underestimate the AIS response on a short-term time scale as our approach does not take into account any current transient changes in the Antarctic ice sheet. Another limitation is related to the construction of our emulator. To avoid overfitting the training data, we tuned the emulator to reproduce the overall trend in the parameters-to-projections relationship but not local variations in the parameter space that may stem from numerical noise and errors.

**5   Conclusions**

We studied the multi-centennial response of the Antarctic ice sheet under uncertainty using methods from the field of UQ. We investigated uncertainties in atmospheric forcing, basal sliding, grounding-line parameterisation, sub-shelf melting, calving, ice-shelf rheology and bedrock relaxation. We used emulation-based methods to represent the parameters-to-projection relationship, stochastic sensitivity analysis to assess the significance of each source of uncertainty in inducing uncertainty in the

projections and confidence regions for excursion sets to assess the risk of grounding-line retreat. We found that all investigated sources of uncertainty, except bedrock relaxation time, contribute to the uncertainty in the projections. We showed that the sensitivity of the projections to uncertainties increases and the contribution of the uncertainty in sub-shelf melting to the uncertainty in the projections becomes more and more dominant as atmospheric and oceanic temperatures rise, with a contribution

to the uncertainty in sea-level rise projections that goes from 5–25 % in RCP 2.6 to more than 90 % in RCP 8.5. We showed that the significance of the AIS contribution to sea level is controlled by MISI in marine basins, with the biggest contribution stemming from the more vulnerable West Antarctic ice sheet. We found that, irrespectively of parametric uncertainty, the strongly mitigated RCP 2.6 scenario prevents the collapse of the West Antarctic ice sheet, that in both RCP 4.5 and RCP 6.0 scenarios the occurrence of MISI in marine basins is more sensitive to parametric uncertainty and that, almost irrespectively of parametric uncertainty, RCP 8.5 triggers the collapse of the West Antarctic ice sheet.

*Data availability.* All datasets used in this paper are publicly available, including Bedmap2 (Fretwell et al., 2013), ocean temperature and salinity (Schmidtko et al., 2014) and geothermal heat flux (An et al., 2015). Results of the RACMO2 model were kindly provided by Melchior Van Wessen. Results from the f.ETISh model for this study are available on request from K. Bulthuis (kevin.bulthuis@uliege.be). The MATLAB code used to analyse the results is also available on request from the same author.

## Appendix A: Polynomial chaos expansion

In this appendix, we concisely provide further details about how we used PC expansions in our study; we refer the reader to, for instance, Ghanem et al. (2017) and Le Maître and Knio (2010) for more comprehensive treatments of the theory and various applications of PC expansions.

Let us represent the ice-sheet model as an abstract model $y = g(\boldsymbol{x})$ where $\boldsymbol{x} = (x_1, \ldots, x_d)$ is a vector of $d$ parameters, $y$ the model response and $g$ the response function. In our study, $d = 5$, $x_1 = F_{\mathrm{calv}}$, $x_2 = F_{\mathrm{melt}}$, $x_3 = E_{\mathrm{shelf}}$, $x_4 = \tau_{\mathrm{e}}$, $x_5 = \tau_{\mathrm{w}}$ and $y = \Delta\mathrm{GMSL}$ at a given time. In our study, the parameters are uncertain and have a probability density function that we denote by $p$.

A polynomial chaos expansion is an approximation of the response function $g$ with a polynomial $g^K$ as

$$g(\boldsymbol{x}) \approx g^K(\boldsymbol{x}) = \sum_{k=0}^{K} c_k \psi_k(\boldsymbol{x}), \tag{A1}$$

where the $\psi_k$ are a basis of predefined polynomials of increasing degree and orthonormal with respect to the probability density function of the parameters, by which we understand that

$$\int_{\mathbb{R}^d} \psi_k(\boldsymbol{x})\psi_l(\boldsymbol{x})p(\boldsymbol{x})\,d\boldsymbol{x} = \begin{cases} 0 \text{ if } k \neq l \\ 1 \text{ if } k = l \end{cases}, \tag{A2}$$

and $K + 1$ is the number of predefined polynomials in the expansion.

In order to fit the PC expansion in Eq. (A1) to the ice-sheet model, we calculate the coefficients using a (weighted) least-squares approach:

$$\boldsymbol{c} = \arg\min_{\boldsymbol{d} \in \mathbb{R}^{K+1}} \sum_{i=1}^{N} w^{(i)} \left( y^{(i)} - \sum_{k=0}^{K} d_k \psi_k(\boldsymbol{x}^{(i)}) \right)^2, \tag{A3}$$

where $\{x^{(i)}, 1 \leqslant i \leqslant N\}$ is a set of $N$ training points in the parameter space, $\{y^{(i)} = g(x^{(i)}), 1 \leqslant i \leqslant N\}$ is the set of model responses at the training points, $c = (c_0, c_1, \ldots, c_K)$ collects the PC coefficients and $\{w^{(i)}, 1 \leqslant i \leqslant N\}$ is the set of weights. We normalise the parameters to accommodate the different orders of magnitude of the parameters and reduce the potential ill-conditioning of the least-squares problem. We generate the set of training points with a maximin Latin hypercube sampling

design (Stein, 1987; Fajraoui et al., 2017; Hadigol and Doostan, 2018), which is a space-filling design that aims at maximising the smallest distance between neighbouring points, thus ensuring a proper coverage of the parameter space. We consider a PC expansion of degree 3, which corresponds to $K = 56$ for $d = 5$, as we strive to reproduce the overall trend of the response function without overfitting small local variations.

We solve Eq. (A3) by solving the normal equations

$$(\mathbf{G}^T \mathbf{W} \mathbf{G}) c = \mathbf{G}^T \mathbf{W} y, \tag{A4}$$

where $y = (y^{(1)}, \ldots, y^{(N)})$ collects the model responses at the training points, $\mathbf{G}$ is the measurement matrix whose entries are given by $\mathbf{G}_{ki} = \psi_k(x^{(i)})$ and $\mathbf{W}$ is a diagonal weight matrix whose diagonal entries are the weights ($\mathbf{W}_{ii} = w^{(i)}$). As we construct the training points by using a Latin hypercube sampling design, the training points $x^{(i)}$ have equal weights of $\mathbf{W}_{ii} = w^{(i)} = 1/N$. We do note that there exist other methods to determine the coefficients in Eq. (A1) including methods

involving deterministic quadrature rules (Le Maître and Knio, 2010). Here, one of our motivations for choosing a least-squares approach is its good ability to handle numerical noise in low-degree expansions (Iskandarani et al., 2016).

We estimate statistical descriptors of the uncertain model response with Monte Carlo simulation in which the PC expansion is used as a computationally efficient substitute for the ice-sheet model. For instance, we estimate the probability density function of the response through kernel density estimation (Scott, 2015), while we approximate the mean $\mu$ and the variance

$\sigma^2$ of the model response as

$$\mu \approx \mu^{K,\nu} = \frac{1}{\nu} \sum_{i=1}^{\nu} g^K(x^{(i)}), \tag{A5}$$

$$\sigma^2 \approx \left(\sigma^{K,\nu}\right)^2 = \frac{1}{\nu} \sum_{i=1}^{\nu} \left(g^K(x^{(i)}) - \mu^{K,\nu}\right)^2, \tag{A6}$$

where $\{x^{(i)}, 1 \leqslant i \leqslant \nu\}$ now denotes an ensemble of $\nu$ independent and identically distributed samples from the probability density function of the parameters. Because the PC expansion is based on orthonormal polynomials with respect to the proba-

bility density function of the uncertain parameters and of increasing degree with $\psi_0 = 1$, we can evaluate some of the statistical descriptors of the uncertain model response directly from the PC coefficients. For example, the mean $\mu$ and the variance $\sigma^2$ of the model response can also be approximated as follows

$$\mu \approx \mu^K = c_0, \tag{A7}$$

$$\sigma^2 \approx \left(\sigma^K\right)^2 = \sum_{k=1}^{K} c_k^2. \tag{A8}$$

The accuracy of the PC expansion has to be assessed with respect to the degree of the PC expansion and the number of training points. We validate the accuracy of the PC expansion using cross-validation and convergence tests. We generate a new set of samples in the parameter space and we compare the exact response of the ice-sheet model with the approximate response of the PC expansion. Figures A1a–c show results of such a cross-validation for a PC expansion of degree 3. These figures
suggest that the PC expansion represents the overall model response with sufficient accuracy. Lower accuracy is achieved near the boundaries of the parameter space. We also carry out convergence tests, as illustrated in Fig. A1d in RCP 8.5 and at time 3000. This figure represents the maximum absolute error and the mean-squared error between the exact response of the ice-sheet model and the approximate response of the PC expansion at the training points as a function of the number of training points. We see that for the 500 training points considered in this paper, reasonable convergence of the PC expansion is
achieved.

## Appendix B: Sobol sensitivity indices

In this appendix, we concisely provide further details about how we used stochastic sensitivity analysis; we refer the reader to, for instance, Ghanem et al. (2017) and Saltelli et al. (2008) for more comprehensive treatments of the theory and various applications.
15       We first assume that the uncertain parameters are statistically independent. Sobol indices are based on the decomposition of the response function $g$ in terms of the main effects associated with the parameters individually and an interaction effect associated with all parameters together:

$$y = g(\boldsymbol{x}) = g_0 + \sum_{i=1}^{d} g_i(x_i) + g_I(\boldsymbol{x}), \tag{B1}$$

where $g_0$ is constant, each $g_i$ the so-called main effect associated with the corresponding parameter $x_i$ and $g_I$ the so-called
interaction effect. The constant, the main effects and the interaction effect are given by

$$g_0 = \int_{\mathbb{R}^d} g(\boldsymbol{x}) p(\boldsymbol{x}) \, d\boldsymbol{x}, \tag{B2}$$

$$g_i = \int_{\mathbb{R}^{d-1}} g(\boldsymbol{x}) p_{-i}(\boldsymbol{x}_{-i}) \, d\boldsymbol{x}_{-i} - g_0, \ 1 \leqslant i \leqslant d, \tag{B3}$$

$$g_I = g - g_0 - \sum_{i=1}^{d} g_i, \tag{B4}$$

where $\boldsymbol{x}_{-i} = (x_1, \ldots, x_{i-1}, x_{i+1}, \ldots, x_d)$ denotes the subset of parameters including all the parameters except $x_i$ and $p_{-i}$
denotes the probability density function for this subset of parameters. The constant $g_0$ is the mean value of $g$. Using the calculus of variations, it can be shown that the main effect $g_i$ is such that the function $g_0 + g_i$ is the least-squares best approximation of $g$ among all functions that depend only on $x_i$. As a consequence of this least-squares best approximation property, the functions $g_0, g_1, \ldots, g_d, g_I$ are orthonormal with respect to the probability density function of the uncertain parameters.

As a consequence of the orthonormality of the functions $g_0, g_1, \ldots, g_d, g_I$, the variance $\sigma^2$ of the model response can be decomposed as

$$\sigma^2 = \left( \sum_{i=1}^{d} S_i + S_I \right) \sigma^2, \tag{B5}$$

with

$$S_i = \frac{1}{\sigma^2} \int_{\mathbb{R}} |g_i(x_i)|^2 p_i(x_i) \, dx_i, \ 1 \leqslant i \leqslant d, \tag{B6}$$

$$S_I = 1 - \sum_{i=1}^{d} S_i, \tag{B7}$$

where $p_i$ denotes the probability density function of $x_i$. Here, $S_i$, which takes a value between 0 and 1, is the Sobol index for the $i$-th parameter and $S_I$ is the interaction index. The Sobol index $S_i$ can be interpreted either as the relative contribution of the uncertainty in the sole $i$-th parameter to the variance of the model response or as the reduction in the variance of the model response that we may expect by learning the exact value of this parameter (Oakley and O'Hagan, 2004). Sobol indices allow to rank the uncertain parameters in terms of their contribution to the variance of the model response, thus indicating which parameters are most influential in inducing the uncertainty in the model response.

Here, we substituted the ice-sheet model by a PC expansion based on orthonormal polynomials and estimated the Sobol indices directly from the PC coefficients (Crestaux et al., 2009), that is,

$$S_i \approx S_i^K = \frac{1}{\left( \sigma^K \right)^2} \sum_{k \in A_i} c_k^2, \tag{B8}$$

where $A_i$ is the set of indices associated with the non-constant polynomials that only depend on $x_i$.

## Appendix C: Confidence regions for grounded ice

In this appendix, we concisely provide further details about how we defined and computed confidence regions for grounded ice. To distinguish between grounded ice and floating ice at a given time, the f.ETISh model (Pattyn, 2017) evaluates the so-called buoyancy imbalance

$$BI(\boldsymbol{x}) = \rho_w b(\boldsymbol{x}) + \rho_i h(\boldsymbol{x}), \tag{C1}$$

where $b$ is the bedrock elevation, $h$ the ice thickness, $\rho_w$ the water density and $\rho_i$ the ice density. The buoyancy imbalance is negative for floating ice, positive for grounded ice and null at the grounding line. Therefore, the grounded ice domain $D_g$ is

$$D_g = \{ \boldsymbol{x} : BI(\boldsymbol{x}) \geqslant 0 \}. \tag{C2}$$

In the presence of uncertainties in the model, we define an $\alpha$ % confidence region $D_g(\alpha)$ for the grounded ice domain as a region of Antarctica that has a probability of at least $\alpha$ to be included in the grounded ice domain, that is,

$$P \{ D_g(\alpha) \subseteq D_g \} \geqslant \alpha. \tag{C3}$$

We compute the confidence regions using an adaptation of a thresholding algorithm by Bolin and Lindgren (2015). We seek the confidence regions in a parametric family indexed by a threshold parameter and based on the marginal probability density functions and we determine the threshold parameter so as to achieve the required level of confidence. While Bolin and Lindgren (2015) consider Gaussian random fields, here we work with non-Gaussian random fields, which requires to evaluate the marginal probability density functions and the probability of inclusion with Monte Carlo simulation.

*Author contributions.* All authors discussed the results presented in this manuscript. KB conducted the design, execution and UQ analysis of the experiments with relevant inputs from MA for the UQ methodology and SS and FP for the physical interpretation of the results. The manuscript was written by KB and MA with relevant comments from all co-authors.

*Competing interests.* The authors declare that they have no conflict of interest.

*Acknowledgements.* We would like to thank Andy Aschwanden and one anonymous referee for their very helpful comments that help improving the overall quality and readability of the manuscript. K. Bulthuis would like to acknowledge Andreas Wernecke for personal communication about the manuscript. K. Bulthuis would like to acknowledge the Fonds de la Recherche Scientifique de Belgique (F.R.S.-FNRS) for its financial support (F.R.S.-FNRS Research Fellowship). Computational resources have been provided by the Consortium des Équipements de Calcul Intensif (CÉCI), funded by the Fonds de la Recherche Scientifique de Belgique (F.R.S-FNRS) under Grant No.2.5020.11.

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

**Figures**

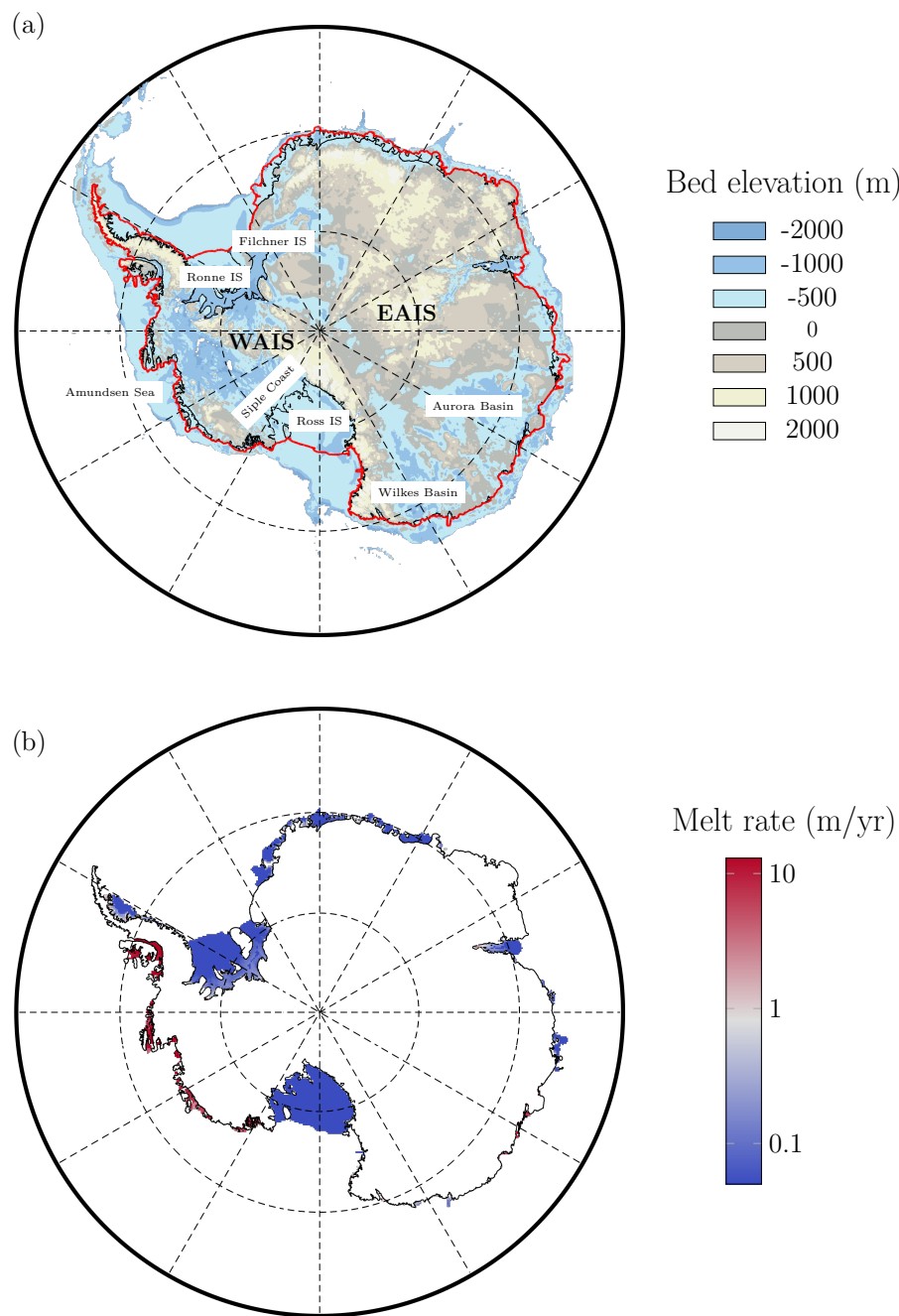

**Figure 1.** Antarctic bedrock topography and sub-shelf melting. (**a**) Bedrock topography (m a.s.l.) (Fretwell et al., 2013) of present-day Antarctic ice sheet and geographic features. WAIS is West Antarctic ice sheet, EAIS is East Antarctic ice sheet and IS is ice shelf. Grounding line is shown in black and ice front is shown in red. (**b**) Sub-shelf melt rate for present-day Antarctic ice sheet computed with the PICO ocean model (Reese et al., 2018a). Refreezing under ice shelves is not allowed.

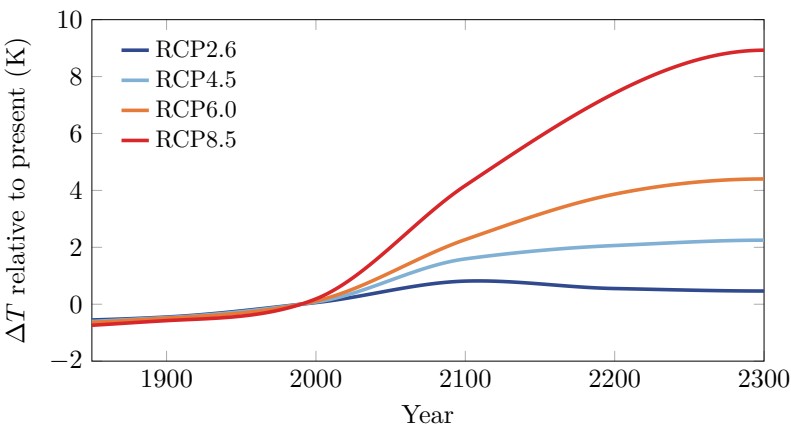

**Figure 2.** Long-term RCP temperature scenarios (Golledge et al., 2015) for Antarctica (60°-90° S) based on the CMIP5 data (Taylor et al., 2012) and extended to 2300. Temperatures are held constant after 2300.

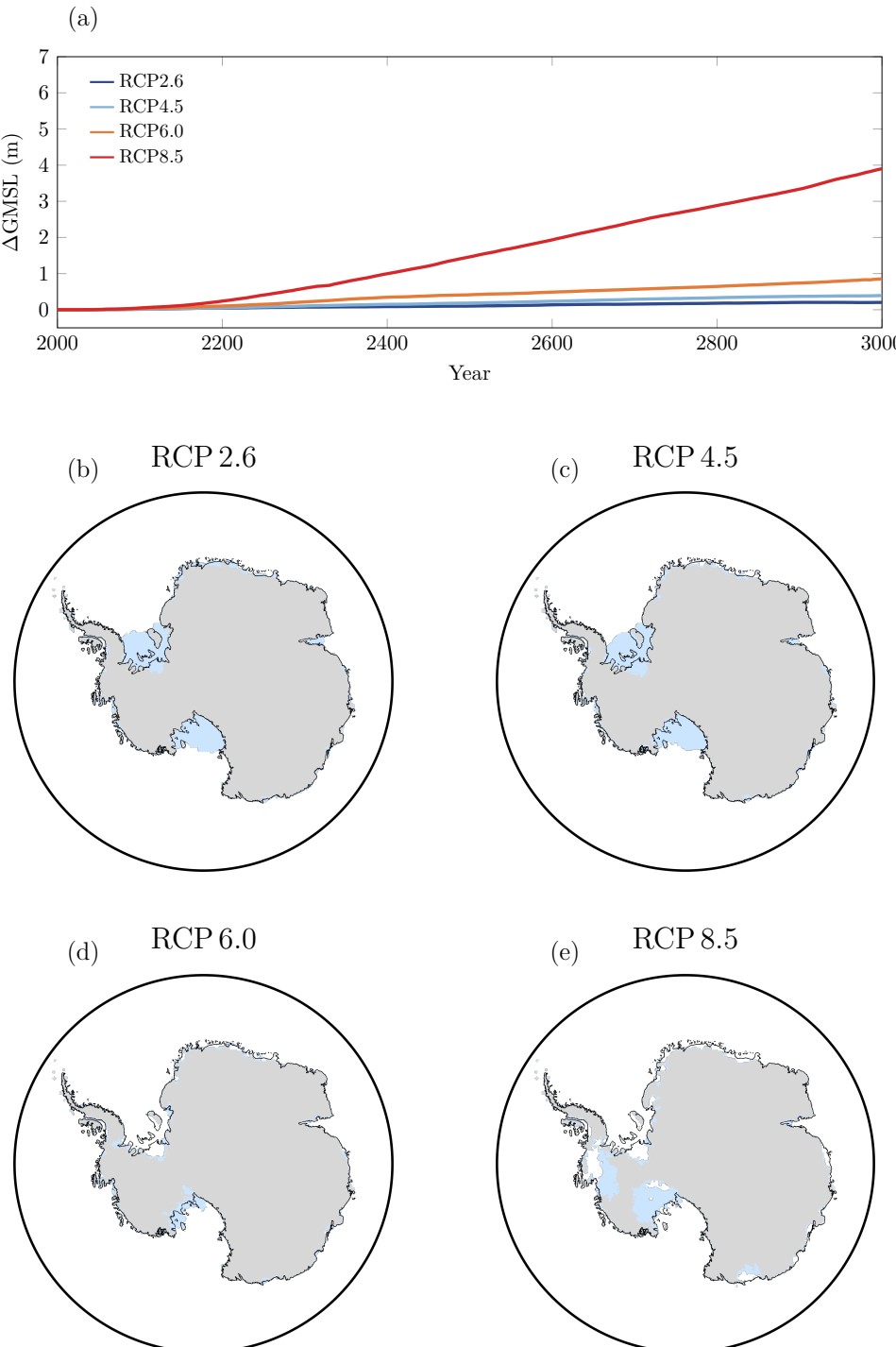

**Figure 3.** (**a**) Nominal AIS contribution to sea level. Grounded ice (grey area) and ice shelves (light blue area) by 3000 in (**b**) RCP 2.6, (**c**) RCP 4.5, (**d**) RCP 6.0 and (**e**) RCP 8.5 for the nominal values of the parameters. Present-day grounding line is shown in black.

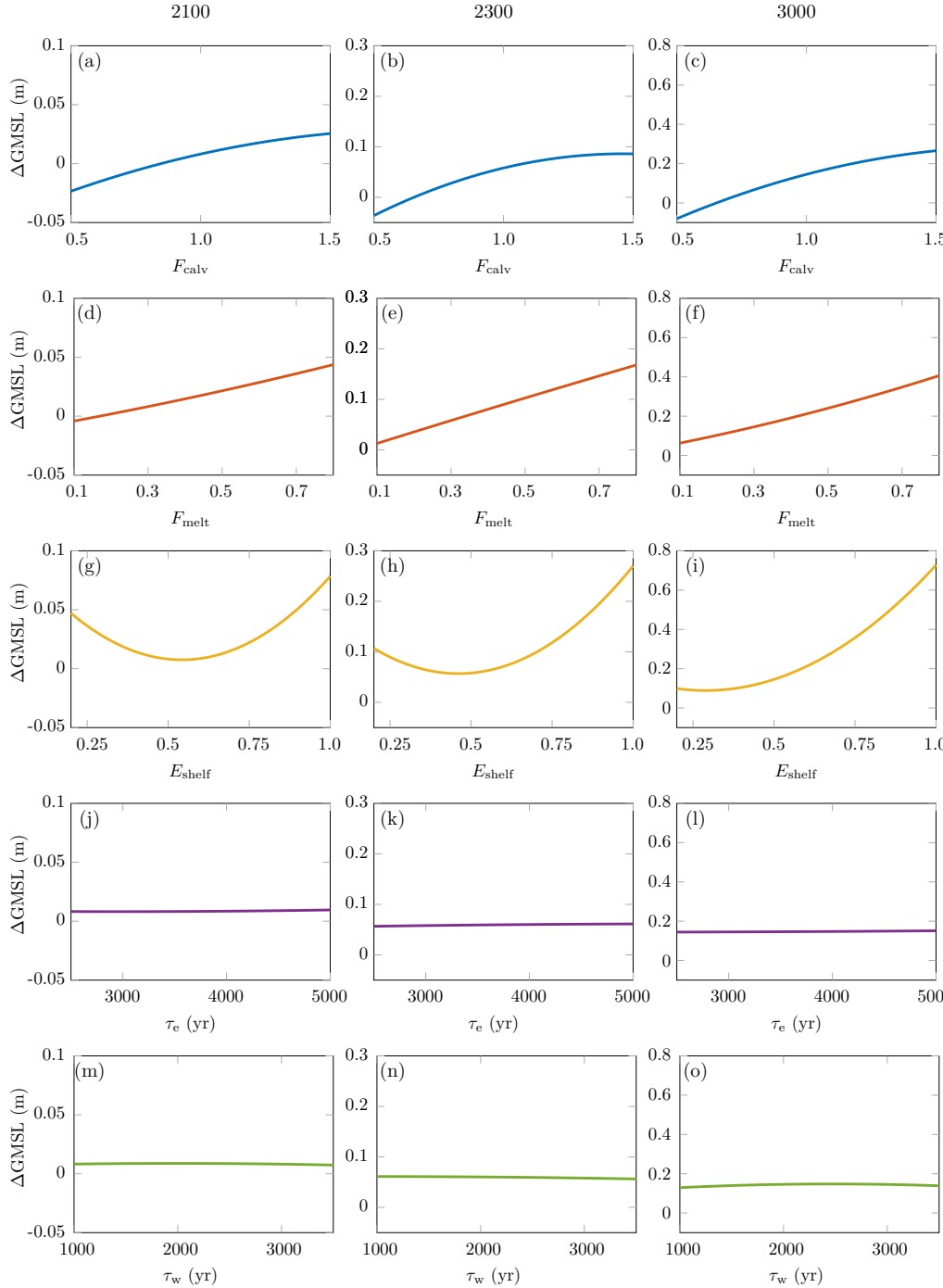

**Figure 4.** Representation of the parameters-to-projection relationship in RCP 2.6 under the weakly nonlinear sliding law as given by a PC expansion. The PC expansion is shown for each parameter individually with the other parameters fixed at their nominal value. PC expansions at (**a, d, g, j, m**) 2100, (**b, e, h, k, n**) 2300 and (**c, f, i, l, o**) 3000.

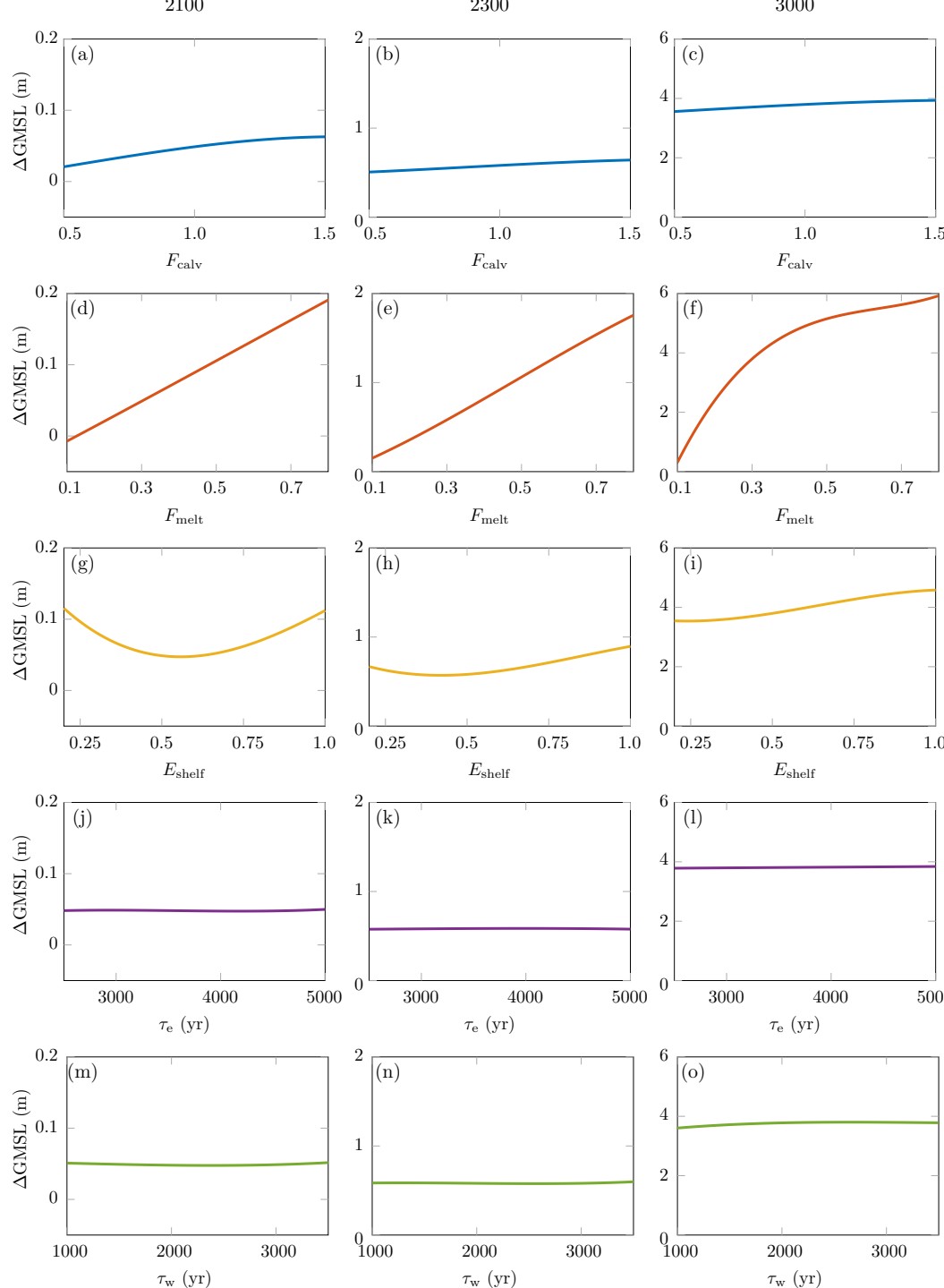

**Figure 5.** Same as Fig. 4 but in RCP 8.5.

(a)

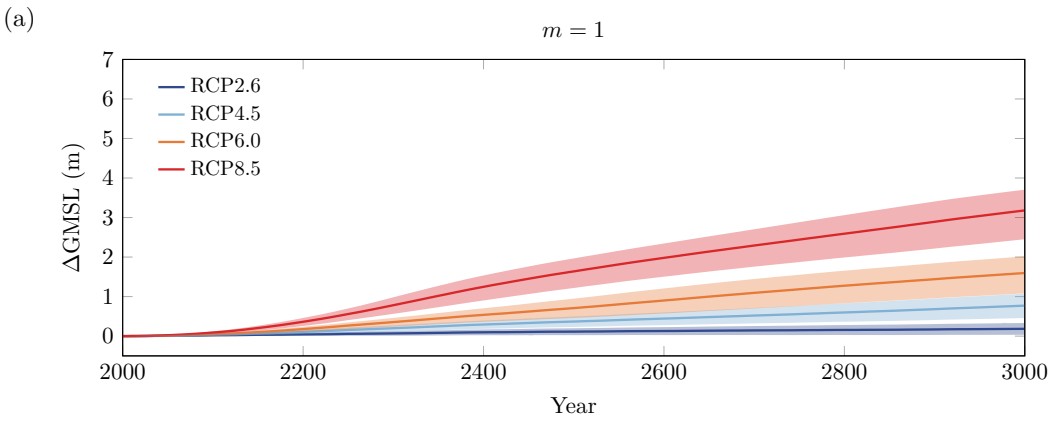

(b)

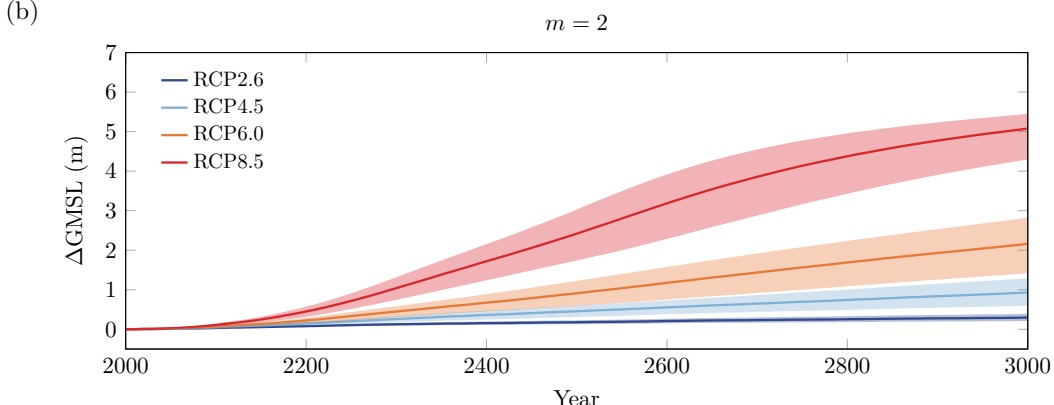

(c)

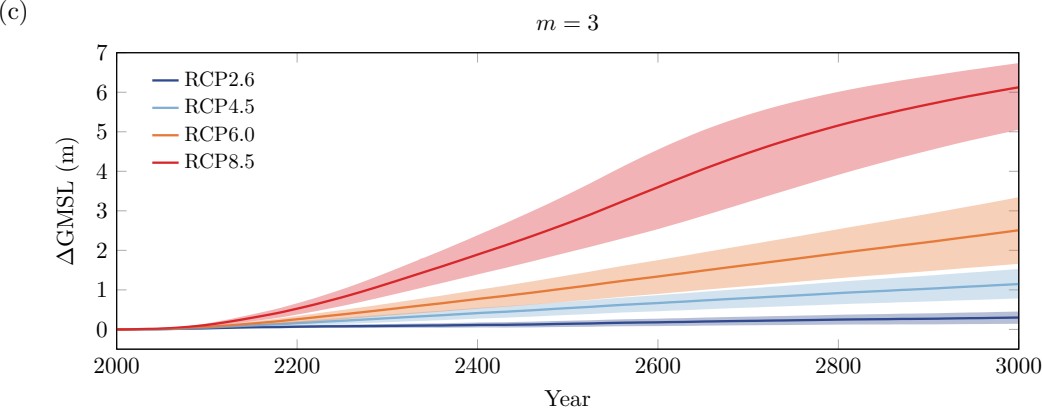

**Figure 6.** AIS contribution to sea level. (**a**) Viscous sliding law, (**b**) weakly nonlinear sliding law and (**c**) strongly nonlinear sliding law. Solid lines are the median projections and the shaded areas are the 33–66 % probability intervals that represent the parametric uncertainty in the model.

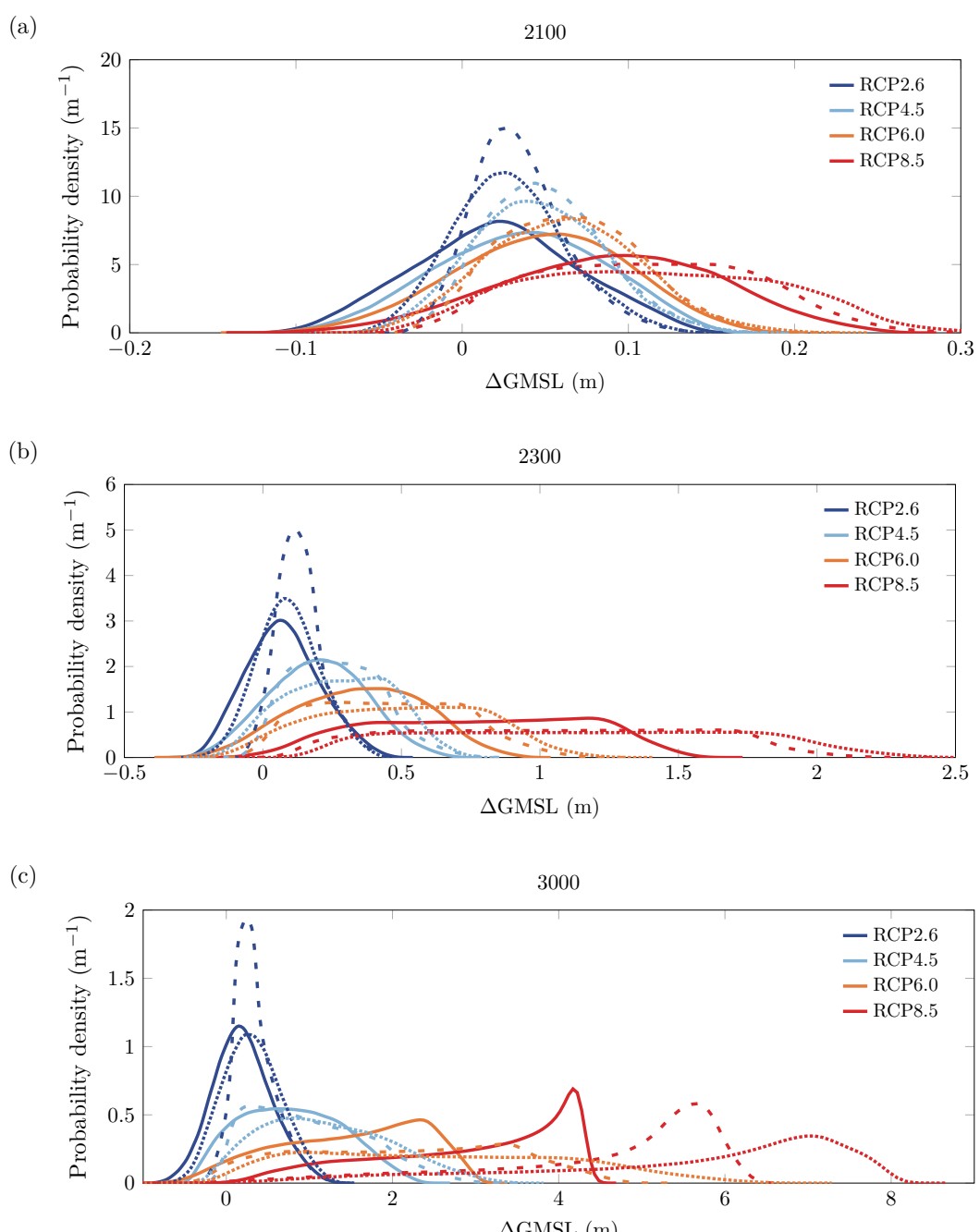

**Figure 7.** Probability density functions for the AIS contribution to sea level at (**a**) 2100, (**b**) 2300 and (**c**) 3000 under the viscous sliding law (solid lines), the weakly nonlinear sliding law (dashed lines) and the strongly nonlinear sliding law (dotted lines).

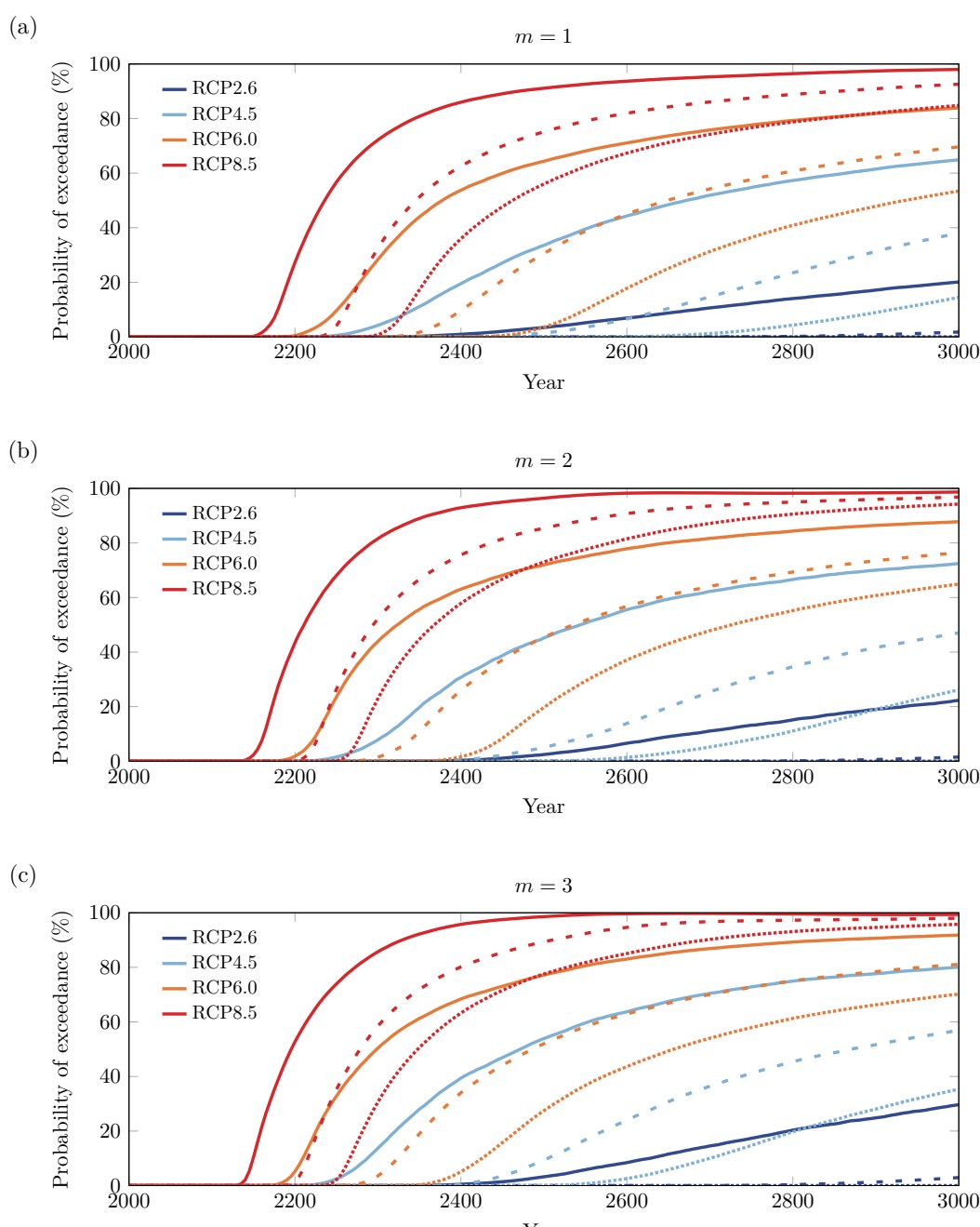

(a)

(b)

(c)

**Figure 8.** Probability of exceeding some characteristic threshold sea-level rise values as a function of time, evaluated from the complementary cumulative distribution functions of the probability density functions for $\Delta$GMSL. Probability of exceedance under (**a**) the viscous sliding law, (**b**) the weakly nonlinear sliding law and (**c**) the strongly nonlinear sliding law. Solid lines correspond to a threshold value of 0.5 m, dashed lines to a threshold value of 1 m and dotted lines to a threshold value of 1.5 m.

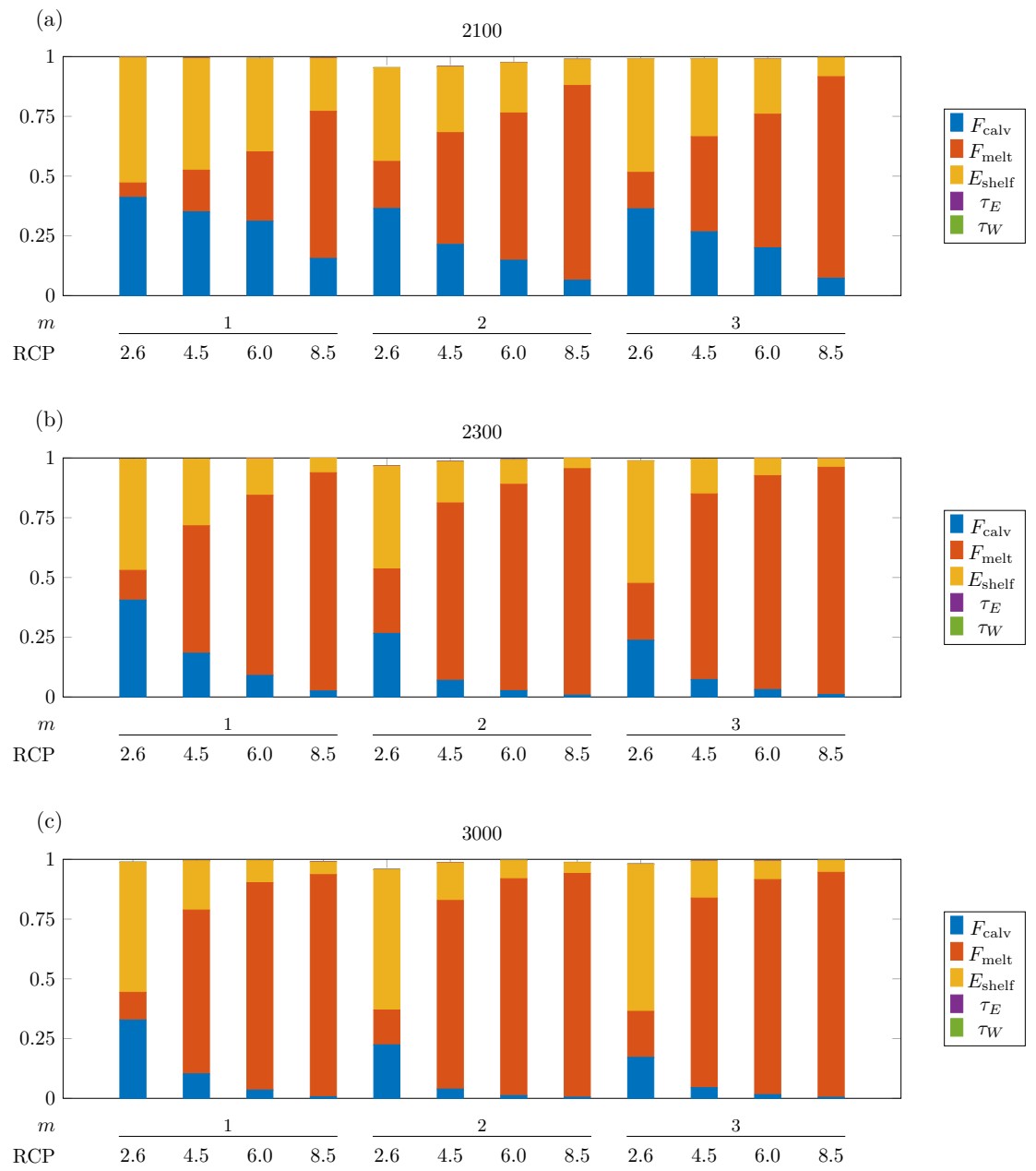

**Figure 9.** Sobol sensitivity indices for the AIS contribution to sea level in different RCP scenarios and for different values of the sliding exponent $m$ in Weertman's sliding law. Sobol indices at (**a**) 2100, (**b**) 2300 and (**c**) 3000. The gap between the height of a bar and the unit value represents the interaction index.

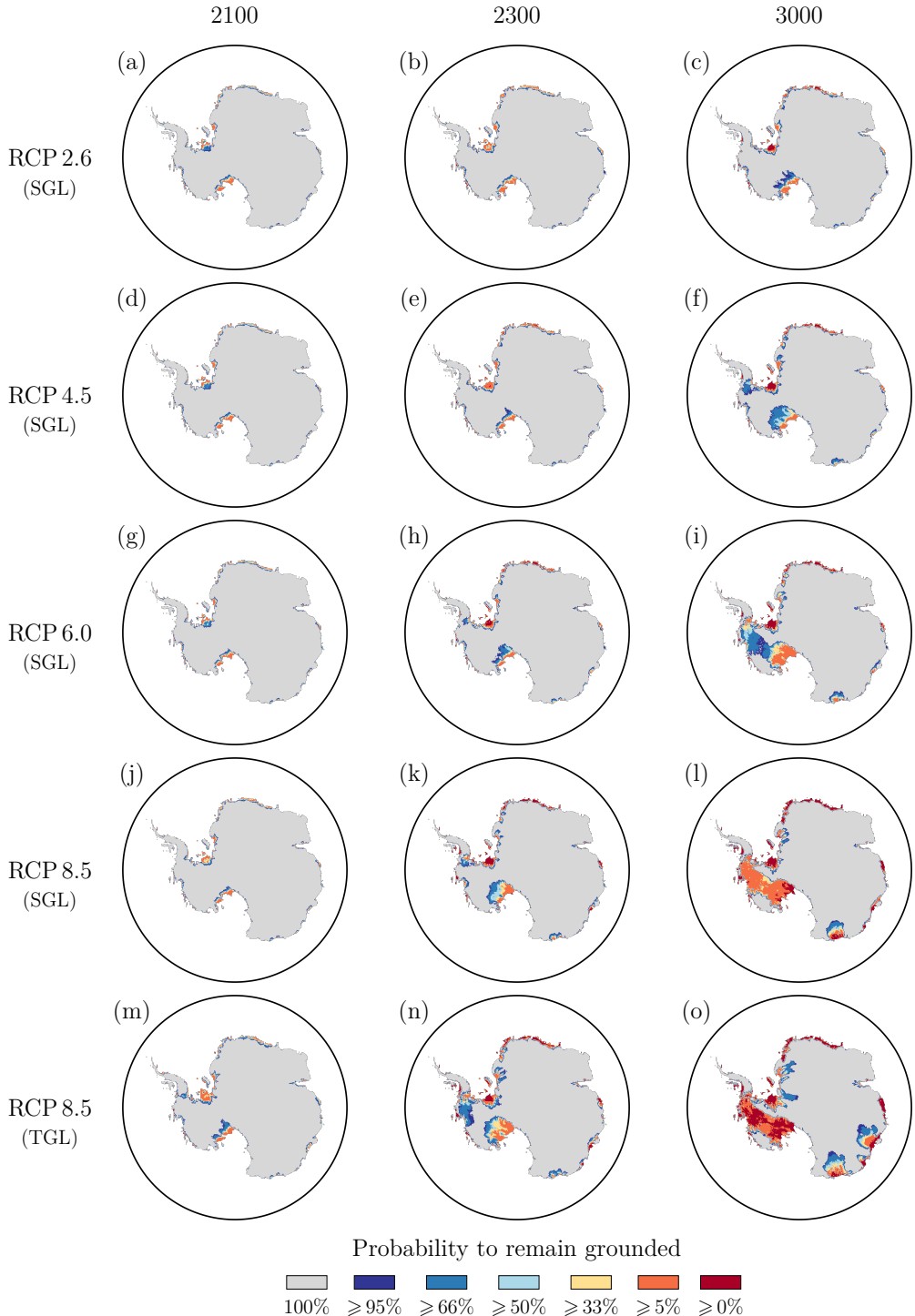

**Figure 10.** Confidence regions for grounded ice under the weakly nonlinear sliding law. Confidence regions are shown at (**a, d, g, j,m**) 2100, (**b, e, h, k,n**) 2300 and (**c, f, i, l,o**) 3000. Results under (**a–l**) the SGL parameterisation and (**m–o**) the TGL parameterisation.

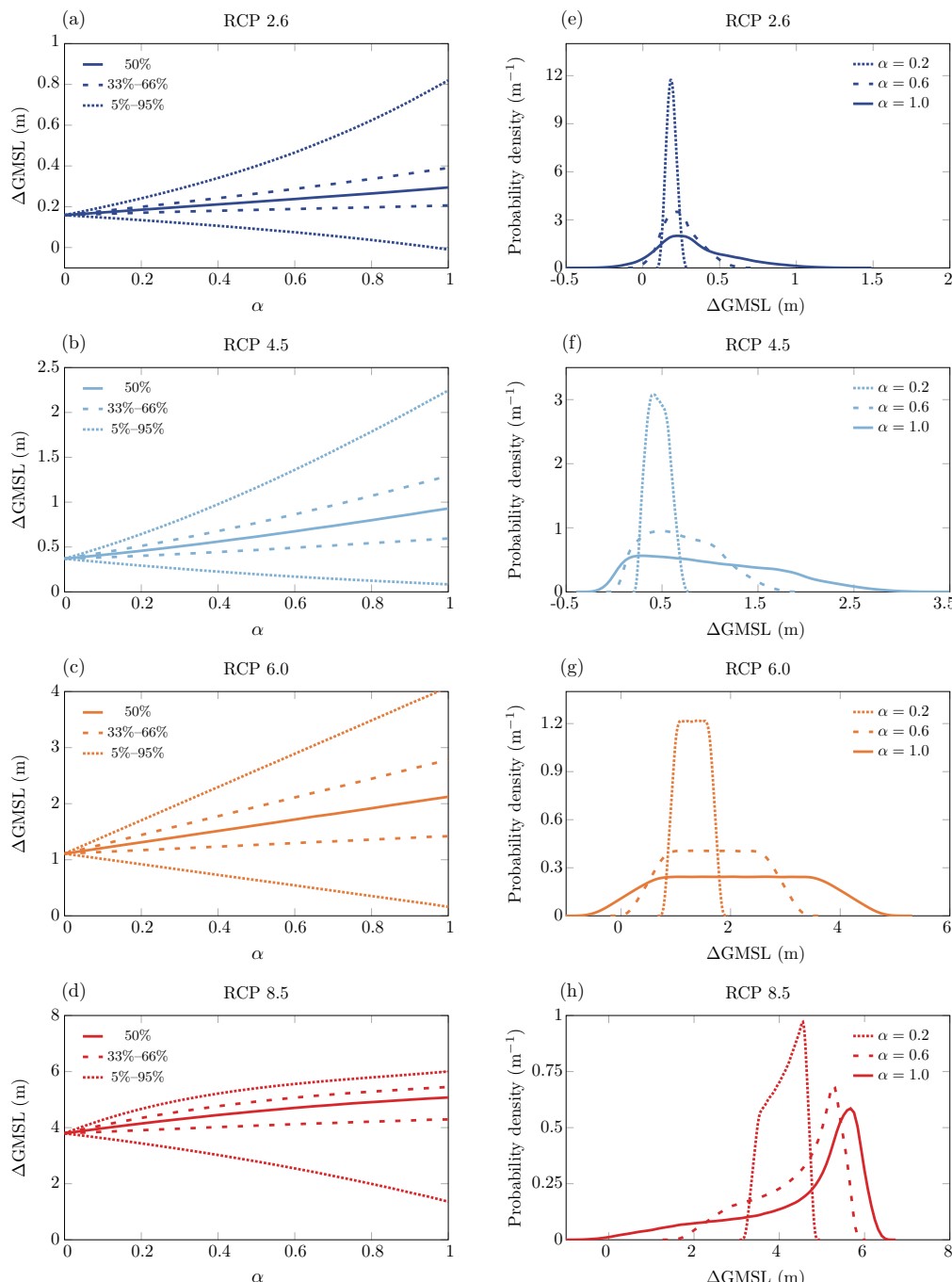

**Figure 11.** Robustness of probabilistic projections with respect to the scaling factor $\alpha$. Median projections and their 33–66 % and 5–95 % probability intervals by 3000 under for the weakly nonlinear sliding law in (**a**) RCP 2.6, (**b**) RCP 4.5, (**c**) RCP 6.0 and (**d**) RCP 8.5. Probability density functions for $\alpha = 0.2$ (dotted lines), $\alpha = 0.6$ (dashed lines) and $\alpha = 1$ (solid lines) in (**e**) RCP 2.6, (**f**) RCP 4.5, (**g**) RCP 6.0 and (**h**) RCP 8.5.

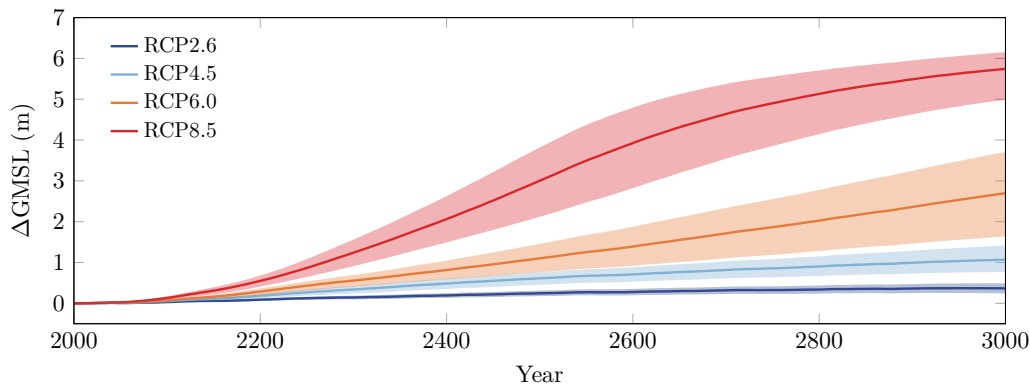

**Figure 12.** Same as Fig. 6 but under Weertman's sliding law with exponent $m = 5$.

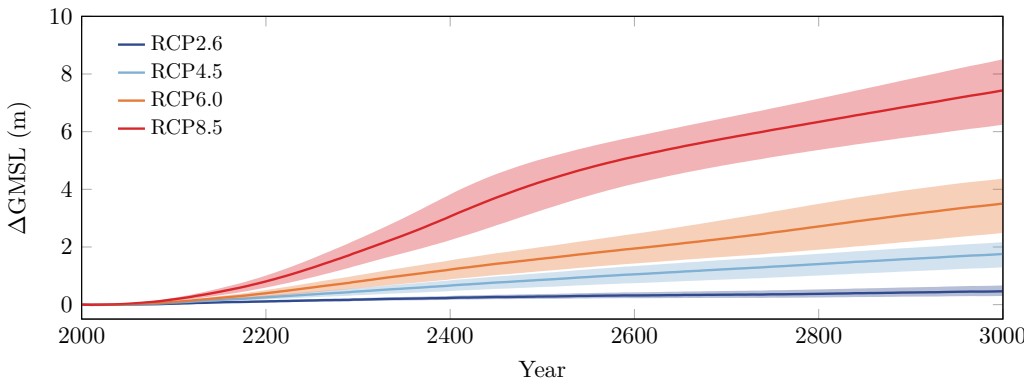

**Figure 13.** Same as Fig. 6 but under the TGL parameterisation and the weakly nonlinear sliding law.

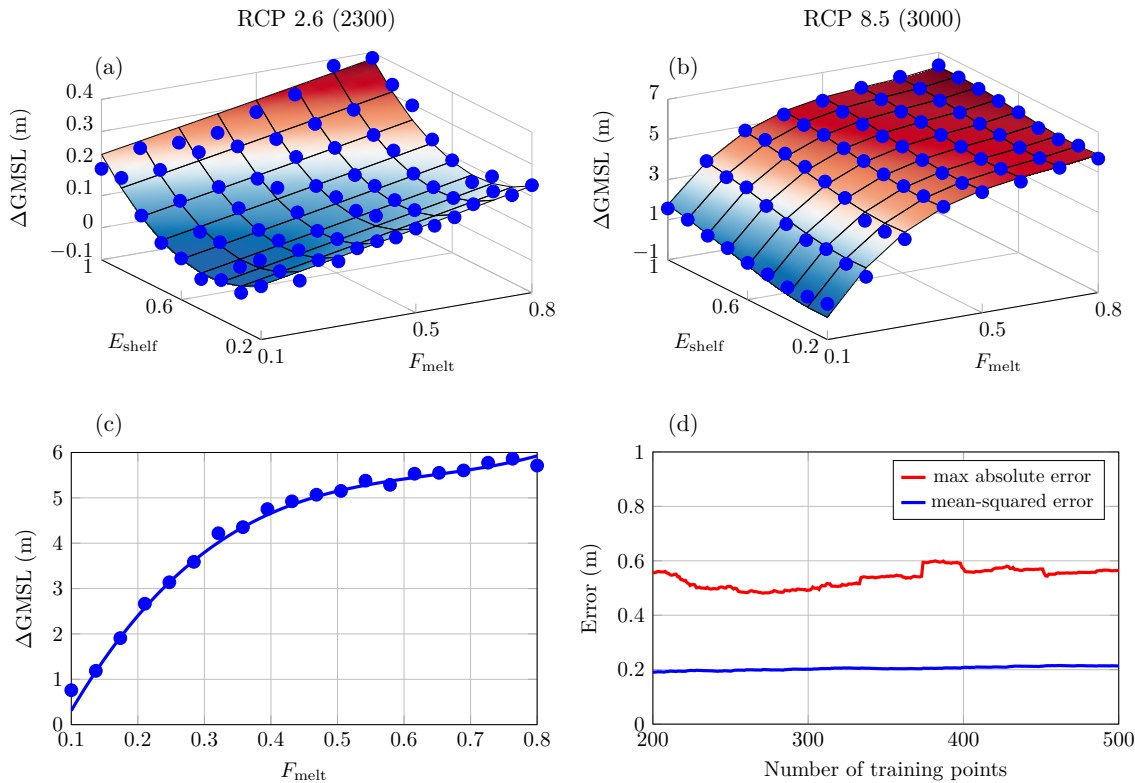

**Figure A1.** Validation tests for the PC expansion. (**a**) Cross-validation for the PC approximation in Fig. S2d and (**b**) cross-validation for the PC approximation in Fig. S2b. The surface represents the PC approximation of the response of the ice-sheet model and the blue dots are the exact response of the ice-sheet model at the test points. (**c**) Cross-validation for the PC approximation in Fig. 5f showing the presence of noisy results. (**d**) Convergence test in RCP 8.5 at time 3000 based on the number of training points used to build the emulator.