# Peer review of "Uncertainty quantification of the multi-centennial response of the Antarctic Ice Sheet to climate change"

_The Cryosphere, 2018_

## Referee Comment (RC1) · Anonymous Referee #1 · 5 Jan 2019

**1   Summary statement**

The manuscript "Uncertainty quantification of the multi-centennial response of the Antarctic Ice Sheet to climate change" by K. Bulthuis et al. provides an assessment of the response of the Antarctic ice sheet and the associated uncertainty to several climate change scenarios over the next few centuries, by combining numerical ice sheet model simulations and probabilistic methods. Using new probabilistic methods, and especially emulators, starts to be used in glaciology and provides an interesting alternative to costly ice flow models. The results show the relative impact of uncertainties in different ice flow model parameters as well as external parameters. They also suggest

that the marine ice sheet instability triggers large contribution to sea level for some combinations of parameters for the RCP 4.5 and 6.0 scenarios, while the instability is almost never triggered for the RCP 2.0 scenario and always triggered for RCP 8.5 scenario.

The paper is usually well written and the figures appropriate, but some references and background material are sometimes missing, especially in the introduction and the model references. I am also surprised by how the problems of the relatively low model resolution and simple parameterization of the grounding line are treated and not discussed in more details. Though it makes sense to use such configurations for large ensemble simulations of the entire Antarctic ice sheet, it seems a bit surprising that using such a configuration is needed when an emulator is used; I though that the emulator was allowing to run fewer but more accurate simulations used only for the calibration and then run a large number of cheap emulated runs to analyze the uncertainties in detail. This should at least be better acknowledged, especially in the limitations.

**2   Major comments**

The manuscript suggests that using a 20 km model resolution and a flux condition at the grounding line are reasonable assumptions and that the impact on the results is limited. I think this is a bit misleading as previous intercomparison experiments (Pattyn et al., 2012; 2013) have rather different conclusions. So this should be at least better explained and acknowledged in the limitations. Furthermore, I thought that the goal of using emulators was to limit the number of runs needed, and therefore that it would allow using more computationally expensive runs to calibrate the emulator, which does not seem to be the case here.

Similarly, basal melting under ice shelves is known to have a very large impact on ice
dynamics over both short and long timescales. What is done for future scenarios regarding this melt is however not clear and should be explained in more details. Also, the ocean warming follows a simple parameterization that mostly depends on atmospheric warming. We know that the future evolution of the Southern Ocean remains very uncertain and that changes in ocean circulation rather than ocean warming are expected to cause the largest changes, so that simple parameterizations are unlikely to accurately represent these changes. This should be more clearly acknowledged and it may be good to provide some perspectives on the unceratintes associated to the future ice shelf melt.

This manuscript is submitted to a cryosphere journal, so many readers (myself included) don't have a very strong background in probabilistic methods, so it would be great to detail terms that are not common. Also, it is not clear how the emulator is calibrated (with how many runs, how are these runs chosen, etc.) and why it needs to be calibrated separately for each RCP scenario. I naively thought that the goal was to calibrate over a wide range of climate conditions, and then be able to investigate different scenarios easily, so without the addition of new physical simulations.

References and background material are sometimes missing, especially in the introduction and in the description of the changes made to the model, so it would be good to add some references and to provide more context.

**3   Specific comments**

p.1 l.10: "except perhaps the relaxation times": Does it contribute to the uncertainty or not? It would be good to put clearer conclusions in the abstract, or to remove this part.

p.1 l.12: maybe quantify how more dominant it becomes with the different warming scenarios.

p.1 l.20: references missing

p.3 l.19: It would be good to define "confidence regions" and add some references.

p.3 l.26: "ice-sheet models; yet, whereas" → "ice-sheet models: whereas ..."

p.4 l.6: Add references for the SIA and SSA approximations.

p.4 l.11-12: I think it is little biased to simply say that using a flux-condition at the grounding line is an appropriate solution. There has been some recent research (Reese et al., 2018) demonstrating that this flux condition does not accurately represent confined ice shelves, which is the case for most ice shelves around Antarctica.

p.4 l.17: "the implementation of an ocean model": is there an ocean model implemented in f.ETISh or a representation of the ocean conditions? I think this part needs to be rephrased.

p.4 l.18: How is the positive degree day model changed? Also add references.

p.4 l.24: The resolution of the model is 20 km, which can be understood for such computations. However, using the argument that the model is "essentially scale independent" is surprising, especially as Frank Pattyn, who is a co-author on this paper, published quite different conclusions following the MISMIP experiments (Pattynet al., 2012; 2013). Furthermore 20 km represents only a few points for important ice shelves such as the Pine Island or Thwaites ice shelves, so it is not clear how accurately these glaciers can be modeled with such a resolution.

p.5 l.5: "linked to the atmospheric forcing": How? What is the link used? A few words should be added to briefly describe this or at least mention that it will be detailed later on.

p.5 l.16: What is the impact of this correction? Can we consider the response to be relatively linear to subtract the reference runs? This needs a bit of clarification.

p.5 l.20: How about ice shelf basal melt, which is a dominant driver of the response of

the Antarctic ice sheet?

p.6 Table 1: Caption: I don't think using "uncertain" is appropriate given that many other parameters are also uncertain. Consider changing. Also why use m=2 for the nominal value? I had the impression that the standard value is usually m=3.

p.6 l.2: I don't agree that the atmospheric forcing can cause large changes on the dynamics of the Antarctic ice sheet. It has a large impact on its volume and therefore sea level rise, but the impact on the dynamics is rather limited, unlike what is observed with changes in oceanic forcing.

p.7 l.18: it would be interesting to reference the buttressing capabilities of ice shelves around Antarctica (Furst et al., 2016).

p.7 l.28-30: It would be good to explain briefly the link between atmospheric and ocean warming (warming of the ocean surface, at depth, ...), especially as changes in ocean circulation rather than ocean warming are expected to happen around Antarctica.

p.9 l.32: Why is it necessary to build a different PC expansion for each RCP scenario? I would have imagined that the climate forcing was just a parameter that could be varied, similar to the other ones.

p.9 l.32: I don't really understand how the emulator is calibrated. Are the 500 forward simulations the training set or the results of the emulator? How many runs are used to calibrate the emulator and how are they chosen? It would be good to add some details in this section.

p.10 l.14: "subregions" → "regions" or "areas"

p.11 l.25-26: "making ice flux at the grounding line less important" → "reducing ice flux at the grounding line"

p.12 l.4-8: How different is it in this case? How significant are the changes for these additional simulations? It would be good to add numbers here, as the results are not

showed.

p.12 l.9: remove "significantly"

p.12 l.12: How about the order of magnitude difference in Fig.5 for example for several parameters? Where does this difference at different times come from? It would be good to discuss this.

p.13 l.5: Why would it be cooler given the values showed on Fig.2?

p.13 l.7: Is it possible to separate the impact of the ocean and the atmosphere?

p.13 l.16: Maybe add that this is because the forcing is so much larger for the high emission scenarios that the uncertainty caused by model parameters (considered similar in all the cases in your simulations) is therefore automatically reduced.

p.13 l.26: "sliding conditions in RCP 2.6" → "sliding conditions. In RCP 2.6"

p.15 l.22: "both regions": what regions are you talking about?

p.16 l.16: What is the distribution in these intervals? Is it always a uniform distribution?

p.16 l.21: I don't understand why it would be different.

Table 4 and Table 5 captions: maybe re-write what are the probability intervals.

p.19 l.11-12: I don't think the results presented show that the atmospheric forcing has a large impact on the ice sheet dynamics, at least for 2100.

p.19 l.21-22: I think this demonstrated mostly that the results are significantly impacted by the calibration of the basal melt forcing.

p.20 l.3: "the contribution of marine drainage basins": I don't understand what you mean here.

p.21 l.4-5: Add some references here, this is an important point.

p.21 l.14: What is the impact of this choice? How different would the results be with a

different choice for the construction of the emulator?

p.21 l.21: "perhaps": Does it impact it or not?

p.24 l.8: What is $g_I$?

Fig.1 caption: "neglected": Is it neglected because it is negligible or because the melt parameterization used does not allow to compute refreezing?

Fig.3: It would be good to add the ice shelves on panels b to e.

Fig.4 and Fig.5: Is it possible to use the same axis for the three columns so that the comparison is easier?

Fig.5: What causes the change of more than 1 order of magnitude at the different times for most of the parameters?

**4 References**

Furst, J., G. Durand, F. Gillet-Chaulet, T. Tavard, M. Rankl, M. Braun, and O. Gagliardini, The safety band of Antarctic ice shelves, Nat. Clim. Change, doi:10.1038/NCLIMATE2912, 2016.

Pattyn, F., et al., Results of the Marine Ice Sheet Model Intercomparison Project, MIS-MIP, Cryosphere, 6(3), 573–588, doi:10.5194/tc-6-573-2012, 2012.

Pattyn, F., et al., Grounding-line migration in plan-view marine ice-sheet models: results of the ice2sea MISMIP3d intercomparison, J. Glaciol., 59 (215), 410–422, doi:10.3189/2013JoG12J129, 2013.

Reese, R., T. Albrecht, M. Mengel, X. Asay-Davis, and R. Winkelmann, Antarctic sub-shelf melt rates via PICO, Cryosphere, 12, 1969–1985, doi:10.5194/tc-12-1969-2018, 2018.

---

## Author Comment (AC1) · 18 Jan 2019

*Response to the Interactive comment on* "Uncertainty quantification of the multi-centennial response of the Antarctic Ice Sheet to climate change" *by* Kevin Bulthuis et al.

We would like to thank anonymous referee #1 for the time dedicated to this manuscript and his/her constructive comments to improve the general quality and readability of the manuscript. We will try to give a proper response to his/her comments. The manuscript has been revised to include more references, background material and discussions

about the description and the limitations of the model and the methodology. For each referee's comment (written in blue), we included below a response (written in black) and proposed means to improve the manuscript.

**1   Summary statement**

The manuscript "Uncertainty quantification of the multi-centennial response of the Antarctic Ice Sheet to climate change" by K. Bulthuis et al. provides an assessment of the response of the Antarctic ice sheet and the associated uncertainty to several climate change scenarios over the next few centuries, by combining numerical ice sheet model simulations and probabilistic methods. Using new probabilistic methods, and especially emulators, starts to be used in glaciology and provides an interesting alternative to costly ice flow models. The results show the relative impact of uncertainties in different ice flow model parameters as well as external parameters. They also suggest that the marine ice sheet instability triggers large contribution to sea level for some combinations of parameters for the RCP 4.5 and 6.0 scenarios, while the instability is almost never triggered for the RCP 2.0 scenario and always triggered for RCP 8.5 scenario.

The paper is usually well written and the figures appropriate, but some references and background material are sometimes missing, especially in the introduction and the model references. I am also surprised by how the problems of the relatively low model resolution and simple parameterization of the grounding line are treated and not discussed in more details. Though it makes sense to use such configurations for large ensemble simulations of the entire Antarctic ice sheet, it seems a bit surprising that using such a configuration is needed when an emulator is used; I though that the emulator was allowing to run fewer but more accurate simulations used only for
the calibration and then run a large number of cheap emulated runs to analyze the uncertainties in detail. This should at least be better acknowledged, especially in the limitations.

We agree with this general comment and we have made changes to the manuscript to account for this comment. See answers below to major comments and specific comments to questions about p.4 l.11-12, p.4 l.24 and p.9 l.32.

**2 Major comments**

The manuscript suggests that using a 20 km model resolution and a flux condition at the grounding line are reasonable assumptions and that the impact on the results is limited. I think this is a bit misleading as previous intercomparison experiments (Pattyn et al., 2012; 2013) have rather different conclusions. So this should be at least better explained and acknowledged in the limitations. Furthermore, I thought that the goal of using emulators was to limit the number of runs needed, and therefore that it would allow using more computationally expensive runs to calibrate the emulator, which does not seem to be the case here.

The referee is right in pointing out that the model resolution and the flux condition should be discussed in more detail given that they represent two major limitations of the simulations. In our opinion, using a 20-km model resolution and a flux condition at the grounding line are acceptable assumptions in large-scale and long-term ice-sheet simulations and large-ensemble simulations (see discussion below).

We are aware of the limitations of the flux condition (short transients, buttressing) but we expect the impact of this parameterisation to be limited when compared to the impact of uncertainties in model parameters and future scenarios. Furthermore, Pattyn et al. (2012, 2013) have shown that the use of a flux condition at the grounding line allows to reproduce qualitatively the marine ice-sheet instability (MISI) and simulations of the Antarctic ice sheet using a flux condition (Pollard and DeConto, 2012; DeConto and Pollard, 2016) have also been able to reproduce qualitatively MISI in large-scale ice-sheet simulations. In addition, it is true that a higher resolution would allow to capture properly certain mechanisms that control grounding-line migration such as bedrock irregularities and ice-shelf pinning points as well as important small glaciers such as Pine Island and Thwaites glaciers, which represent only a few grid points with a 20-km resolution. Yet, Pattyn (2017) has shown that using the f.ETISh model with higher spatial resolutions gives similar results for given perturbations. We have modified Sect. 2.1 to discuss in more detail the applicability and limitations of using a flux condition at the grounding line (see also answer about p.4 l.11–12 in the specific comments).

The referee is also right in pointing out that using an emulator allows to limit the number of computationally expensive runs. Yet, in this manuscript, we are interested in performing an uncertainty quantification analysis of the model under different sources of uncertainty. A typical uncertainty quantification analysis with a Monte Carlo approach (or related approach such as Latin hypercube sampling) requires in general tenths/hundreds of thousands to millions of runs to achieve proper estimations for the statistical descriptors (mean, variance, . . . ) of the output. This is intractable even with a 20-km model resolution (with a computational cost of $\sim 8$h per forward simulation). In addition, we tried to investigate a sufficiently broad ensemble of uncertainties with an ensemble of five parameters and an ensemble of 20 distinct model configurations (4 RCP scenarios with 4 sliding laws and Tsai's flux condition). Each of these 20 distinct model configurations requires the construction of a different emulator because these model configurations represent discrete/categorical variables. For each of these configurations, we ran a set of 500 forward simulations (training set) to build the PC emulator, that is, we had to perform a total of 10000 forward simulations for

the 20 model configurations. This is still a large total computational time (given our computational ressources) even with a 20-km model resolution. The emulator does not replace the forward ice-sheet numerical model, which is necessary to capture all possible nonlinear effects of the ice-sheet response. The choice of a 20-km model resolution is a tradeoff between the total computational time and the number of distinct configurations and uncertain parameters investigated in this paper. We clarify the choice for this resolution in the manuscript and its limitations (notably the inability to represent accurately important ice streams such as Pine Island and Thwaites glaciers).

Similarly, basal melting under ice shelves is known to have a very large impact on ice dynamics over both short and long timescales. What is done for future scenarios regarding this melt is however not clear and should be explained in more details. Also, the ocean warming follows a simple parameterisation that mostly depends on atmospheric warming. We know that the future evolution of the Southern Ocean remains very uncertain and that changes in ocean circulation rather than ocean warming are expected to cause the largest changes, so that simple parameterisations are unlikely to accurately represent these changes. This should be more clearly acknowledged and it may be good to provide some perspectives on the uncertainties associated to the future ice shelf melt.

This suggestion is totally relevant given the very large impact of sub-shelf melting on ice dynamics and there is a clear need for more discussion about this process in the manuscript. The proper way to include ocean warming would actually require the coupling of the ice-sheet model with an ocean model in order to account for oceanic processes such as changes in ocean currents. Still, the representation of sub-shelf melting heavily depends on how ocean models represent these processes.

We have developed the discussion about sub-shelf melting and the limitations of its

representation in Sect 2. We have included a new description of the link between atmospheric and oceanic warmings in Sect. 2.1 and we have also included a more detailed discussion about the physics of sub-shelf melting and how it is implemented and linked with atmospheric forcing in the f.ETISH model in Sect. 2.2.5 and given more details about the limitations in Sect. 4.5 (see also the answers to the specific comments about p.4. l.17, p.5 l.5 and p.7 l.28–30 for more details about these changes in the manuscript).

This manuscript is submitted to a cryosphere journal, so many readers (myself included) don't have a very strong background in probabilistic methods, so it would be great to detail terms that are not common. Also, it is not clear how the emulator is calibrated (with how many runs, how are these runs chosen, etc.) and why it needs to be calibrated separately for each RCP scenario. I naively thought that the goal was to calibrate over a wide range of climate conditions, and then be able to investigate different scenarios easily, so without the addition of new physical simulations.

We agree that the manuscript should be made accessible to the largest possible audience. We tried to reduce technical jargon at a minimum while being consistent with a coherent probabilistic/uncertainty quantification framework.

The emulator is calibrated separately for each RCP scenario because we modelled the uncertainty in the climate forcing as a categorical variable (with one of the RCP scenarios). Constructing an (PCE) emulator for a categorical variable is not easy. In addition, we think that it is more coherent to build a separate emulator for each RCP scenario because we do not intent to study the uncertainty to climate forcing for a general forcing.

The emulator is built for each of the model configurations (that is RCP scenario and sliding law) from an ensemble of 500 training points (hence 500 forward simulations) in the parameter space of the parameters $F_{\text{calv}}$, $F_{\text{melt}}$, $E_{\text{shelf}}$, $\tau_{\text{e}}$ and $\tau_{\text{w}}$ with a maximin Latin hypercube sampling design.

See also answers below to questions about p.9 l.32 for additional details.

References and background material are sometimes missing, especially in the introduction and in the description of the changes made to the model, so it would be good to add some references and to provide more context.

We agree with this suggestion. We have added additional references and background material in the introduction and the description of the model (see answers to the specific comments for more details about these changes in the manuscript).

**3 Specific comments**

- p.1 l.10: "except perhaps the relaxation times": Does it contribute to the uncertainty or not? It would be good to put clearer conclusions in the abstract, or to remove this part.

Based on our experimental set-up, the relaxation times do no contribute to the uncertainty (at least significantly). We have removed the word "perhaps" to have clearer conclusions in the abstract and left the discussion about the impact of the relaxation times on the uncertainty in the main text.

- p.1 l.12: maybe quantify how more dominant it becomes with the different warming scenarios.

We have added in the abstract that the contribution of the uncertainty in sub-shelf melting to the uncertainty in sea-level rise projections goes from 5–25 % in RCP 2.6 to more than 90 % in RCP 8.5.

- p.1 l.20: references missing

We have added some references (Fretwell et al., 2013; Vaughan et al., 2013) for this sentence.

- p.3 l.19: It would be good to define "confidence regions" and add some references.

We have added a physical interpretation of a confidence region in glaciology to make it clearer. We also have added references (Bolin and Lindgren, 2015; French and Hoeting, 2015) regarding the definition and construction of confidence regions in uncertainty quantification analysis.

- p.3 l.26: "ice-sheet models; yet, whereas" ? "ice-sheet models: whereas ..."

The sentence has been changed following the referee's suggestion.

- p.4 l.6: Add references for the SIA and SSA approximations.

We have added references for the SIA (Hutter, 1983; Greve and Blatter, 2009) and SSA (Morland, 1987; MacAyeal, 1989; Weis et al., 1999) approximations.

[Figure]

- p.4 l.11-12: I think it is little biased to simply say that using a flux-condition at the grounding line is an appropriate solution. There has been some recent research (Reese et al., 2018b) demonstrating that this flux condition does not accurately represent confined ice shelves, which is the case for most ice shelves around Antarctica.

We agree that using a flux condition at the grounding line may be not appropriate to represent grounding-line migration notably in the presence of (strongly) buttressed ice shelves. We agree that a flux condition is an approximation (parameterisation) for the dynamics of the grounding line at coarse resolution but it has been shown to capture the steady-state behaviour of grounding lines under weak buttressing (Schoof 2007; Docquier et al., 2011; Pattyn et al, 2012). In addition, the implementation of a flux condition at the grounding line has been shown to reproduce qualitatively ice-sheet dynamics as determined with other ice-sheet models (especially the SSA model); yet, it cannot reproduce quantitatively changes in ice-sheet mass and sea-level rise contribution as determined from models with a higher level of complexity, such as the Blatter–Pattyn and the full Stokes models that include vertical shearing at the grounding line, especially for short transients (decadal time scales) as these flux conditions have been derived at steady state (Drouet et al., 2013; Pattyn et al., 2013; Pattyn and Durand, 2013). We are aware that a proper representation of grounding-line migration without the need for a flux condition would require a very fine grid resolution (possibly less than 200 metres), which is intractable for large-scale AIS modelling within an ensemble. Using a flux condition may also suffer limitations in the presence of buttressed ice shelves as suggested by Reese et al. (2018b). Yet, Reese et al. focus primarily on the physical representation of buttressing and not on the parameterisation of buttressing, which aims at translating a complex physical process into a simplified representation. Reese et al. also predict and discuss the appearance of non-physical negative buttressing, which is not allowed with parameterisations (Pollard et DeConto, 2012; Pattyn, 2017). Furthermore, the analysis of Reese et al. is purely diagnostic, while many of the effects described in their paper vanish with an

evolving ice sheet (as it is the case with the f.ETISh model).

We have written a new paragraph and added additional references (Docquier et al., 2011; Pattyn et al., 2012, 2013; Drouet et al., 2013; Pattyn and Durand, 2013) to discuss in more detail the applicability (reproduce the migration of the grounding-line at coarse resolution for the SSA model) and limitations (short transients, buttressing) of using a flux condition at the grounding line.

- p.4 l.17: "the implementation of an ocean model": is there an ocean model implemented in f.ETISh or a representation of the ocean conditions? I think this part needs to be rephrased.

We have changed this sentence to specify that sub-shelf melt rates are computed with an ocean-model coupler based on the Postdam Ice-shelf Cavity mOdel (PICO) ocean-coupler model (Reese et al., 2018a) rather than a simpler parameterisation of sub-shelf melting (Beckmann and Goosse, 2003; Holland et al., 2008; Pollard and DeConto, 2012; de Boer et al., 2015; Cornford et al., 2016).

- p.4 l.18: How is the positive degree day model changed? Also add references.

In f.ETISh version 1.0 (Pattyn, 2017), the positive degree-day model was based on the implementation by Huybrechts and de Wolde (1999), while it is based on the implementation by Janssens and Huybrechts (2000) in f.ETISh version 1.2 for numerical efficiency. We have changed the sentence to give more details about the description of atmospheric forcing in f.ETISh version 1.2 and added corresponding references. We now explain that the description of atmospheric forcing is based on a parameterisation of the changes in atmospheric temperature and precipitation rate (Huybrechts et al., 1998; Pollard and DeConto, 2012), a parameterisation of surface melt with a positive

degree-day model (Janssens and Huybrechts, 2000) and the inclusion of meltwater percolation and refreezing (Huybrechts and de Wolde, 1999).

- p.4 l.24: The resolution of the model is 20 km, which can be understood for such computations. However, using the argument that the model is "essentially scale independent" is surprising, especially as Frank Pattyn, who is a co-author on this paper, published quite different conclusions following the MISMIP experiments (Pattynet al., 2012; 2013). Furthermore 20 km represents only a few points for important ice shelves such as the Pine Island or Thwaites ice shelves, so it is not clear how accurately these glaciers can be modeled with such a resolution.

We agree that stating that the model is "essentially scale independent" is certainly a very strong statement and such statement should be avoided. Yet, Pattyn (2017) has suggested that his model was able to produce rather scale-independent results for a resolution of the model around 20 km due to the flux condition that makes the model results rather independent from the resolution (at least for the SSA model (Docquier et al., 2011; Pattyn et al., 2012)) and allows to reproduce MISI in large-scale ice-sheet simulations (the flux condition has been shown to pass the tests on reversibility, which is a major criterion in establishing MISI (Pattyn et al., 2012, 2013)). However, such spatial resolution cannot capture properly certain mechanisms that control grounding-line migration such as bedrock irregularities and ice-shelf pinning points even using sub-grid parameterisations of these mechanisms. Hence, we may expect discrepancies between our results and results at higher spatial resolutions (<5 km), especially for important small glaciers such as Pine Island and Thwaites glaciers, which represent only a few grid points with a 20-km resolution. Yet, using a higher spatial resolution (even with an emulator and given our computational ressources) remains rather intractable for large-scale and long-term ice-sheet simulations and large-ensemble simulations. We expect the uncertainty in the results due to the choice of the resolution to be limited when compared to the uncertainty in the results due to
the uncertainty in the model parameters and forcing.

We have changed this paragraph to discuss more thoroughly the choice and the limitations of using a 20-km resolution.

- p.5 l.5: "linked to the atmospheric forcing": How? What is the link used? A few words should be added to briefly describe this or at least mention that it will be detailed later on.

We have changed the sentence to make more explicit the link between the atmospheric forcing and the oceanic forcing. The link is explicitly written as $T_{oc} = T_{oc}^{obs} + F_{melt}\Delta T$, where $T_{oc}$ is the ocean temperature on the continental shelf, $T_{oc}^{obs}$ is the observed present-day ocean temperature on the continental shelf, $F_{melt}$ the ocean melt factor and $\Delta T$ the change in background atmospheric temperature. The equation for $T_{oc}$ with $F_{melt} = 0.25$ has been shown to reproduce trends in ocean temperatures following an analysis of the Climate Model Intercomparison Project phase 5 (CMIP5) data set (Taylor et al., 2012) for changes in atmospheric and ocean temperatures (Golledge et al., 2015).

- p.5 l.16: What is the impact of this correction? Can we consider the response to be relatively linear to subtract the reference runs? This needs a bit of clarification.

This is a good question. Such initial drifts are not necessarily undesirable (given that the present-day Antarctic ice sheet is not at equilibrium) and simply subtracting the results from the initial drift may be a questionable assumption as it assumes that the model response to perturbations is totally independent of this initial drift. Yet, we have decided to ignore these initial drifts for practical experimental considerations as carried out similarly in other experimental set-ups (Golledge et al., 2015; Goelzer et al., 2018,

Schlegel et al., 2018). The initial drifts for the reference runs reach a quasi-equilibrium at the end of the simulation and range from 0.1 to 0.2 metre of sea-level rise by 3000 for the different sliding conditions. The impact of the initial drift is more important for projections on a short-term time scale and has a limited impact on medium-term and long-term time scales. To isolate the impact of the sliding conditions on short-term projections and avoid any spurious impact of the initialisation procedure, we had to correct for the initial drifts, with differences between the reference runs for the different sliding laws comparable to the ordre of magnitude of the projections by 2100. We have added a bit of clarification about this correction in Sect. 2.1 and Sect. 4.5 and its limitations.

- p.5 l.20: How about ice shelf basal melt, which is a dominant driver of the response of the Antarctic ice sheet?

We did not mention the uncertainty in sub-shelf melting in the list as this source of uncertainty is accounted for through the uncertainty in a physical model parameter ($F_{\text{melt}}$). We agree that this source of uncertainty should be written more explicitly in the sentence given its significance for the response of the Antarctic ice sheet. Hence, we have added the phrase "sub-shelf melting" in the sentence. We have changed the overall sentence to list the uncertain physical processes that we consider and added a new sentence that explains that we account for uncertainty in these physical processes by considering an ensemble of uncertain parameters in the f.ETISh model.

- p.6 Table 1: Caption: I don't think using "uncertain" is appropriate given that many other parameters are also uncertain. Consider changing. Also why use m=2 for the nominal value? I had the impression that the standard value is usually m=3.

We have replaced the phrase "uncertain parameters" by "Parameters with probabilistic

representation" to avoid any ambiguity. We agree that the standard value is usually considered as $m = 3 = n$ (Weertman, 1957) with $n$ the exponent in Glen's flow law and this value has been used in numerous studies (Schoof, 2007; Pattyn et al., 2012, 2013; Brondex et al., 2017; Gladstone et al., 2017). Yet, a linear relationship ($m = 1$) is also used commonly in numerical simulations (Larour et al., 2012; Schäfer et al., 2012; Gladstone et al., 2014; Yu et al., 2018). Here, we have adopted an intermediate value ($m = 2$) between these two usual values ($m = 1$ and $m = 3$) as the nominal value of the sliding exponent in our simulations. We have added a new paragraph in Sect. 2.2.2 to clarify this (maybe unconvential) choice.

- p.6 l.2: I don't agree that the atmospheric forcing can cause large changes on the dynamics of the Antarctic ice sheet. It has a large impact on its volume and therefore sea level rise, but the impact on the dynamics is rather limited, unlike what is observed with changes in oceanic forcing.

We agree with this remark. We have changed the sentence to highlight that the atmospheric forcing is expected to cause changes in the mass balance of the Antarctic ice sheet rather than dynamical changes. We have added some references for this sentence (Lenaerts et al., 2016; Pattyn et al., 2018).

- p.7 l.18: it would be interesting to reference the buttressing capabilities of ice shelves around Antarctica (Furst et al., 2016).

We have followed the referee's suggestion and added this reference.

- p.7 l.28-30: It would be good to explain briefly the link between atmospheric and ocean warming (warming of the ocean surface, at depth, ...), especially as changes in ocean circulation rather than ocean warming are expected to happen around

[Figure]

Antarctica.

We have added a new paragraph that explains how ocean circulation results in sub-shelf melting and how changes in atmospheric and ocean warmings can affect ocean circulation. We have also added a new paragraph to explain how changes in atmospheric and ocean warmings are linked to sub-shelf melting in the PICO model (Reese et al., 2018a).

- p.9 l.32: Why is it necessary to build a different PC expansion for each RCP scenario? I would have imagined that the climate forcing was just a parameter that could be varied, similar to the other ones.

In our experimental set-up, the climate forcing (represented by the change in atmospheric temperature $\Delta T$) is represented as a discrete/categorical parameter with four different trajectories (the 4 RCP scenarios). This is similar to the sliding exponent, which can only take discrete values ($m = 1, 2, 3$ and $5$) in our experimental set-up. Actually, we consider 20 distinct model configurations given by each combination of RCP scenario with a sliding law ($m = 1, 2, 3$ or $5$) and each combination of RCP scenario with the TGL parameterisation. Building an emulator based on a PC expansion for discrete/categorical parameters/configurations is not straightforward and we prefer keeping them distinct for clarity.

- p.9 l.32: I don't really understand how the emulator is calibrated. Are the 500 forward simulations the training set or the results of the emulator? How many runs are used to calibrate the emulator and how are they chosen? It would be good to add some details in this section.

In our experimental setting, we consider 20 distinct model configurations given by each combination of RCP scenario with a sliding law ($m = 1, 2, 3$ or $5$) and each combination of RCP scenario with the TGL parameterisation. An emulator is built for each of these model configurations from an ensemble of 500 training points (hence 500 forward simulations) in the parameter space of the parameters $F_{calv}$, $F_{melt}$, $E_{shelf}$, $\tau_e$ and $\tau_w$ with a maximin Latin hypercube sampling design. In total, we carried out 10000 forward simulations of the f.ETISh model for the 20 model configurations. More details about the construction of the PC expansion are given in Appendix A. We have changed the paragraph and given more details about the construction of the PC expansion.

- p.10 l.14: "subregions" → "regions" or "areas"

The word "subregions" has been replaced by "regions"

- p.11 l.25-26: "making ice flux at the grounding line less important" ? "reducing ice flux at the grounding line"

The sentence has been changed to follow the referee's suggestion.

- p.12 l.4-8: How different is it in this case? How significant are the changes for these additional simulations? It would be good to add numbers here, as the results are not showed.

It is difficult to draw general conclusions as the results are dependent of the characterisation of the uncertainty in the bedrock relation times. Yet, we found in RCP 2.6 that the uncertainty in the bedrock relation time for West Antarctica can account for 10 % of the uncertainty in $\Delta$GMSL when $\tau_w$ is allowed to vary widely between 50 years and

3500 years. We wrote this number in the text to give the reader an idea about the influence of $\tau_w$.

- p.12 l.9: remove "significantly"

The word "significantly" has been removed from the manuscript.

- p.12 l.12: How about the order of magnitude difference in Fig.5 for example for several parameters? Where does this difference at different times come from? It would be good to discuss this.

As explained in Sec. 3.2. we used the PC expansions to visualise how the projections depend on each parameter individually (one-at-a-time) while keeping the other parameters fixed at their nominal value. This means in particular that $F_{\mathrm{melt}} = 0.3$ in Fig. 5(a–c) and Fig. 5(g–o) (this represents a background oceanic forcing). So the response of the Antarctic ice sheet is still driven strongly by ocean warming in RPC 8.5 (even if we assume no uncertainty in $F_{\mathrm{melt}}$). This explains the difference of magnitude between the difference time scales as ocean warming triggers MISI over time.

- p.13 l.5: Why would it be cooler given the values showed on Fig.2?

Beyond 2000 (start of our simulations), the RCP scenarios get warmer (RCP 8.5 warmer than RCP 6.0, RCP 6.0 warmer than RCP 4.5 and RCP 4.5 warmer than RCP 2.6). It is more likely to observe a decrease in sea-level rise in RCP 2.6 than for instance RCP 6.0 or RCP 8.5, that is, under cooler atmospheric conditions.

- p.13 l.7: Is it possible to separate the impact of the ocean and the atmosphere?

In our own opinion, it is not clear how to separate properly the impact of the ocean and the atmosphere because both are linked in our model. The atmospheric forcing (through the RCP scenario) and the ocean melt factor $F_{\text{melt}}$ both contribute to the uncertainty in the ocean temperature and therefore to the uncertainty in sub-shelf melting. Hence, the ocean forcing and the atmospheric forcing are dependent variables and their individual impact cannot be directly isolated. On the other hand, the five parameters $F_{\text{calv}}$, $F_{\text{melt}}$, $E_{\text{shelf}}$, $\tau_{\text{e}}$ and $\tau_{\text{w}}$ are (assumed) independent and their individual impact on the projections can be isolated. It would be possible to separate the impact of the ocean and the atmosphere using another representation of these processes. However, the direct impact of the atmospheric forcing is quite limited as sub-shelf melting (indirect impact of the atmospheric forcing) is for instance an order of magnitude larger than surface melt. So basically, we witness response to ocean warming.

- p.13 l.16: Maybe add that this is because the forcing is so much larger for the high emission scenarios that the uncertainty caused by model parameters (considered similar in all the cases in your simulations) is therefore automatically reduced.

We have added that this result is due to a much larger increase in the values of the $\Delta$GSLM projections compared to the increase in their dispersion when the RCP scenario gets warmer. Hence, the relative uncertainty in $\Delta$GSLM projections is automatically reduced.

- p.13 l.26: "sliding conditions in RCP 2.6" → "sliding conditions. In RCP 2.6"

This sentence has been corrected based on the referee's comment.

- p.15 l.22: "both regions": what regions are you talking about?

[Figure]

The phrase "both regions" refer to the 100 % confidence region for grounded ice and the present-day grounded ice region. We have modified the sentence to make it clearer. "differences between both regions are only of few tenths of kilometres by 3000" has been replaced by "the 100 % confidence region for grounded ice by 3000 only differ from the present-day grounded ice region by a few tenths of kilometres".

- p.16 l.16: What is the distribution in these intervals? Is it always a uniform distribution?.

The distribution is still uniform. In the other sections, we considered a uniform distribution with support $[F_{\text{calv,min}}, F_{\text{calv,max}}]$. In Sect. 3.6, the distribution is still uniform but with support $[F_{\text{calv,nom}} + \alpha(F_{\text{calv,min}} - F_{\text{calv,nom}}), F_{\text{calv,nom}} + \alpha(F_{\text{calv,max}} - F_{\text{calv,nom}})]$ and the size of the support (thus the uncertainty) is controlled by the parameter $\alpha \in [0, 1]$. Idem for the other parameters. We have changed the first paragraph in Sect. 3.6 to make it clearer.

- p.16 l.21: I don't understand why it would be different.

The emulator is an approximation to the parameters-to-projection relationship determined by the ice-sheet model. The difference between the response of the emulator and the ice-sheet model should of course be as small as possible.

- Table 4 and Table 5 captions: maybe re-write what are the probability intervals.

We have rewritten what are the probability intervals in the captions of Table 4 and Table 5.

- p.19 l.11-12: I don't think the results presented show that the atmospheric forcing

has a large impact on the ice sheet dynamics, at least for 2100.

We agree that this sentence is a bit too general and does not reflect the results shown in the manuscript. Hence, we have changed it to point out that the impact of atmospheric forcing becomes important for medium-term and long-term time scales.

- p.19 l.21-22: I think this demonstrated mostly that the results are significantly impacted by the calibration of the basal melt forcing.

The referee is right in his/her interpretation. We have changed the sentence as "Schlegel et al. (2018) found that grounding-line retreat is most significant in the Amundsen Sea sector under generalised ocean warming experiments for the Antarctic ice sheet, but, after calibrating sub-shelf melt rates with bounds that vary region by region and are assigned values deduced from the literature and model sensitivity studies, they found that the western Ronne basin has the greater sensitivity" to emphasise more clearly the impact of the calibration of the basal melt forcing.

- p.20 l.3: "the contribution of marine drainage basins": I don't understand what you mean here.

We mean that the significance of the contribution of the Antarctic ice sheet to sea-level rise under climate change is primarily controlled by the sensitivity, the response time and the vulnerability of its marine drainage basins. The phrase "the contribution of marine drainage basins" has been replaced by the more explicit phrase "the sensitivity, the response time and the vulnerability of its marine drainage basins"

- p.21 l.4-5: Add some references here, this is an important point.

We have followed the referee's suggestion and added some references (Timmermann et al., 2012; Depoorter et al., 2013; Rignot et al., 2013; Moholdt et al., 2014)

- p.21 l.14: What is the impact of this choice? How different would the results be with a different choice for the construction of the emulator?

This is a good point. We have checked that considering a PC expansion of order 4 instead of order 3 gave similar results to those in the manuscript. We may expect that for a PC expansion with a higher polynomial degree the results would be degraded due to the fitting of numerical noise (it has not been checked as the number of training points is insufficient to build a PC expansion with a too high polynomial degree) .

A discussion about a different choice for the construction of the emulator (maybe a Gaussian emulator) is beyond the scope of this paper but it would be worthwhile to gain insight into the efficiency of different choices of emulator in glaciology.

- p.21 l.21: "perhaps": Does it impact it or not?

Based on our experimental set-up, the relaxation times do no contribute to the uncertainty (at least significantly). We have removed the word "perhaps" to have clearer conclusions.

- p.24 l.8: What is $g_I$?

The term $g_I$ is just the remaining term of the decomposition of $g(\mathbf{x})$ in terms of the

mean value $g_0$ and the main effects $g_i$'s. It is simply given as

$$g_I(\mathbf{x}) = g(\mathbf{x}) - g_0 - \sum_{i=1}^{d} g_i(x_i). \tag{1}$$

We did not bring attention to this term as it is not required to compute the Sobol indices (and the interaction index $S_I$ can be computed as $S_I = 1 - \sum_{i=1}^{d} S_i$). We have added Eq. eq:gI in appendix B to clarify what is $g_I$.

- Fig.1 caption: "neglected": Is it neglected because it is negligible or because the melt parameterization used does not allow to compute refreezing?

We agree that this sentence is a bit ambiguous. Here, we do not allow for refreezing under ice shelves. Still, refreezing rates are in general small compare to melt rates. We have changed the caption to clarify that refreezing is not taken into account in the model.

- Fig.3: It would be good to add the ice shelves on panels b to e.

We have added the ice shelves as suggested by the referee.

- Fig.4 and Fig.5: Is it possible to use the same axis for the three columns so that the comparison is easier?

We prefer keeping different axes for the three columns as our goal for these figures is to show the relationship between each individual input parameter and ΔGMSL rather than comparing the magnitude of the response. Given the difference in the order of

magnitude of ∆GMSL for the different time scales, it would be difficult to clearly see the shape of the relationship with the same axis.

- Fig.5: What causes the change of more than 1 order of magnitude at the different times for most of the parameters?

See answer to question about p.12 l.12.

**4  References**

[revised manuscript text omitted]

---

## Referee Comment (RC2) · Aschwanden (Referee) · 1 Feb 2019

This manuscript uses a framework comprising an ice sheet model, an emulator, and uncertainty quantification methods to assess the response of the Antarctic Ice Sheet to climate change. The paper is generally well written and boasts beautiful figures. Its strength lies in the comprehensive probabilistic approach, with a thoughtful use various uncertainty quantification methods. While I remain suspicious of the applicability of emulators, as they could miss discontinuities and strong non-linearities, I must also admit that I am not sufficiently familiar with the topic to have an informed opinion.

Best wishes,

[Figure]

Andy Aschwanden

Major Comments:

I am a bit puzzled by the model setup. With reported computational costs of 16 CPU hours per simulation, one wonders if an emulator is indeed need as on the order of thousands simulations are computationally tractable? As discussed in the manuscript, the horizontal resolution of the ice sheet model is coarse, and topographic details will be missed. A demonstration of convergence under grid refinement would provide some confidence that the ice sheet model simulations are indeed robust. Nonetheless, one should not hold this too much against the manuscript, because, as I wrote above, the strength is clearly in the uncertainty quantification. Main-effect Sobel indices could become a standard tool in ice sheet modeling. Aschwanden et al (under review) use a similar approach for Greenland.

Why this confusing setup? P 10, l 26-26: "All results to follow have been obtained with the SGL parameterisation for the grounding-line migration, except in a discussion at the end of the section, where we discuss the impact of using the TGL parameterisation." Wouldn't it be easier to discuss the experimental design and results if the choice of grounding line parameterization were an additional parameters in the uncertainty quantification (i.e. by prescribing a uniform probability density function).

Minor Comments:

I find the introduction, discussion and use of "Uncertainty Community" somewhat awkward. While I think I understand what the authors try to express (and I like this approach), some clarification is warranted. What is the "recently formed uncertainty quantification community"? Maybe there is a white paper or similar that could be cited? UQ has been an integral part in numerical modeling outside glaciology but is only now making its way in ice sheet modeling (well, better late than never).

Detailed Comments:

[Figure]

everywhere: say "the contribution to sea-level is..." instead of "the contribution to sea-level rise". "Rise" is not needed here.

P 1

l 2-3: ...remain challenging due to...

l 10: sources of uncertainty, except bedrock relaxation time, contribute...

l 12: "as the scenario gets warmer" sounds awkward. Maybe "as temperatures rise"?

P 2

l 10-11: ..., with differences and uncertainty ranges of several meters of eustatic sea level.

l 24: ...initial state, climate forcing, and parameters...

P 3

l 2: ...based on probability theory,...

P 6

Table 1 and parameters. Rephrase "Uncertain parameters". Maybe "List of parameters and parameter ranges used in the uncertainty analysis" or similar. Also, it is not clear what distributions are used. Maybe the use of a "nominal" parameter suggests a Gaussian distribution. Or do all values in the parameter interval have equal probability? Note: I found that is is later explained on page 8, lines 24–31. Please add to the table legend that all parameters are drawn with equal probability. I am not sure that this is a good assumption though. Do you have any prior information that supports this?

P 7

l 15: Please explain what $F_{calv}$ is. It appears to be a scalar multiplier of something (a calving rate, a stress condition)?

P 8

l 29: "We limit the probabilistic characterisation to assuming uniform probability density functions and we do not address how this probabilistic characterisation could be refined by using expert assessment, data and statistical methods such as Bayesian inference." OK, that's fine. For the exponent of the sliding law, one could perform a calibration and compare simulated and observed surface speeds to assess how well a certain exponent is able to explain the observations. See Aschwanden, Fahnestock, and Truffer (2016). This then be used as a prior for describing a PDF (done in Aschwanden et al, under review).

P 9

l 32ff: I'm afraid I do not follow here. Please detail how many (and for which parameter configurations) the ice sheet model was run, and how the emulator was used to fill in the space (is 500 the number of ice sheet model or emulator runs)?

P 11

l 3-6. Change "Under nominal conditions, we find (Table 2) in RCP 2.6 a nominal AIS contribution to sea-level rise of 0.02 m by 2100, 0.07 m by 2300 and 0.20 m by 3000 and in RCP 8.5 a nominal AIS contribution to sea-level rise of 0.05 m by 2100, 0.59 m by 2300 and 3.9 m by 3000." to "Under nominal conditions, we find (Table 2) for RCP 2.6 an AIS contribution to sea-level rise of 0.02 m by 2100, 0.07 m by 2300 and 0.20 m by 3000 and for RCP 8.5 an AIS contribution to sea-level rise of 0.05 m by 2100, 0.59 m by 2300 and 3.9 m by 3000."

l 19 (and elsewhere): I'm not a native English speaker, but I think "Figure 4a–c shows that in RCP 2.6" should read "Figure 4a–c shows that for RCP 2.6" or "Figure 4a–c shows that under RCP 2.6".

l 23: "This quadratic dependence can be explained by the influence of $E_{shelf}$ on two competing processes: a higher value of $E_{shelf}$ softens the ice, thus leading to faster

ice flow in the ice shelves; but a higher value of $E_shelf$ also leads to ice-shelf thinning, thus reducing grounding line ice flux."

P 12

l 18ff: you use "median" but provide a range. Please clarify. Do the ranges represent the 16/84th percentiles, for example?

l 22-23: "...except for certain cases..."

l 26-27: "...and starts to increase around 2100..."

l 29: maybe reference Fig. 6 here?

l 31: "...which contributes 3–3.5 metres to sea-level, followed by a slower retreat of the East Antarctic ice sheet."

l 33: "by the year 3000"

P 13

l 1: "For RCP 2.6, we find (Table 3) 5–95% probability intervals..."

l 4: "...an increase in sea-level, though a decrease cannot be ruled out for a viscous sliding law and cooler atmospheric conditions."

P 14

l 6: It is interesting that for RCP 2.6 rheology contributes a similar amount 40-60

P 15

l 23: "ice is certain to..." I would refrain from using strong words like "certain".

P 17

l 4-10: Rather than writing down numbers (which are listed in Table 4 anyway), I suggest to tell the reader how the plastic sliding law compares to the intermediate case

since this is what one cares about.

l 18: "..., which could explain the smaller ice loss in our results under m = 5 than m = 3." That's interesting, I would not have guessed.

P 19

l 11: "the pivotal role played by atmospheric forcing". I think you mean the role of the emission scenario.

l 26: Moreover, the lower sensitivity of the Amundsen Sea sector in our simulations may arise...

P 20

l 6-10: Very long sentence, maybe split into two. Figures

l 23-29: I understand what you are trying to say but I'm not sure the formulation "does (not) question the nominal projections". Maybe "in agreement with" or similar? The probabilistic framework expands upon the nominal projections, and your results provide good evidence that a thorough risk assessment must include UQ.

P 21

l 20: We found that all investigated sources...

3 b-e: Show outline of present-day grounded area for better visual comparison.

9: What does the $m$ in the lower left corners of the plot mean?
* * *

---

## Author Comment (AC2) · 13 Feb 2019

*Response to the Interactive comment on* "Uncertainty quantification of the multi-centennial response of the Antarctic Ice Sheet to climate change" *by* Kevin Bulthuis et al.

We would like to thank Andy Aschwanden for the time dedicated to this manuscript and his constructive comments to improve the general quality and readability of the manuscript. We will try to give a proper response to his comments and revise the manuscript accordingly. For each referee's comment (written in blue), we included

below a response (written in black) and proposed means to improve the manuscript.

**1 Summary statement**

This manuscript uses a framework comprising an ice sheet model, an emulator, and uncertainty quantification methods to assess the response of the Antarctic Ice Sheet to climate change. The paper is generally well written and boasts beautiful figures. Its strength lies in the comprehensive probabilistic approach, with a thoughtful use various uncertainty quantification methods. While I remain suspicious of the applicability of emulators, as they could miss discontinuities and strong non-linearities, I must also admit that I am not sufficiently familiar with the topic to have an informed opinion.

Best wishes,

Andy Aschwanden

We would like to thank Andy Aschwanden for this kind summary statement. The referee is right in pointing out that discontinuities and nonlinearities could pose challenges to the construction of emulators. However, we believe that our approach is valid when considering the global response of the Antarctic ice sheet (AIS) (e.g. $\Delta$GMSL) as we may expect possible local and regional strong non-linearities and discontinuities to be smoothed out at the continental scale. Our belief is further backed by other studies of the continental response of the Antarctic ice sheet (Ritz et al., 2015; Golledge et al., 2015, 2017; DeConto and Pollard, 2016; Schlegel et al., 2018) that also suggest that this response does not exhibit strong non-linearities and discontinuities. Conversely, we did not compute the confidence regions with an emulator as the applicability of an emulator to the local/regional behaviour of the AIS is more questionable and we used

directly the training points and forward simulations to estimate them. We leave for a further work the investigation of emulators for estimating confidence regions (Bulthuis et al., in preparation). We have added a sentence in the manuscript to clarify the use of an emulator to represent the continental response of the Antarctic ice sheet, as measured through $\Delta$GMSL.

**2 Major comments**

I am a bit puzzled by the model setup. With reported computational costs of 16 CPU hours per simulation, one wonders if an emulator is indeed need as on the order of thousands simulations are computationally tractable? As discussed in the manuscript, the horizontal resolution of the ice sheet model is coarse, and topographic details will be missed. A demonstration of convergence under grid refinement would provide some confidence that the ice sheet model simulations are indeed robust. Nonetheless, one should not hold this too much against the manuscript, because, as I wrote above, the strength is clearly in the uncertainty quantification. Main-effect Sobel indices could become a standard tool in ice sheet modeling. Aschwanden et al (under review) use a similar approach for Greenland.

We agree with the referee that a resolution of 20 km may be not able to capture properly certain mechanisms that control grounding-line migration such as bedrock irregularities and ice-shelf pinning points as well as important small glaciers such as Pine Island and Thwaites glaciers. We refer the referee to our response to anonymous referee # 1 regarding the limitations of this approach and the changes we brought at the manuscript to discuss the applicability and limitations of this approach (https://doi.org/10.5194/tc-2018-220-AC1).

The referee is also right in pointing out that our ice-sheet numerical model would allow to perform thousands of simulations, thus questioning the (full) need for an emulator. Yet, this approach would be feasible only for a few configurations of the model. In our manuscript, we investigated a set of 20 distinct configurations given by each combination of RCP scenario with a sliding law ($m = 1, 2, 3$ or $5$) and each combination of RCP scenario with the TGL parameterisation in an attempt to investigate a sufficiently broad ensemble of model configurations. For each of the configurations, we ran a set of 500 forward simulations (training set) to build a PC emulator as performing thousands of simulations for each of the configurations would have been less computationally tractable. In addition, we had two definite goals in writing down this manuscript such as (1) providing the reader with a comprehensive probabilistic approach and various uncertainty quantification methods for uncertainty analysis in ice-sheet models (as highlighted by the referee) and (2) investigating a sufficiently broad ensemble of uncertainties to give a large insight into the impact of uncertainties in ice-sheet models.

Following the referee's suggestion, we have added a new figure (Fig. S1 in the supplementary material or Fig. 1 in the response) in the supplementary material that provides a comparison for the AIS contribution to sea level as a function of spatial resolution (20 km in Fig. 1a vs 16 km in Fig. 1b) to give an idea about the impact of the model resolution on our results. This figure suggests that the uncertainty in the projections due to the model resolution is (far) less important than the uncertainty due to the uncertainty in the parameters.

Why this confusing setup? p.10, l.26–26: "All results to follow have been obtained with the SGL parameterisation for the grounding-line migration, except in a discussion at the end of the section, where we discuss the impact of using the TGL parameterisation." Wouldn't it be easier to discuss the experimental design and results if the choice of grounding line parameterization were an additional parameters in

the uncertainty quantification (i.e. by prescribing a uniform probability density function).

We agree with the referee that this sentence sounds a bit awkward and does not bring any additional information to the reader, so we have removed it from the manuscript. In addition, we make it clear in Sect. 2.2.3 that most results were obtained with Schoof's grounding-line parameterisation (reference parameterisation) and Tsai's grounding-line parameterisation is only discussed in Sect. 3.8. The choice of the grounding-line parameterisation can be seen as an additional parameter, but as this parameter is categorical (either Schoof's grounding-line parameterisation or Tsai's grounding-line parameterisation) it cannot be prescribed with a probability density function. We could have assigned a weight to the choice of the grounding-line parameterisation, but we have no evidence for a suitable choice of weights.

**3 Minor comments**

I find the introduction, discussion and use of "Uncertainty Community" somewhat awkward. While I think I understand what the authors try to express (and I like this approach), some clarification is warranted. What is the "recently formed uncertainty quantification community"? Maybe there is a white paper or similar that could be cited? UQ has been an integral part in numerical modeling outside glaciology but is only now making its way in ice sheet modeling (well, better late than never).

We agree that our terminology might sound a bit awkward, especially for most readers of the Cryosphere journal who are not familiar with uncertainty analysis. Uncertainty analysis has been a long standing part of numerical/experimental studies in science and engineering, but the coherent formalisation of uncertainty analysis in a probabilistic framework is rather recent and is still an ongoing work. The phrase "Uncertainty

Community" refers to this community that aims at formalising uncertainty quantification analysis. In this manuscript, we have tried to bridge the gap between UQ outside of glaciology and glaciology. To avoid any ambiguity, we have decided to remove the phrase "Uncertainty Community" from the manuscript and talked more generally about UQ analysis rather than referring to the "Uncertainty Community".

**4   Detailed comments**

- Everywhere: say "the contribution to sea-level is..." instead of "the contribution to sea- level rise". "Rise" is not needed here.

We have followed the referee's suggestion and changed the phrase "the contribution to sea-level rise" by "the contribution to sea level" all along the manuscript.

- p.1 l.2–3: . . . remain challenging due to . . .

The words "to establish" have been removed from the manuscript.

- p.1 l.10: sources of uncertainty, except bedrock relaxation time, contribute . . .

We have changed the sentence in the manuscript based on the referee's suggestion.

- p.1 l.12: "as the scenario gets warmer" sounds awkward. Maybe as "temperatures rise"?

We agree with the referee that this phrase sounds a bit awkward. We have replaced it by "as atmospheric and oceanic temperatures rise" following the referee's suggestion.

- p.2 l.10–11: . . . , with differences and uncertainty ranges of several meters of eustatic sea level.

We have changed the sentence in the manuscript based on the referee's suggestion.

- p.2 l.24: . . . initial state, climate forcing, and parameters . . .

We thank the referee for this suggestion. Yet, we tried not to use the Oxford (serial) comma all along the manuscript unless necessary to remain consistent in the manuscript (as suggested in the manuscript preparation guidelines for authors in the Cryosphere jornal).

- p.3 l.2: . . . based on probability theory . . . .

We have changed the phrase in the manuscript based on the referee's suggestion.

- p.6 Table 1 and parameters: Rephrase "Uncertain parameters". Maybe "List of parameters and parameter ranges used in the uncertainty analysis" or similar. Also, it is not clear what distributions are used. Maybe the use of a "nominal" parameter suggests a Gaussian distribution. Or do all values in the parameter interval have equal probability? Note: I found that is is later explained on page 8, lines 24–31. Please add to the table legend that all parameters are drawn with equal probability. I am not sure that this is a good assumption though. Do you have any prior information that supports this?

We have rephrased "Uncertain parameters" as "List of parameters and parameter ranges used in the uncertainty analysis" following the referee's suggestion (The inappropriateness of the phrase "uncertain parameters" was also pointed out by anonymous referee #1 (https://doi.org/10.5194/tc-2018-220-RC1)). We use the term "nominal value" to refer to the accepted/reference value of a parameter that we would have used in the absence of an uncertainty analysis, without any reference to an underlying probability distribution for the parameters.

In addition, we did not specify the probability distribution for the parameters in Table 1 as we only intended to discuss the sources of uncertainty in Sect. 2.2 and discuss the uncertainty quantification framework we used in Sect. 2.3 (notably the probabilistic characterisation of the parameters is given in Sec. 3.2.1).

We agree that assuming uniform probability distributions for the parameters may not be an appropriate assumption but as discussed in Sec. 3.2.1, a proper characterisation of the probability distributions of the parameters with expert assessment, data and statistical methods is beyond the scope on this paper. While a more refined uncertainty characterisation could be carried out, we used a uniform probability distribution for the parameters as this distribution is known to satisfy the principle of indifference (a.k.a the principle of maximum entropy) in the absence of any prior information about the distribution of the parameters except for their minimum and maximum values (Kapur, 1989), which we assume here for the sake of simplicity. Here, we determined the minimum and maximum values for the parameters based on expert assessment and literature.

- p.7 l.15: Please explain what $F_{calv}$ is. It appears to be a scalar multiplier of something (a calving rate, a stress condition)?

The referee is right in his interpretation of $F_{\text{calv}}$. It is a scalar multiplier factor of the calving rate. Based on the referee's comment, we have changed Sect. 2.2.4 to give more details about the parameterisation of the calving rate in our model and the actual meaning of $F_{\text{calv}}$ (scalar multiplier factor of the calving rate).

- p.8 l.29: "We limit the probabilistic characterisation to assuming uniform probability density functions and we do not address how this probabilistic characterisation could be refined by using expert assessment, data and statistical methods such as Bayesian inference." OK, that's fine. For the exponent of the sliding law, one could perform a calibration and compare simulated and observed surface speeds to assess how well a certain exponent is able to explain the observations. See Aschwanden, Fahnestock, and Truffer (2016). This then be used as a prior for describing a PDF (done in Aschwanden et al, under review).

We thank the referee for his thoughtful comment. We have added additional references (Aschwanden et al., 2016; Gillet-Chaulet et al., 2016) to clarify the fact that the probabilistic characterisation of the sliding exponent can be based on a calibration of the sliding exponent based on a comparison between simulated and observed surface velocities.

- p.9 l.32: I'm afraid I do not follow here. Please detail how many (and for which parameter configurations) the ice sheet model was run, and how the emulator was used to fill in the space (is the 500 the number of ice sheet model or emulator runs).

This paragraph clearly lacks some explanations as already pointed out by referee #1 (https://doi.org/10.5194/tc-2018-220-RC1). In our experimental set-up, we consider 20 distinct model configurations given by each combination of RCP scenario with a sliding law ($m = 1, 2, 3$ or $5$) and each combination of RCP scenario with the TGL

parameterisation. An emulator is built for each of these model configurations from an ensemble of 500 training points (hence 500 forward simulations) in the parameter space of the parameters $F_{\mathrm{calv}}$, $F_{\mathrm{melt}}$, $E_{\mathrm{shelf}}$, $\tau_{\mathrm{e}}$ and $\tau_{\mathrm{w}}$ with a maximin Latin hypercube sampling design. In total, we carried out 10000 forward simulations of the f.ETISh model for the 20 model configurations. More details about the construction of the PC expansion are given in Appendix A.

We have changed the paragraph and given more details about the construction of the PC expansion. See also response to anonymous referee #1 (https://doi.org/10.5194/tc-2018-220-AC1) for more details on the experimental set-up.

- p.11 l.3–6: Change "Under nominal conditions, we find (Table 2) in RCP 2.6 a nominal AIS contribution to sea-level rise of 0.02 m by 2100, 0.07 m by 2300 and 0.20 m by 3000 and in RCP 8.5 a nominal AIS contribution to sea-level rise of 0.05 m by 2100, 0.59 m by 2300 and 3.9 m by 3000." to "Under nominal conditions, we find (Table 2) for RCP 2.6 an AIS contribution to sea-level rise of 0.02 m by 2100, 0.07 m by 2300 and 0.20 m by 3000 and for RCP 8.5 an AIS contribution to sea-level rise of 0.05 m by 2100, 0.59 m by 2300 and 3.9 m by 3000."

The sentence has been changed based on the referee's suggestion (see answer about p.11 l.19 for the use of "in RCP" instead of "for RCP").

- p.11 l.19 (and elsewhere): I'm not a native English speaker, but I think "Figure 4a–c shows that in RCP 2.6" should read "Figure 4a–c shows that for RCP 2.6" or "Figure 4a–c shows that under RCP 2.6".

We thank the referee for this comment. None of the authors is also a native English speaker. Throughout the manuscript, we tried to be consistent and use the same

preposition "in" every time we refer to a RCP scenario ("in RCP 2.6" stands for an ellipsis for "in the scenario RCP 2.6"). We did not find a general consensus in the literature regarding the use of a specific preposition when referring to the RCP scenarios (Golledge et al., 2015; De Conto and Pollard, 2016).

- p.11 l.23: "This quadratic dependence can be explained by the influence of $E_{\text{shelf}}$ on two competing processes: a higher value of $E_{\text{shelf}}$ softens the ice, thus leading to faster ice flow in the ice shelves; but a higher value of $E_{\text{shelf}}$ also leads to ice-shelf thinning, thus reducing grounding line ice flux."

The sentence has been changed based on the referee's suggestion.

- p.12 l.18: you use "median" but provide a range. Please clarify. Do the ranges represent the 16/84th percentiles, for example?

This range is simply the range of values taken by the median values for the three sliding exponents $m = 1, 2, 3$ as shown in Table 3. A formulation such that "a median AIS contribution to sea-level rise of 0.07–0.13 m in RCP 2.6 by 2300" takes into account the median value 0.07 m for $m = 1$, 0.13 m for $m = 2$ and 0.09 m for $m = 3$. We have changed our sentence following the referee's comment to avoid any ambiguity.

- p.12 l.22-23: "...except for certain cases ..."

We have removed the word "a" from the sentence.

- p.12 l.26-27: "...and starts to increase around 2100 ..."

We have changed the phrase based on the referee's suggestion.

- p.12 l.29: maybe reference Fig. 6 here?

We agree with this suggestion. We have added the suggested reference at the end of the sentence.

- p.12 l.31: ". . . which contributes 3–3.5 metres to sea-level followed by a slower retreat of the East Antarctic ice sheet."

We have changed the phrase following the referee's suggestion.

- p.12 l.33: "by the year 3000"

We have changed the phrase following the referee's suggestion.

- p.13 l.1: "For RCP 2.6, we find (Table 3) 5–95 % probability intervals . . . "

We have changed the sentence following the referee's suggestion (see also answer about p.11 l.19 for the use of "in RCP" instead of "for RCP").

- p.13 l.4: ". . . an increase in sea-level, though a decrease cannot be ruled out for a viscous sliding law and cooler atmospheric conditions."

We have simplified the sentence based on the referee's suggestion.

- p.14 l.6: It is interesting that for RCP 2.6 rheology contributes a similar amount 40–60

We agree with this comment. Maybe this suggests the need to properly calibrate $E_{\text{shelf}}$ alongside the sliding coefficient in the initialisation procedure.

- p.15 l.23: "ice is certain to . . . " I would refrain from using strong words like "certain".

We agree that using strong words like "certain" is probably not relevant and could lead to misinterpretations. Hence, we have removed the word "certain" as suggested by the referee.

- p.17 l.4–10: Rather than writing down numbers (which are listed in Table 4 anyway), I suggest to tell the reader how the plastic sliding law compares to the intermediate case since this is what one cares about.

We agree with the referee that all these numbers do not bring additional information to the manuscript (as they are listed in Table 4). We have removed all these numbers and written down the paragraph as a discussion about the difference with the other sliding laws.

- p.17 l.18: ". . . , which could explain the smaller ice loss in our results under $m = 5$ than $m = 3$." That's interesting, I would not have guessed.

We thank the referee for this comment.

- p.19 l.11: "the pivotal role played by atmospheric forcing". I think you mean the role of the emission scenario.

The referee is right in his interpretation. We have changed our phrase according to his suggestion to make it clearer.

- p.19 l.26: Moreover, the lower sensitivity of the Amundsen Sea sector in our simulations may arise...

We have changed the phrase "in our findings" by "in our simulations" as suggested by the referee.

- p.20 l.6–10: Very long sentence, maybe split into two. Figures

We agree with this suggestion. The sentence has been split accordingly.

- p.20 l.23–29: I understand what you are trying to say but I'm not sure the formulation "does (not) question the nominal projections". Maybe "in agreement with" or similar? The probabilistic framework expands upon the nominal projections, and your results provide good evidence that a thorough risk assessment must include UQ.

The referee is right in his interpretation of this sentence. We agree with his suggestion to change the formulation of the sentence. We have changed the sentence as "...accommodating parametric uncertainty in the ice-sheet model leads to projections in agreement with the nominal projections of limited ice loss ..."

- p.21 l.20: We found that all investigated sources ...

We agree with this suggestion. The sentence has been changed accordingly.

- Fig. 3 b–e: Show outline of present-day grounded area for better visual comparison.

We have added the present-day grounding-line position as suggested by the referee.

- Fig. 9: What does the $m$ in the lower left corners of the plot mean?

The parameter $m$ is the sliding exponent in Weertman's sliding law. We have replaced the phrase "under different sliding laws" by "for different values of the sliding exponent m in Weertman's sliding law" to remind the reader about the meaning of $m$.

**5 References**

Aschwanden, A., Fahnestock, M. A., and Truffer, M.: Complex Greenland outlet glacier flow captured, Nat. Commun., 7, https://doi.org/10.1038/ncomms10524, 2016.

DeConto, R. M. and Pollard, D.: Contribution of Antarctica to past and future sea-level rise, Nature, 531, 591–597, https://doi.org/10.1038/nature17145, 2016.

Gillet-Chaulet, F., Durand, G., Gagliardini, O., Mosbeux, C., Mouginot, J., Rémy, F., and Ritz, C.: Assimilation of surface velocities acquired between 1996 and 2010 to constrain the form of the basal friction law under Pine Island Glacier, Geophys. Res. Lett., 43, 10,311–10,321, https://doi.org/10.1002/2016gl069937, 2016.

[Figure]

Golledge, N. R., Kowalewski, D. E., Naish, T. R., Levy, R. H., Fogwill, C. J., and Gasson, E. G. W.: The multi-millennial Antarctic commitment to future sea-level rise, Nature, 526, 421–425, https://doi.org/10.1038/nature15706, 2015.

Golledge, N. R., Levy, R. H., McKay, R. M., and Naish, T. R.: East Antarctic ice sheet most vulnerable to Weddell Sea warming, Geophys. Res. Lett., 44, 2343–2351, https://doi.org/10.1002/2016gl072422, 2017.

Kapur, J.N.: Maximum-Entropy Models in Science and Engineering.Wiley, New Delhi, 1989.

Ritz, C., Edwards, T. L., Durand, G., Payne, A. J., Peyaud, V., and Hindmarsh, R. C. A.: Potential sea-level rise from Antarctic ice-sheet instability constrained by observations, Nature, 528, 115–118, https://doi.org/10.1038/nature16147, 2015.

Schlegel, N.-J., Seroussi, H., Schodlok, M. P., Larour, E. Y., Boening, C., Limonadi, D., Watkins, M. M., Morlighem, M., and van den Broeke, M. R.: Exploration of Antarctic Ice Sheet 100-year contribution to sea level rise and associated model uncertainties using the ISSM framework, The Cryosphere, 12, 3511–3534, https://doi.org/10.5194/tc-12-3511-2018, 2018.

[Figure]

(a)

[Figure]

(b)

[Figure]

**Fig. 1.**

---

## Referee Report (RR1)

**Review of "Uncertainty quantification if the multi-centennial response of the Antarctic Ice Sheet to climate change"**

**1 Summary statement**

The revisions made to "Uncertainty quantification if the multi-centennial response of the Antarctic Ice Sheet to climate change" address the comments raised by the two reviewers, with additional details and clarifications on the impact of model resolution, grounding line parameterization, ocean warming and uncertainty quantification methods. I am still a little skeptical about the accuracy of a 20 km resolution model to correctly represent relatively small outlet glaciers and to accurately simulate the evolution of the grounding lines around the Antarctic ice sheet. This is discussed at length in the new version of the manuscript and seems supported by previously published results, but I would have appreciated to see some runs performed with a different resolution to confirm this point and to provide an attempt to quantify the impact of the resolution used in these simulations on sea level estimates.

**2 Specific comments**

Page and line numbers refer to the version of the manuscript with changes tracked.

p.1 l.15: "the marine ice-sheet instability"

p.11 l.14: "a shelf tune" → "an ice shelf tune"

p.15 l.8-11 (and also later in the text, at least in sections 3.6 and 3.8): use either "Figure XX shows" or "Figures XX show".

p.16 l.15: Is it 2500 or 3000?

p.20 l.25: How large is the difference between the modeled and emulated sea level? Is that what is show on Fig.A1d? I would expect the difference between the modeled and emulated results to be almost negligible compared to the signal estimated, because a mean difference of 0.2 m seems relatively large, but I am not entirely sure what is the error presented on this figure.

---

## Author Response (AR2)

***Response to Review of*** "Uncertainty quantification of the multi-centennial response of the Antarctic Ice Sheet to climate change"

We would like to thank anonymous referee #1 for this additional review regarding the quality and readibility of the manuscript. We will try to give a proper response to his/her suggestions. For each referee's comment (written in blue), we included below a response (written in black) and proposed means to improve the manuscript.

**1 Summary statement**

The revisions made to "Uncertainty quantification if the multi-centennial response of the Antarctic Ice Sheet to climate change" address the comments raised by the two reviewers, with additional details and clarifications on the impact of model resolution, grounding line parameterisation, ocean warming and uncertainty quantification methods. I am still a little skeptical about the accuracy of a 20 km resolution model to correctly represent relatively small outlet glaciers and to accurately simulate the evolution of the grounding lines around the Antarctic ice sheet. This is discussed at length in the new version of the manuscript and seems supported by previously published results, but I would have appreciated to see some runs performed with a different resolution to confirm this point and to provide an attempt to quantify the impact of the resolution used in these simulations on sea level estimates.

We are pleased that the referee is satisfied with the changes during the review process. In our response to referee #2 (https://doi.org/10.5194/tc-2018-220-AC2), we provided a comparison for the AIS contribution to sea level as a function of spatial resolution (20 km in Fig. 1a vs 16 km in Fig. 1b) to give an idea about the impact of the model resolution on our results. This figure suggests that the uncertainty in the projections due to the model resolution is (far) less important than the uncertainty due to the uncertainty in the parameters. The figure is available in the revised version of the supplementary material but an explicit reference to this figure in the manuscript was lacking. We have now explicitly stated this comparison in the manuscript.

**2 Specific comments**

Page and line numbers refer to the version of the manuscript with changes tracked

p.1 l.15: "the marine ice-sheet instability"

The phrase has been changed based on the referee's suggestion.

p.11 l.14: "a shelf tune" → "an ice shelf tune"

The phrase has been changed based on the referee's suggestion.

p.15 l.8-11 (and also later in the text, at least in sections 3.6 and 3.8): use either "Figure XX shows" or "Figures XX show".

We have decided to write "Figures XX show" all along the manuscript.

We thank the referee for identifying this typo. The results are for 3000 and not 2500.

p.20 l.25: How large is the difference between the modeled and emulated sea level? Is that what is show on Fig.A1d? I would expect the difference between the modeled and emulated results to be almost negligible compared to the signal estimated, because a mean difference of 0.2 m seems relatively large, but I am not entirely sure what is the error presented on this figure.

In Sect. 3.6, the differences between the nominal projections as given by the f.ETISh model and the emulator are generally of a few centimetres. For instance, the projection of $\Delta$GMSL by 3000 in RCP 4.5 is of 0.39 m for the f.ETISh model and of 0.37 m for the emulator.

The referee is right in interpreting Fig. 1Ad as a measure of the accuracy of the emulator. Yet, it does not represent the differences between the nominal projections as given by the f.ETISh model and the emulator. In this figure, we showed a convergence test for RCP 8.5 at time 3000 based on the number of training points used to build the emulator. The blue curve represents the mean-squared error between the f.ETISh model and the emulator at a set of cross-validation points as a function of the number of training points while the red curve represents the maximum absolute error for the same set of cross-validation points. The mean-squared error is of the order of 0.2 m for this scenario. We believe such an error to be acceptable for a median response of over 5 m.

***Response to editor decision of*** **"Uncertainty quantification of the multi-centennial response of the Antarctic Ice Sheet to climate change"**

We would like to thank Olivier Gagliardini for the publication of our manuscript and the suggestions for minor revisions. For each editor's comment (written in blue), we included below a response (written in black) and proposed means to improve the manuscript.

**1 Comments to the author**

Dear Kevin,

Thanks for this revised version of your paper. The reviewer is satisfied with the changes made during the review process. I am therefore happy to accept your paper for publication in The Cryosphere.

The reviewer has only few minor remarks that you should account for before sending your last version. I have also added two minor comments below.

Thanks for having choosing The Cryosphere to publish this very nice piece of work!

Regards,
Olivier Gagliardini

We would like to thank Olivier Gagliardini for the time dedicated to this manuscript and these kind comments. We have included in the last version of the manuscript the referee's minor remarks and the editor's minor comments (see below).

**2 Specific comments**

p.12 l.11: may be repeat that TGL is only for $m = 5$ so that it is easier to understand that $20 = 4 \times 4 + 4$!

We have added this clarification in the manuscript based on the referee's suggestion.

p.20 l.20: results under Schoof's parameterisation -> results under SGL parameterisation

The phrase has been changed based on the referee's suggestion.

[revised manuscript text omitted]